# In-Context Linear Regression Demystified: Training Dynamics and Mechanistic Interpretability of Multi-Head Softmax Attention

**Jianliang He** [1]  **Xintian Pan** [2]  **Siyu Chen** [1]  **Zhuoran Yang** [1]

## Abstract

We study how multi-head softmax attention models are trained to perform in-context learning on linear data. Through extensive empirical experiments and rigorous theoretical analysis, we demystify the emergence of elegant attention patterns: a diagonal and homogeneous pattern in the key-query weights, and a last-entry-only and zero-sum pattern in the output-value weights. Remarkably, these patterns consistently appear from gradient-based training starting from random initialization. Our analysis reveals that such emergent structures enable multi-head attention to approximately implement a debiased gradient descent predictor — one that outperforms single-head attention and nearly achieves Bayesian optimality up to proportional factor. We also extend our study to scenarios with anisotropic covariates and multi-task linear regression. Our results reveal that in-context learning ability emerges from the trained transformer as an aggregated effect of its architecture and the underlying data distribution, paving the way for deeper understanding and broader applications of in-context learning. Our code is available at https://github.com/XintianPan/ICL_linear.

## 1. Introduction

Large language models (LLMs) built on transformer architectures (Vaswani, 2017) have revolutionized artificial intelligence research. A key capability of these models is their ability to perform in-context learning (ICL) (Dong et al., 2022; Brown et al., 2020), which refers to learning and adapting to new tasks or concepts simply by being provided with a few examples or instructions within the input context, without the need for explicit retraining or fine-tuning. Unlike traditional approaches that rely on extensive fine-tuning, ICL enables LLMs to infer patterns directly from input sequences and generalize to unseen examples in a single forward pass (Huang et al., 2022). This emergent behavior is largely attributed to the self-attention mechanisms in transformers, which allow the models to dynamically capture intricate relationships within the data. Recently, there has been growing interest in exploring transformer-based ICL in structured learning settings, such as linear regression, using synthetic data (Bai et al., 2024; Akyürek et al., 2022; Ahn et al., 2023a; Fu et al., 2023; Mahankali et al., 2023). These studies not only shed light on the inner workings of transformers but also deepen our understanding of their statistical properties and optimization dynamics, offering a foundation for more interpretable and efficient AI systems.

Prior work has extensively studied how transformers perform in-context learning for linear regression, exploring various attention architectures and learning strategies. It has been shown that single-head linear attention effectively implements preconditioned gradient descent on linear data (Zhang et al., 2024; Von Oswald et al., 2023; Akyürek et al., 2022). This analysis has been extended to single-head softmax transformers (Li et al., 2024a; Huang et al., 2023) as well as multi-head attention models (Chen et al., 2024a;c; Deora et al., 2023; Kim & Suzuki, 2024). Beyond linear regression, recent studies have investigated how transformers learn feature representations in more complex settings, extending in-context learning to nonlinear data distributions (Huang et al., 2023; Yang et al., 2024; Kim & Suzuki, 2024; Chen et al., 2024b).

Despite significant progress in understanding transformer-based in-context learning (ICL), critical gaps persist in characterizing the training dynamics of multi-head softmax transformers for these tasks. While prior work has yielded insights under constrained theoretical regimes—including the neural tangent kernel (NTK) framework (Deora et al., 2023), mean-field approximations (Kim & Suzuki, 2024), and specialized initialization protocols (Chen et al., 2024a)—fundamental questions about their broader behavior remain unanswered:

---

[1]Department of Statistics and Data Science, Yale University
[2]Kuang Yaming Honors School, Nanjing University. Correspondence to: Jianliang He <jianliang.he@yale.edu>, Zhuoran Yang <zhuoran.yang@yale.edu>.

*Proceedings of the $42^{nd}$ International Conference on Machine Learning*, Vancouver, Canada. PMLR 267, 2025. Copyright 2025 by the author(s).

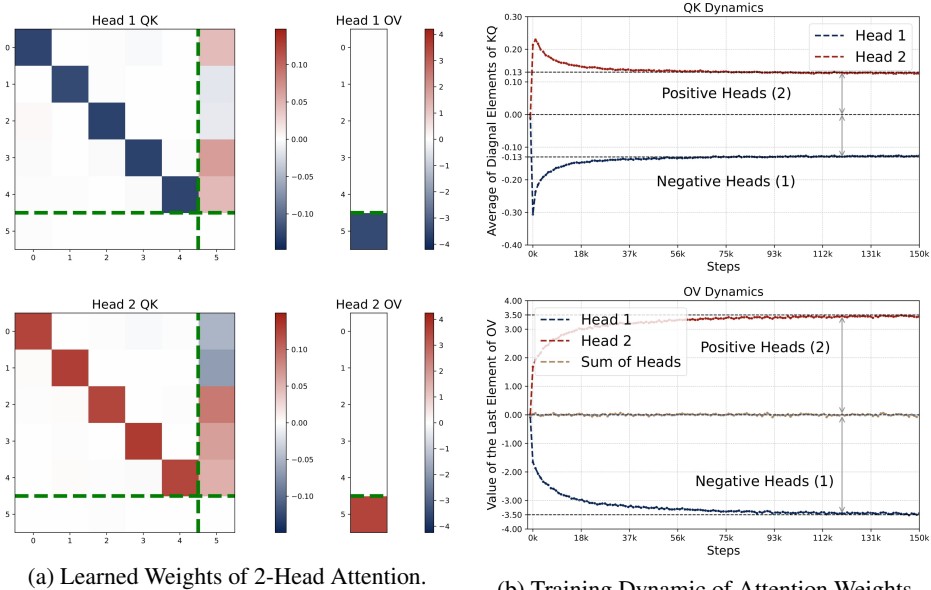

(a) Learned Weights of 2-Head Attention.

(b) Training Dynamic of Attention Weights.

*Figure 1.* Visualization of main results. Panel (a) presents the heatmaps of 2-head attention matrices trained over $d = 5$, $L = 40$. The trained model learns one positive and one negative head. KQ and OV share the same sign for each head, while the parameter scaling is homogeneous across heads. Panel (b) plots the dynamics of average diagonal values in KQ matrices and the last entry of OV vectors for a 2-head attention model along the training, zooming into the opposite pattern.

(i) How do the weights of randomly initialized multi-head softmax transformers evolve during training on linear ICL tasks?

(ii) What is the final learned transformer model and what are its statistical properties under standard training?

(iii) Does multi-head attention confer measurable advantages over single-head architectures for linear tasks?

(iv) Does softmax attention outperform linear attention in linear ICL—particularly given the latter's proven equivalence to gradient-based optimization (e.g., Von Oswald et al., 2023)?

(v) How do the insights from linear ICL with isotropic covariates extend to scenarios involving anisotropic covariates or multiple linear regression tasks?

We answer these questions through both empirical and theoretical analysis of training a multi-head softmax transformer on linear regression task. We characterize the training dynamics, demystify the emergence of distinct weight patterns in learned models, and elucidate the underlying mechanisms. We summarize our contributions as follows.

(i) We empirically analyze the training dynamics of a multi-head softmax transformer on a single linear regression task. We find that in each head, the key-query (KQ) and output-value (OV) matrices exhibit a universal structure that implements a kernel regressor, where the kernel is determined by the query and covariates.

(ii) We identify two global patterns emerging: (a) The KQ weight matrices develop a diagonal structure for the block associated with the covariates and the query. (b) The diagonal entries of the KQ weights and the effective OV weights share the same sign. These patterns lead to the automatic grouping of positive and negative heads, allowing the model to effectively capture both positive and negative components. See Figure 1.

(iii) In the trained model, the diagonal KQ weights across all heads are nearly homogeneous in magnitude, and the average effective OV weight is approximately zero. We provide theoretical insights into how these patterns emerge through training dynamics.

(iv) We discover the positive and negative heads jointly approximate a one-step debiased gradient descent—letting multi-head transformers outperform single-head models on linear ICL—and the learned model is provably nearly Bayesian-optimal up to a constant factor.

(v) Furthermore, we show that softmax attention outperforms linear attention in linear ICL tasks, as it learns an algorithm that successfully generalizes to longer sequences at test time.

(vi) We extend our analysis to anisotropic covariates and multi-task settings. In the anisotropic case, we find that the learned model s the learned model implements a pre-conditioned version of debiased gradient descent. In the multi-task setting, the model behavior depends on the task-to-head ratio. Interestingly, when the num-

ber of tasks exceeds the number of heads but remains below twice that number, we observe a superposition phenomenon, where individual heads encode multiple tasks simultaneously.

**Related Work.** Several works have probed transformers on linear-regression tasks to elucidate their ICL capabilities. Empirically, Garg et al. (2022) demonstrate near Bayes-optimal performance, and Von Oswald et al. (2023) show that a simplified linear transformer implements a gradient-based inference algorithm. Theoretically, one-layer linear attention provably learns preconditioned gradient descent via training dynamics analysis in Zhang et al. (2024) and loss-landscape analysis in Ahn et al. (2023a). Chen et al. (2024a) provide the first insight into the standard softmax attention, proving convergence to a kernel regressor under specified initialization. In addition, Bai et al. (2024) characterizes the expressive power of transformers for various linear regression algorithms. Extensions to more complex settings include two-stage least squares for endogeneity (Liang et al., 2024), adaptive sparse regression (Chen et al., 2024c), EM-based mixture modeling (Jin et al., 2024), multi-step gradient-descent (Gatmiry et al., 2024), and analyses of the nonlinear softmax component in regression tasks (Aksenov et al., 2024; Sun et al., 2025).

More related works and notation is provided in §A.

## 2. Preliminaries

We consider the setting of training transformers over the task of in-context linear regression, following the framework widely considered in literature (e.g., Garg et al., 2022).

**Data Distribution and Embedding.** Let $x_\ell \in \mathbb{R}^d$ denote the $\ell$-th covariate i.i.d drawn from a distribution $\mathsf{P}_x$ and let $x_q \overset{\text{i.i.d.}}{\sim} \mathsf{P}_x$ represent a new test input. The coefficient $\beta \in \mathbb{R}^d$ is sampled from $\mathsf{P}_\beta$, and the corresponding response is generated as $y_\ell = \beta^\top x_\ell + \epsilon_\ell$, where the noise $\epsilon_\ell$ i.i.d sampled from $\mathcal{N}(0, \sigma^2)$. To perform ICL, we embed the dataset $\{(x_\ell, y_\ell)\}_{\ell \in [L]}$ along with the test input $x_q \sim \mathsf{P}_x$ into a sequence $Z_{\text{ebd}} \in \mathbb{R}^{(d+1) \times (L+1)}$, formatted as follows:

$$Z_{\text{ebd}} = \begin{bmatrix} Z & z_q \end{bmatrix} = \begin{bmatrix} x_1 & x_2 & \dots & x_L & x_q \\ y_1 & y_2 & \dots & y_L & 0 \end{bmatrix}, \quad (2.1)$$

where we also denote by $X = [x_1, x_2, \dots, x_L]^\top \in \mathbb{R}^{L \times d}$ and $y = [y_1, y_2, \dots, y_L]^\top \in \mathbb{R}^L$. In this paper, we focus on the isotropic case where $\mathsf{P}_x = \mathcal{N}(0, I_d)$ and $\mathsf{P}_\beta = \mathcal{N}(0, I_d/d)$, and the non-isometric cases are in §B.3.

**Multi-Head Softmax Attention.** The attention model is a sequence-to-sequence model that takes $Z_{\text{ebd}}$ as input and outputs a sequence of the same shape. We focus on the one-layer softmax attention with $H$ heads, parametrized by

$\theta = \{O^{(h)}, V^{(h)}, K^{(h)}, Q^{(h)}\}_{h \in [H]} \subseteq \mathbb{R}^{(d+1) \times (d+1)}$:

$$\text{TF}_\theta(Z_{\text{ebd}}) = Z_{\text{ebd}} + \sum_{h=1}^{H} O^{(h)} V^{(h)} Z_{\text{ebd}}$$
$$\cdot \text{smax} \circ \text{msk}(Z_{\text{ebd}}^\top K^{(h)^\top} Q^{(h)} Z_{\text{ebd}}), \quad (2.2)$$

where $\text{smax}(\cdot)$ denotes the column-wise softmax operation and $\text{msk}(\cdot)$ dontes the element-wise causal mask, i.e., $\text{msk}(\cdot)_{ij} = \mathbb{1}(i < j) - \infty \cdot \mathbb{1}(i \geq j)$. Since the $(K, Q)$ and $(O, V)$ matrices always appear in tandem, we simplify the notation by grouping them into single matrices $KQ$ and $OV$, respectively (Elhage et al., 2021). Following the embedding in (2.1), we extract the prediction $\hat{y}_q$ from the $(d + 1, L + 1)$-th entry. The goal is to estimate the conditional expectation $\mathbb{E}[y_q \mid x_q] = \beta^\top x_q$. Accordingly, we define the model output as

$$\hat{y}_q := \hat{y}_q(x_q; \{(x_\ell, y_\ell)\}_{\ell \in [L]}) = \text{TF}_\theta(Z_{\text{ebd}})_{d+1, L+1} \in \mathbb{R}.$$

Compared to the standard decoder-only transformer model (e.g., Vaswani, 2017), we omit layer normalization and positional embeddings. This choice arises because a single-layer architecture does not suffer from vanishing or exploding gradients. In addition, the associated linear regression problem is permutation-invariant in $\{(x_\ell, y_\ell)\}_{\ell \in [L]}$, making positional information unnecessary.

**KQ and OV Circuits.** Note the computation in (2.2) only depends on matrix products $K^{(h)^\top} Q^{(h)}$ and $O^{(h)} V^{(h)}$ respectively. Following Elhage et al. (2021), we refer to these matrices as the KQ and OV circuits and simplify the notation as $KQ^{(h)} = K^{(h)^\top} Q^{(h)}$ and $OV^{(h)} = O^{(h)} V^{(h)}$. In specific, KQ circuits characterize to what extent a query token attends to a key token, and OV circuits determine how a token contributes to the output when attended to.

Notice that the last entry of $z_q$ is zero and we use the last row of $O^{(h)}$ to get $\hat{y}_q$. Thus, in terms of computing $\hat{y}_q$, we can write the KQ and OV circuits as

$$KQ^{(h)} = \begin{bmatrix} KQ_{11}^{(h)} & * \\ KQ_{21}^{(h)} & * \end{bmatrix}, OV^{(h)} = \begin{bmatrix} * & * \\ OV_{21}^{(h)} & OV_{22}^{(h)} \end{bmatrix}.$$

Here we use "$*$" to denote the entries that do not affect $\hat{y}_q$. In the above two-by-two block matrix format, the top-left blocks are $d$-by-$d$ matrices. See Figure 6 for a complete depiction of the transformer architecture.

**Training Setup.** To investigate how transformers learn to solve the linear regression problem in context during pretraining, we examine both the training dynamics and loss landscape. Specifically, we train the model by minimizing the population mean-squared error:

$$\mathcal{L}(\theta) = \mathcal{E}(\hat{y}_q) = \mathbb{E}[(y_q - \text{TF}_\theta(Z_{\text{ebd}})_{d+1, L+1})^2], \quad (2.3)$$

where the expectation is taken w.r.t $\beta \sim \mathsf{P}_\beta$ and $(x_\ell, y_\ell) \overset{\text{i.i.d.}}{\sim} \mathsf{P}_x \otimes \mathsf{P}_{y|x}(\cdot; \beta)$ for all $\ell \in [L] \cup \{q\}$.

## 3. Empirical Insights

Although previous studies have explored how transformers perform ICL under the linear regression framework, much of this research has been limited to experimental analyses (Garg et al., 2022), linear transformers (Von Oswald et al., 2023), or single-head attention (Huang et al., 2023). This leaves a gap in understanding multi-head softmax attention models, which are used in practice. To this end, we first conduct experiments to empirically investigate how multi-head softmax attention models learn to solve ICL with linear data. See §C.1 for experimental setup and additional results.

**Observation 1.** For any number of heads with $H \geq 1$, in the trained one-layer multi-head attention model, the KQ and OV circuits take the following form:

$$KQ^{(h)} = \begin{bmatrix} \omega^{(h)} I_d & * \\ \mathbf{0}_d^\top & * \end{bmatrix}, OV^{(h)} = \begin{bmatrix} * & * \\ \mathbf{0}_d^\top & \mu^{(h)} \end{bmatrix}. \quad (3.1)$$

Moreover, KQ and OV share the same signs within each head, i.e., $\text{sign}(\omega^{(h)}) = \text{sign}(\mu^{(h)})$. We refer to this property as sign-matching. In some cases, dummy heads may emerge, where $\omega^{(h)} \approx 0$ and $\mu^{(h)} \approx 0$. Notably, these patterns develop early in training and remain consistent throughout the optimization process.

We illustrate the KQ and OV circuit patterns for both one-head and two-head attention models in Figures 2a and 1a, respectively. For the OV circuits, we only plot the transpose of the last rows, i.e., $OV_{21}^{(h)}$ and $OV_{22}^{(h)}$. These figures show that the trained multi-head softmax attention models with various numbers of heads exhibit a consistent pattern:

(a) For each KQ circuit, the top-left $d$-by-$d$ submatrix is diagonal and proportional to an identity matrix, i.e., $KQ_{11}^{(h)} = \omega^{(h)} \cdot I_d$ for some $\omega^{(h)} \in \mathbb{R}$. Moreover, the first $d$ entries of the last row are all approximately zero, i.e., $KQ_{21}^{(h)} = \mathbf{0}^\top$.

(b) The last column of each OV circuit, as a vector in $\mathbb{R}^{d+1}$, admits a last-entry-only pattern. That is, only the last entry is non-zero, which is given by $\mu^{(h)} \in \mathbb{R}$.

(c) We observe that $\mu^{(h)}$ and $\omega^{(h)}$ always have the same sign. Thus, each head can be categorized into either a positive or negative head, depending on the sign of $\omega^{(h)}$. Positive and negative heads are defined as $\mathcal{H}_+ = \{h \colon \omega^{(h)} > 0\}$ and $\mathcal{H}_- = \{h \colon \omega^{(h)} < 0\}$.

The pattern of KQ and OV circuits shows that the weight matrices of the learned transformer essentially are governed by $2H$ numbers $\mu = (\mu^{(1)}, \dots, \mu^{(H)})^\top$ and $\omega =$

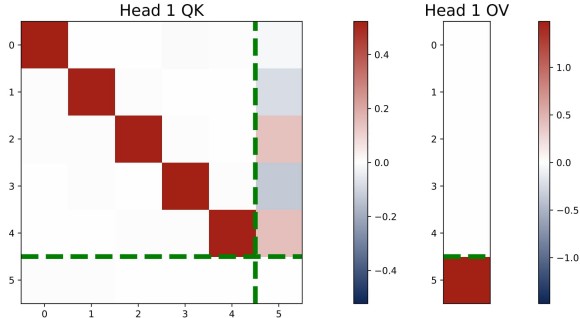

(a) Learned Weights of Single-head Attention.

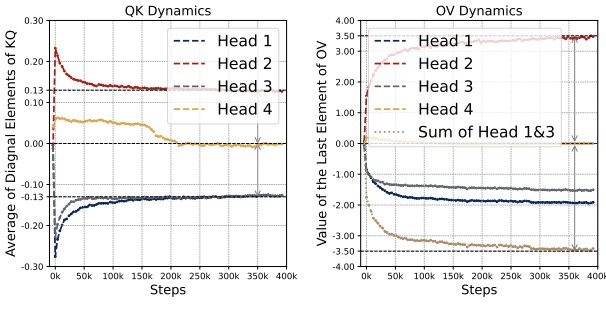

(b) 4-Head KQ Dynamics.   (c) 4-Head OV Dynamics.

*Figure 2.* Learned pattern of single-head attention and training dynamics of 4-head model. Panel (a) plots the heatmap of 1-head attention trained under the same setting as in Figure 1a. Panel (b), and (c) plot the dynamics of average diagonal values in KQ matrices and the last entry of OV vectors for a 2-head attention model along the training. Head 2 is the positive head, Head 1&4 are coupled to act as the negative head, and Head 3 is dummy.

$(\omega^{(1)}, \dots, \omega^{(H)})^\top$. Under such a structure, the transformer predictor takes the form:

$$\widehat{y}_q = \sum_{h=1}^{H} \mu^{(h)} \cdot \left\langle y, \mathtt{smax}(\omega^{(h)} \cdot X x_q) \right\rangle$$

$$= \sum_{h=1}^{H} \mu^{(h)} \cdot \sum_{\ell=1}^{L} \frac{y_\ell \cdot \exp(\omega^{(h)} \cdot x_\ell^\top x_q)}{\sum_{\ell=1}^{L} \exp(\omega^{(h)} \cdot x_\ell^\top x_q)} \in \mathbb{R}. \quad (3.2)$$

Thus, each head acts as a separate kernel regressor, and thus the attention model can be interpreted as the sum of kernel regressors (see §E.1). Here, $1/\omega^{(h)}$ plays the role of bandwidth of the kernel. Each term in the sum in (3.2) is a weighted sum of responses $\{y_\ell\}_{\ell \in [L]}$ in the ICL samples, and the weights are computed based on the similarity between the test input $x_q$ and the sample covariate $x_\ell$'s. The notion of similarity is governed by the parameter $\omega^{(h)}$'s.

The single-head case has been studied in (Chen et al., 2024a), which shows that the single-head model functions as a kernel estimator to solve linear regression, where $\omega^{(1)} = 1/\sqrt{d}$ and $\mu^{(1)} \asymp \sqrt{d}$ when $L$ is large. We replicate this finding in Figure 2a, where $\omega^{(1)} \approx 0.52 \approx 1/\sqrt{d}$. Additionally, we show that such a pattern is shared by multi-head

attention models with $H \geq 2$.

Moreover, when $H > 2$, as we show in Figure 13, this pattern persists. Additionally, we see a dummy head (Head 4 in this case), i.e., a head whose KQ and OV matrices are close to zero. This means that this head does not contribute to the prediction. However, the appearance of dummy heads is not guaranteed, even with a large number of heads, as it results from the randomness of the optimization algorithm. As we will show later, dummy heads do not affect the statistical properties of the learned model and thus can be ignored in statistical analysis.

Finally, in Figure 4 and 13, we plot the KQ and OV circuits of two-head and four-head attention models, respectively, along the training trajectory. We observe that the patterns in (3.1) emerge early during training and persist throughout.

**Observation 2.** When $H \geq 2$, $(\omega, \mu)$ of the learned attention model satisfies that

 (i) Homogeneous KQ scaling: The scaling of the top-left diagonal submatrix of each $KQ^{(h)}$ is nearly identical across all positive and negative heads, i.e., $|\omega^{(h)}| \approx \gamma$ for all $h \in \mathcal{H}_+ \cup \mathcal{H}_-$.
 (ii) Zero-sum OV: The sum of $\{\mu^{(h)}\}_{h\in[H]}$ is approximately zero, i.e., $\langle \mu, \mathbf{1}_H \rangle \approx 0$.

By examining Figures 1a more closely, the two attention heads in the two-head attention model converge to opposite heads, i.e., $\omega^{(1)} \approx -\omega^{(2)}$ and $\mu^{(1)} \approx -\mu^{(2)}$. Moreover, as we show in Figures 2b and 2c, similar patterns emerge in the four-head attention model. In particular, Head 4 is a dummy head, i.e., $\omega^{(4)} \approx \mu^{(4)} \approx 0$. Head 1 and Head 3 are negative heads and Head 2 is a positive head. Thus, we conclude

 (a) For non-dummy heads, the scaling of the non-zero entries of the KQ circuits is nearly identical across heads, i.e., $|\omega^{(h)}| \approx \gamma$ for all $h \in \mathcal{H}_+ \cup \mathcal{H}_-$.
 (b) For the OV circuits, it holds that $\langle \mu, \mathbf{1}_H \rangle \approx 0$ such that $\sum_{h\in\mathcal{H}_+} \mu^{(h)} \approx -\sum_{h\in\mathcal{H}_-} \mu^{(h)}$.

In single-head attention, the attention head is either positive or negative. The multi-head attention models have both positive and negative heads, with special patterns in $(\omega, \mu)$.

**Observation 3.** In terms of the ICL prediction error, the two-head attention model outperforms single-head model. Moreover, multi-head models with $H \geq 2$ exhibit similar performance, closely approximating that of the vanilla gradient descent (GD) predictor $\widehat{y}_q^{\text{vgd}} = L^{-1} \cdot \sum_{\ell=1}^{L} x_\ell^\top x_q \cdot y_\ell$. Furthermore, all softmax attention models can generalize in length during the test time. In addition, the statistical error of the multi-head attention model is comparable to the optimal Bayes estimator up to a proportionality factor.

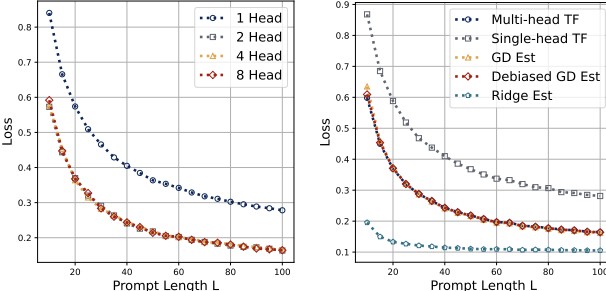

(a) Loss of Attention Model with Different Head Number.

(b) Loss of Attention Models and Canonical Estimators.

*Figure 3.* Loss comparison of different models with different numbers of heads and the canonical estimators. The models are trained over $L = 40$ and evaluated over different input lengths.

Comparing Figures 1a with 2a, we observe that in the two-head attention model, the magnitude of KQ circuits ($\sim 0.13$) is smaller than that of the single-head attention ($\sim 0.52$), and the magnitude of OV circuits ($\sim 3.50$) is larger than that of the single-head attention model ($\sim 1.42$). Hence, the two-head attention model learns a different predictor than the single-head attention model.

More interestingly, revisiting the patterns learned in Figures 1b, 2b, and 2c we see that the two-head and four-head attention model learns the same predictor. To see this, note that Figures 1b and 2b have the same scaling in the KQ circuits ($\sim 0.13$). Similarly, the magnitude of OV circuits aligns across positive and negative groups ($\sim 3.5$). This implies that the four-head attention model ultimately converges to the same predictor as the two-head model. More broadly, this phenomenon holds across all multi-head attention models with $H \geq 2$ which consistently learn nearly identical predictors. Furthermore, to study the statistical errors of these learned attention models, we compare their ICL loss in the test time, defined in (2.3). In particular, we consider length generalization where $Z_{\text{ebd}}$ in (2.1) involves a different length $L$ that might be different from the training data with $L = 40$. We plot the ICL losses of various models in Figure 3a, which reveals the following findings:

 (a) Multi-head attention outperforms single-head attention while maintaining nearly identical performance across different the number of heads used.

 (b) All softmax attention models generalize in length and the ICL loss decreases as $L$ increases.

Moreover, we plot the ICL losses of the trained attention models and the classical statistical estimators in Figure 3a. This figure demonstrates that multi-head attention models closely track the performance of both the vanilla GD predictor and its debiased variant (see §4.3). This suggests

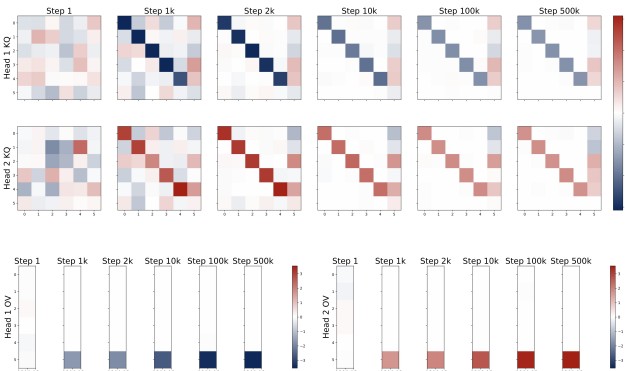

*Figure 4.* Heatmaps of the KQ matrices and OV vectors along training epochs with $H = 2$, $d = 5$, and $L = 40$, trained over $10^6$ steps with random initialization. KQ matrices (top) and OV vectors (bottom) form the diagonal and the last-entry-only patterns respectively during the early stages of training and then optimize within the simplified regime described in (3.1).

that multi-head attention model approximately implements a version of GD algorithm.

> **Observation 4.** The training of KQ and OV circuits across different heads follow similar trajectories under random initializations. In particular, the patterns found by Observations 1 and 2 appear early in the optimization trajectory, and are preserved during training. Moreover, the magnitude of $\omega^{(h)}$ in positive or negative heads first increases and then decreases to converge, while the $\sum_{h \in \mathcal{H}_+} \mu^{(h)}$ monotonically increase until convergence.

Despite random initializations, the parameter evolution follows a highly consistent trajectory throughout training. As shown in Figure 4, the attention model quickly develops the pattern in (3.1), identified in Observation 1, during the early stages of training. The attention model continues to optimize the loss function with this pattern preserved throughout the training process. Moreover, by looking at the dynamics of $(\omega, \mu)$, we observe that the two properties identified by Observation 2 are also preserved during the training. Furthermore, the training dynamics of the KQ circuits are not monotonic. The magnitudes of $\omega^{(h)}$'s first increase and then decrease to stabilize. Whereas the magnitude of OV circuits, i.e., $\sum_{h \in \mathcal{H}_+} \mu^{(h)}$, increases steadily.

# 4. Mechanistic Interpretation

## 4.1. Reparametrization of Attention Model

As shown in Observation 4 in §3, the attention model develops a diagonal-only pattern in KQ circuits and a last-entry only pattern in OV circuits during training. This structure emerges early in the training and then continues optimizing within this regime (see Figure 4). Thus, it is sufficient to

consider the parameterization in (3.1) to interpret the core aspects of model training. With this reparameterization, the evolution of the model model can be analyzed by tracking $(\omega, \mu)$, and the transformer predictor follows (3.2). Motivated by the analysis in (Chen et al., 2024a), we present a refined argument for approximating the population loss in (2.3) under the reparameterization in (3.1). As we will see later, gradient flow based on this approximate loss reveals the empirical observations from a theoretical perspective.

**Proposition 4.1** (Informal). *Consider an $H$-head attention model parameterized by* (3.1) *with dimension $d \in \mathbb{Z}^+$ and sample size $L \in \mathbb{Z}^+$. Suppose that $d > \log L$ and parameters $(\omega, \mu) \subseteq \mathbb{R}^{2H}$ satisfies that $\|\omega\|_\infty \lesssim \sqrt{\log L/d}$ and $\|\mu\|_\infty \lesssim L^{1/5}$, then it holds that*

$$\mathcal{L}(\omega, \mu) = 1 + \sigma^2 - 2\mu^\top \omega + \mu^\top \big(\omega\omega^\top + (1 + \sigma^2) \cdot L^{-1}$$
$$\cdot \exp(d\omega\omega^\top)\big)\mu + O(dH^2 \cdot L^{-1/5}).$$

The formal statement and proof of Proposition 4.1 are deferred to §E.2, where the result extends beyond the case $d > \log L$ and allows a more flexible trade-off between the scaling and the resulting approximation error. Proposition 4.1 establishes that, under certain scaling assumptions, the true loss can be well approximated by a simplified formulation in terms of $(\omega, \mu)$ with the approximation error vanishing as the ICL sample size $L$ increases. In addition, we provide experimental validations for Proposition 4.1. As shown in Figure 5, the approximation can effectively capture the true loss landscape (see Figure 5b,) and performs particularly well when $\omega$ or $\mu$ is small (see Figure 5a). See §C.4 for more detailed explanations and additional experimental validations of Proposition 4.1. Hence, to analyze the parameter evolution of the transformer, it suffices to consider the gradient dynamics of the approximate loss, which is given by

$$\widetilde{\mathcal{L}}(\omega, \mu) = \sigma^2 + (1 - \mu^\top \omega)^2$$
$$+ \mu^\top (1 + \sigma^2) \cdot L^{-1} \cdot \exp(d\omega\omega^\top)\mu. \quad (4.1)$$

## 4.2. Training Dynamics, Emerged Patterns and Solution Manifold

In this section, we provide a detailed analysis of the evolution of attention weights during the training process, with a focus on multi-head attention models. Starting from random initializations, we show that the parameters evolve to form a pattern consistently similar. To interpret such a phenomenon, we provide a theoretical explanation and follow the training dynamics based on the approximated loss in (4.1). Our analysis focuses on the training process using gradient descent (GD). For a learning rate $\eta > 0$, the parameter update at step $t \in \mathbb{N}$ is given by

$$\mu_{t+1} \leftarrow \mu_t - \eta \cdot \nabla_\mu \widetilde{\mathcal{L}}(\mu_t, \omega_t), \ \omega_{t+1} \leftarrow \omega_t - \eta \cdot \nabla_\omega \widetilde{\mathcal{L}}(\mu_t, \omega_t).$$

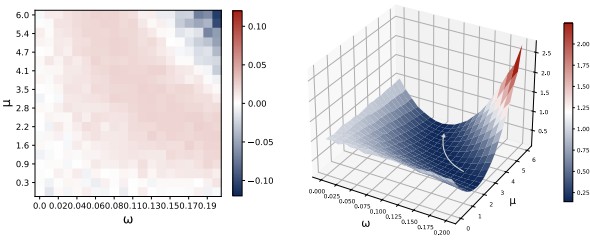

(a) Loss Difference $\mathcal{L} - \widetilde{\mathcal{L}}$. (b) Landscape of True Loss.

*Figure 5.* Experimental validation for Proposition 4.1 with 2-head attention model under $(\omega, -\omega)$ and $(\mu, -\mu)$ with $L = 40$, $d = 5$ and $\sigma^2 = 0.1$. Panel (a) plots the heatmap of difference between the actual and approximated loss $\mathcal{L}(\mu, \omega) - \widetilde{\mathcal{L}}(\mu, \omega)$. Panel (b) illustrate the actual and approximated loss landscape, respectively. The gray arrow highlights the "river valley" loss landscape where the loss decreases gradually in that direction towards minimum.

Following this, the training dynamic undergoes two stages:

- In **Stage I**, two patterns emerged—*sign matching* and *zero-sum OV*, which optimizes the low-order terms mainly due to the small initialization and are maintained along training (see (4.2)). In addition, optimality equality in (4.3) holds approximately, which facilitates subsequent evolution.

- In **Stage II**, the $\omega^{(h)}$'s converge to *homogeneous KQ scaling* in (4.4) to optimize the remaining high-order terms. Based on the signs, these heads group into *positive*, *negative* and *dummy heads*. Finally, $\mu^{(h)}$'s fall into the solution manifold in (4.5) defined by scale of $\omega^{(h)}$.

**Stage I: Establish the Sign-matching and Zero-sum OV Pattern.** Consider starting from a small initialization and assume that $\|\omega_0\|_\infty \leq 1/\sqrt{d}$ and $\|\mu_0\|_\infty \leq c\sqrt{d}$ hold for all $t \in \mathbb{N}$ with constant $c \in (0, 1)$. As shown in Figure 1b, at the very beginning of the training process, the model quickly develops the *sign matching* and *zero-sum OV* pattern, summarized below: for all $h \in [H]$, it holds that

$$\langle \mu, \mathbf{1}_H \rangle = 0, \quad \text{and} \quad \text{sign}(\omega^{(h)}) = \text{sign}(\mu^{(h)}). \quad (4.2)$$

To understand the mechanisms, we examine the gradient update. By applying the Taylor expansion over $\exp(d\omega\omega^\top)$ at $\omega = \mathbf{0}_H$, the gradient update of $\mu$ at step $t$ followz

$$\mu_{t+1} \leftarrow \mu_t + 2\eta \cdot \underbrace{\left(1 - \left(1 + (1 + \sigma^2)dL^{-1}\right)\langle \mu_t, \omega_t \rangle\right)\omega_t}_{\text{Sign-Matching Term}}$$

$$-\frac{2\eta(1 + \sigma^2)}{L} \cdot \Big( \underbrace{\langle \mu_t, \mathbf{1}_H \rangle \mathbf{1}_H}_{\text{Zero-sum OV Term}} + \underbrace{\sum_{k=2}^\infty \frac{d^k}{k!}\langle \mu_t, \omega_t^{\odot k}\rangle \omega_t^{\odot k}}_{\text{High-Order Terms}} \Big),$$

and the updating scheme of $\omega$ shares a similar form as

$$\omega_{t+1} \leftarrow \omega_t + 2\eta \cdot \underbrace{\left(1 - \left(1 + (1 + \sigma^2)dL^{-1}\right)\langle \mu_t, \omega_t \rangle\right)\mu_t}_{\text{Sign-Matching Term}}$$

$$-\underbrace{\frac{2\eta \cdot (1 + \sigma^2)}{L}\sum_{k=2}^\infty \frac{d^k}{(k-1)!}\cdot \langle \mu_t, \omega_t^{\odot k}\rangle \cdot \mu_t \odot \omega_t^{\odot k-1}}_{\text{High-Order Terms}}.$$

In the early stage of training, the higher-order terms are negligible due to small initializations, and the sign-matching terms and zero-sum OV terms dominate. For the zero-sum OV term, it is straightforward to see that it encourages $\langle \mu_t, \mathbf{1}_H \rangle = 0$. For sign matching terms, $\langle \omega_t, \mu_t \rangle$ remains small in the initial stage such that $(1 + (1 + \sigma^2) \cdot dL^{-1}) \cdot \langle \mu_t, \omega_t \rangle < 1$. Thus, $\mu_t$ and $\omega_t$ are updated in each other's direction, gradually aligning to share the same sign. Also, the sign-matching terms contribute to the gradient update until the following optimality equality is established:

$$\left(1 + (1 + \sigma^2) \cdot dL^{-1}\right) \cdot \langle \mu_t, \omega_t \rangle = 1. \quad (4.3)$$

In this stage, we ignore the contribution of the high-order terms to the gradient as both $\mu_t$ and $\omega_t$ have small magnitude. After (4.2) and (4.3) are establised, the training enters the second stage where the high-order terms dominate.

**Stage II: Emergence of Homogeneous KQ Scaling.** As shown in Figure 1b, after some training steps, these heads are grouped into positive, negative, and dummy heads. For dummy heads, all parameters degenerate to zero, indicating that these heads do not contribute to the final predictor and output 0. For positive and negative heads, their KQ parameters share the same scaling with different signs:

$$\omega^{(h)} = \gamma \cdot \text{sign}(\omega^{(h)}) \text{ with } \gamma > 0, \ \forall h \in [H]. \quad (4.4)$$

To understand how the homogeneous scaling develops, which significantly simplifies the predictor given by the multi-head attention (see §4.3), we revisit the population loss and focus on the regime where (4.2) and (4.3) are well-established after Stage I. Note that

$$\widetilde{\mathcal{L}}(\omega_t, \mu_t) = \sigma^2 + (1 + \sigma^2) \cdot L^{-1} \cdot \langle \mu_t, \mathbf{1}_H \rangle^2$$

$$+ (1 - \langle \mu_t, \omega_t \rangle)^2 + (1 + \sigma^2) \cdot L^{-1} \cdot \sum_{k=1}^\infty \frac{d^k}{k!} \cdot \langle \mu_t, \omega_t^{\odot k}\rangle^2,$$

where the first two terms are constants given the zero-sum $\mu$ and transitional optimality. Hence, the remaining high-order terms dominates the dynamic, which satisfies

$$\sum_{k \in \{2q:q \in \mathbb{Z}^+\}} \frac{d^k}{k!}\langle \mu_t, |\omega_t|^{\odot k}\rangle^2 + \sum_{k \in \{2q-1:q \in \mathbb{Z}^+\}} \frac{d^k}{k!}\langle |\mu_t|, |\omega_t|^{\odot k}\rangle^2$$

$$\geq \sum_{k \in \{2q-1:q \in \mathbb{Z}^+\}} \frac{d^k}{k!} \cdot \|\mu_t\|_1^{2(1-k)} \cdot \langle |\mu_t|, |\omega_t| \rangle^{2k},$$

where the first equality follows $\text{sign}(\omega^{(h)}) = \text{sign}(\mu^{(h)})$ and the last one uses non-negativity of even-power terms and Hölder's inequality for odd-power terms. Given zero-mean $\mu_t$, the lower bound is attained when (i) the even-power terms reach the 0-minima by taking $|\omega_t| = \gamma \cdot \mathbf{1}_H$ since $\langle \mu_t, |\omega_t|^{\odot k} \rangle = \gamma^k \cdot \langle \mu_t, \mathbf{1}_H \rangle = 0$ for all $k \in \{2q : q \in \mathbb{Z}^+\}$, (ii) the odd-power terms attain the lower bound if and only if $|\mu_t| \odot |\omega_t|^{\odot k} \propto |\mu_t|$ for all $k \in \{2q - 1 : q \in \mathbb{Z}^+\}$ such that $|\omega_t| \propto \mathbf{1}_H$, leading to the homogeneous pattern.

**Solution Manifold.** Suppose patterns in (4.2) and (4.4) fully emerge. At this convergence stage, for arbitrary scale $\gamma$, with a bit of abuse of notation, the approximated loss can be rewritten as:

$$\widetilde{\mathcal{L}}(\gamma, \mu) = \sigma^2 + (1 - \gamma \cdot \|\mu\|_1)^2$$
$$+ (1 + \sigma^2) \cdot L^{-1} \cdot \sinh(d\gamma^2) \cdot \|\mu\|_1,$$

where we use the Taylor series of hyperbolic sine functions. Following this, the loss is quadratic with respect to $\|\mu_t\|_1$, thus for any fixed scale $\gamma > 0$, the minimizer satisfies that

$$\|\mu\|_1 = \left( \gamma^2 + (1 + \sigma^2) \cdot L^{-1} \cdot \sinh(d\gamma^2) \right)^{-1} \cdot \gamma := 2\mu_\gamma.$$

Here we define $\mu_\gamma$ as one half of the optimal value of $\|\mu\|_1$. Hence, we know that the limiting $\omega$ and $\mu$ are characterized by a solution manifold, given by $\mathscr{S}^* = \{\mathscr{S}_\gamma\}_{\gamma > 0}$, where

$$\mathscr{S}_\gamma = \Big\{ (\omega, \mu) \subseteq \mathbb{R}^H : \omega^{(h)} = \gamma \cdot \text{sign}(\mu^{(h)}),$$
$$\sum_{h \in \mathcal{H}_+} \mu^{(h)} = - \sum_{h \in \mathcal{H}_-} \mu^{(h)} = \mu_\gamma \Big\}. \quad (4.5)$$

We remark that the learned scale $\gamma$ depends on the learning rate, training steps, and batch size (if SGD or Adam is applied) due to the "river valley" loss landscape near the global minimum (see Figure 5b). This phenomenon can be attributed to *edge of stability* (Cohen et al., 2021), a behavior commonly observed in neural network training. In this regime, gradient descent progresses non-monotonically, oscillating between the "valley walls" of the loss surface and failing to fully converge to the minimum.

Although the learned $\gamma$ can vary slightly, $\mu$ always lies within its corresponding solution manifold $\mathscr{S}_\gamma$. The optimal value of $\gamma$ and the resulting transformer-based ICL predictor are derived later in §4.3. Furthermore, to explain the evolution of the parameters, we provide a more detailed analysis of the gradient flow of training a two-head attention based on the approximate loss in §D under a well-specified initialization, where the full gradient flow dynamics can be characterized explicitly.

## 4.3. Expression, Approximation and Optimality

In this section, we examine the statistical properties of the learned attention models. Our experiments show that, regardless of the number of heads, attention models consistently learn the diagonal-only and last-entry-only patterns in (3.1) while using different predictors. For multi-head attention with $H \geq 2$, heads are grouped into positive, negative and dummy heads as discussed in §4.2. Positive and negative heads are coupled to approximate *gradient descent* (GD), consistent with the findings from linear transformers (Mahankali et al., 2023; Ahn et al., 2023a; Zhang et al., 2024), but with an additional *debiasing term*. As we will show below, this is a result of the fact that the limiting values of attention parameters fall in the solution manifold $\mathscr{S}_\gamma$ with a small $\gamma$. Moreover, when $L$ is much larger than $d$, debiased GD coincides with vanilla GD, which means that multi-head attention learns the same algorithm as the linear attention in this regime.

**Multi-head Attention Predictor.** Consider an arbitrary scale $\gamma > 0$, then for any parameters on the solution manifold $(\omega, \mu) \subseteq \mathscr{S}_\gamma$, the attention predictor takes the form:

$$\widehat{y}_q = \sum_{h \in \mathcal{H}_+} \mu^{(h)} \cdot \sum_{\ell=1}^L \frac{y_\ell \cdot \exp(\gamma \cdot x_\ell^\top x_q)}{\sum_{\ell=1}^L \exp(\gamma \cdot x_\ell^\top x_q)}$$
$$+ \sum_{h \in \mathcal{H}_-} \mu^{(h)} \cdot \sum_{\ell=1}^L \frac{y_\ell \cdot \exp(-\gamma \cdot x_\ell^\top x_q)}{\sum_{\ell=1}^L \exp(-\gamma \cdot x_\ell^\top x_q)}.$$

Then, for any $H \geq 2$, the multi-head attention model implements an *equivalent predictor as 2-head model* with $\omega_{\text{eff}} = (\gamma, -\gamma)$ and $\mu_{\text{eff}} = (\sum_{h \in \mathcal{H}_+} \mu^{(h)}, \sum_{\mathcal{H}_-} \mu^{(h)})$ with $\sum_{h \in \mathcal{H}_+} \mu^{(h)} = -\sum_{\mathcal{H}_-} \mu^{(h)} = \mu_\gamma$. For small scale $\gamma > 0$, by linearizing $\exp(\cdot)$ around 0, we have

$$\widehat{y}_q \approx \frac{2\mu_\gamma \gamma}{L} \cdot \sum_{\ell=1}^L y_\ell \cdot \bar{x}_\ell^\top x_q \implies \widehat{y}_q^{\text{gd}}(\eta) = \frac{\eta}{L} \cdot \sum_{\ell=1}^L y_\ell \cdot \bar{x}_\ell^\top x_q,$$

where we denote $\bar{x}_\ell = x_\ell - \frac{1}{L} \sum_{\ell=1}^L x_\ell$ and the approximation also uses the first-order approximation of reciprocal (see Remark F.1). Thus, multi-head attention emulates the one-step *pre-conditioned gradient descent*[1] to optimize the empirical loss $\frac{1}{2L} \sum_{\ell=1}^L (\beta^\top \bar{x}_\ell - y_\ell)^2$ with initialization $\beta_0 = 0$, learning rate $2\mu_\gamma \gamma$ and the *debiased* covariates $\bar{x}_\ell$'s. In experiments, the learned $\gamma$ is very small ($\sim 0.1$), which ensures the validity of the approximations. This observation suggests that, despite the nonlinear nature of softmax, the model can still capture the inner linearity. To achieve

---

[1]Under the isotropic setup, the pre-conditioned GD coincides with the standard GD due to the identity covariance matrix. Here, we refer to it as pre-conditioned GD to align with experimental observations in the anisotropic case (see §B.3), where clear evidence indicates that it utilizes a pre-conditioner.

this, the softmax attentions work in the *linear regime* by taking small-scale $\gamma$ and canceling out the constants via *coupled positive-negative heads*. As shown in Figure 3b, the learned predictor performs nearly identically to the debiased GD predictor, which also matches the standard GD predictor when $L$ is large since $\frac{1}{L}\sum_{\ell=1}^{L} x_\ell \approx \mathbb{E}[x_\ell] = \mathbf{0}_d$ and $\frac{1}{L}\sum_{\ell=1}^{L} y_\ell \approx \mathbb{E}[\beta^\top x_\ell + \epsilon_\ell] = 0$.

**Single-Head Attention Predictor.** Different from multi-head models, single-head attention has less expressivity, and the learned predictor takes the form of

$$\widehat{y}_q = \mu \cdot \sum_{\ell=1}^{L} \frac{y_\ell \cdot \exp(\omega \cdot x_\ell^\top x_q)}{\sum_{\ell=1}^{L} \exp(\omega \cdot x_\ell^\top x_q)}.$$

By minimizing the approximate loss, it is shown in (Chen et al., 2024a) that the minimizer is given by $\omega^* = 1/\sqrt{d}$ and $\mu^* = \sqrt{d} \cdot (1 + e \cdot (1 + \sigma^2) \cdot dL^{-1})^{-1}$. We remark that the single-head attention precisely implements a Nadaraya-Watson estimator with a Gaussian RBF kernel, when the covariates are sampled from a sphere, i.e., $\text{supp}(\mathsf{P}_x) = \mathbb{S}^{d-1}$. Moreover, the optimal parameter $\omega^*$ corresponds to a bandwidth with value $d^{1/4}$. Hence, we see why having one additional attention head significantly improves the statistical rate in in-context linear regression as shown in Figure 3— one-head attention corresponds to a nonparametric predictor while a multi-head attention yields a parametric predictor.

In the following, we provide a formal statement of the above argument and establish the optimality of the learned predictor in a high-dimensional asymptotic regime.

**Theorem 4.2.** *Consider a larger parameter space $\bar{\mathscr{S}} \supseteq \mathscr{S}^*$, defined as*

$$\bar{\mathscr{S}} = \{(\omega, \mu) : \min\{|\mathcal{H}_+|, |\mathcal{H}_-|\} > 1,$$
$$\forall \gamma > 0, \ \omega^{(h)} = \gamma \cdot \text{sign}(\omega^{(h)}) \ \forall h \in [H]\} \quad (4.6)$$

*and denote $\sum_{h \in \mathcal{H}_+} \mu^{(h)} = \mu_+$ and $\sum_{h \in \mathcal{H}_-} \mu^{(h)} = \mu_-$.*

*(i)* **(Approximation)** *Let $\eta > 0$ be a constant learning rate and let $\delta \in (0, 1)$ be a given failure probability. For any scaling $\gamma > 0$, we define $\breve{\mu} = \eta/(2\gamma)$. Consider a multi-head attention with no dummy head and $\theta \in \bar{\mathscr{S}}$. Moreover, we set $\mu_+ = -\mu_- = \breve{\mu}$. Then, when $L \gtrsim \log(1/\delta)$ and $\gamma \lesssim (\sqrt{d} \cdot \log(L/\delta))^{-1}$, with probability at least $1 - \delta$, we have*

$$|\widehat{y}_q(\theta) - \widehat{y}_q^{\mathsf{gd}}(\eta)| \leq \widetilde{O}\big(\sqrt{1 + \sigma^2} \cdot \gamma \cdot d\big),$$

*where $\widetilde{O}(\cdot)$ omits logarithmic factors. In particular, suppose we drive $\gamma$ to zero while keeping $\breve{\mu} = \eta/(2\gamma)$, the resulting $\widehat{y}_q(\theta)$ coincides with $\widehat{y}_q^{\mathsf{gd}}(\eta)$.*

*(ii)* **(Optimality)** *Consider minimizing $\widetilde{\mathcal{L}}(\omega, \mu)$ over $\bar{\mathscr{S}}$. The minimum is attained in $\mathscr{S}^*$. In particular, the minimizer can be obtained by taking $\gamma \to 0^+$ in (4.5). As*

a result, the optimal attention model that minimizes $\widetilde{\mathcal{L}}(\omega, \mu)$ coincides with the debiased GD estimator $\widehat{y}_q^{\mathsf{gd}}(\eta^*)$, where we define $\eta^* = (1 + (1 + \sigma^2) \cdot d/L)^{-1}$.

*(iii)* **(Bayes Risk)** *Consider the high-dimensional regime where $L \to \infty$ and $d/L \to \xi$, where $\xi \in (0, \infty)$ is a constant. Suppose the noise level $\sigma^2 > 0$ and $\xi$ is sufficiently small such that $\sigma^2 + \xi^{-1} > 1$. Then*

$$\frac{\mathcal{E}(\widehat{y}_q^{\mathsf{gd}}(\eta^*))}{\mathsf{BayesRisk}_{\xi,\sigma^2}} \leq 1 + \sigma^{-2} \cdot \big\{(1 + \xi\sigma^2) \cdot (1 + \sigma^2 + \xi^{-1})\big\}^{-1},$$

*where $\mathsf{BayesRisk}_{\xi,\sigma^2}$ denotes the limiting Bayes risk.*

The proof of Theorem 4.2 is deferred to §F. In part (i), we show that softmax transformer can represent the linear pre-conditioned GD predictor with small scale $\gamma$. Part (ii) analyzes the approximated global minimizer corresponds to the debiased GD predictor. Finally, in part (iii), we compare the learned predictor with the Bayes risk and show that it is nearly optimal in the proportional regime, particularly when observations are noisy or the dimensionality of the covariates is high.

**Discussions on Linear Transformer.** Building on our analysis, we can show that by working in the linear regime, linear and softmax attention can achieve comparable expressive power. However, linear transformers rely on a fixed normalization factor, limiting their ability to generalize across different sequence lengths, whereas softmax transformers dynamically adjust for flexible length generalization (see §B.1 for detailed discussions).

**Extentions.** We extend our analysis to anisotropic and multi-task regression settings. In the anisotropic case, trained multi-head attention continues to learn a preconditioned gradient descent predictor. For multi-task learning, the number of attention heads determines distinct learning strategies, ranging from weighted kernel regression to independent preconditioned gradient descent predictors for each task, revealing an intriguing superposition phenomenon (see §B.3, §C.2 and C.3 for details).

# 5. Conclusion

Our study provides a comprehensive analysis of multi-head softmax transformers for in-context learning in linear regression, uncovering key structural patterns that emerge during training. We characterize the model's development and demonstrate its implementation of preconditioned gradient descent, shedding light on its underlying optimization dynamics. These findings enhance our understanding of transformer-based ICL, highlighting its advantages over single-head and linear attention models and offering valuable insights into its broader applicability.

## Impact Statement

This paper presents work whose goal is to advance the field of Machine Learning. There are many potential societal consequences of our work, none which we feel must be specifically highlighted here

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

# A. Additional Background and Related Works

**Notation.** For some $n \in \mathbb{N}^+$, let $[n] = \{i \in \mathbb{Z} : 1 \leq i \leq n\}$. For vector $v \in \mathbb{R}^d$, denote by $\odot$ the element-wise Hadamard product and $v^{\odot k} = (v_1^k, \ldots, v_d^k)$. Let $\|\cdot\|_p$ denote the $\ell_p$-norm and softmax operator is defined as $\mathtt{smax}(v) = (\mathtt{smax}(v)_i)_{i \in [d]}$ where $\mathtt{smax}(v)_i = \exp(v_i)/\sum_j \exp(v_j)$. Let $\mathrm{sign}(\cdot)$ denote the sign function that returns 1, $-1$ or 0 based on the sign of input $x \in \mathbb{R}$. Let $\mathbf{1}_d$ and $\mathbf{0}_d$ denote the all-one and all-zero vector of size $d$ and denote $I_d$ as the $d$-by-$d$ identity matrix. For vector $\nu \in \mathbb{R}^d$ and indices set $\mathcal{I} \subseteq [d]$, we define $\nu_{\mathcal{I}} = (\nu_i)_{i \in \mathcal{I}} \in \mathbb{R}^{|\mathcal{I}|}$. For two functions $f(x) \geq 0$ and $g(x) \geq 0$ defined on $x \in \mathbb{R}^+$, we write $f(x) \lesssim g(x)$ or $f(x)$ as $O(g(x))$ if there exists two constants $c > 0$ such that $f(x) \leq c \cdot g(x)$, we write $f(x) \asymp g(x)$ or $f(x) = \Theta(g(x))$ if $f(x) \lesssim g(x)$ and $g(x) \lesssim f(x)$.

## A.1. Related Works

**In-Context Learning (ICL).** LLMs exhibit strong reasoning abilities, with their ICL capability playing a crucial role in their performance. Unlike fine-tuned models customized for specific tasks, LLMs demonstrate comparable capabilities by learning from informative prompts (Liu et al., 2021; Min et al., 2021; Nie et al., 2022). The theoretical understanding of ICL remains an active area of research. One line of research interprets ICL as a form of Bayesian inference embedded within transformer architectures (Xie et al., 2021; Zhang et al., 2023; Jeon et al., 2024; Hu et al., 2024). Another line of work focuses on understanding how transformers internally emulate specific algorithms to address ICL tasks (Garg et al., 2022; Nichani et al., 2024; Chen et al., 2024a; Fu et al., 2024; Sheen et al., 2024; Li et al., 2024b). Among these works, some focus on ICL for classification problems or learning with a finite dictionary (Sheen et al., 2024; Huang et al., 2023; Yang et al., 2024; Nichani et al., 2024), while another line of work examines how the attention model performs in-context linear regression (Von Oswald et al., 2023; Zhang et al., 2024; Chen et al., 2024a;c; Zhang et al., 2025). Meanwhile, multiple works demonstrate the remarkable ability of the model in feature learning (Kim & Suzuki, 2024; Yang et al., 2024; Huang et al., 2023) and decision-making (Sinii et al., 2023; Lin et al., 2023; He et al., 2024). Beyond these works, many researchers seek to uncover the internal ICL procedure of transformers, typically in more complex algorithms (Fu et al., 2023; Giannou et al., 2024; Cheng et al., 2023; Lin & Lee, 2024). Other works investigate the training process with multiple tasks (Kim et al., 2024; Tripuraneni et al., 2021), which are broadly related.

**Comparison with Related Work.** We provide a detailed discussion of the differences between our work and that of Chen et al. (2024a) and Cui et al. (2024), which are among the most closely related studies.

Chen et al. (2024a) considers multi-head attention in the context of multi-task linear regression, where the number of heads matches the number of tasks. Under a specialized initialization, they show that each head independently learns to solve a distinct task—effectively reducing to a single-head per task setup, which corresponds to the single-head case in our framework. In contrast, our setup allows multiple heads to be flexibly allocated to a single task, enabling a more expressive and complex model architecture. As a result, while Chen et al. (2024a) shows that the single-head model learns a nonparametric, kernel-type predictor with scaling of KQ parameters as $1/\sqrt{d}$, we demonstrate that the multi-head model instead learns a parametric gradient descent predictor with KQ converging to $0^+$. This not only recovers the known results for linear attention (Zhang et al., 2024) but also reveals that multi-head softmax attention can outperform the single-head one by effectively encoding the linear architecture through an explicit approximation.

Cui et al. (2024) also identifies the diagonal KQ patterns with potentially positive and negative values in two-head softmax attention, and observe identical performance when the number of heads exceeds two. We go further by quantitatively characterizing the learned model. Specifically, we reveal detailed sign-matching, homogeneous KQ magnitudes, and zero-sum OV patterns for head counts beyond $H = 2$, and show that multi-head softmax attention learns to implement a gradient descent predictor. From a theoretical perspective, Cui et al. (2024) adopts full-model parameterization and conducts a loss landscape analysis. In contrast, we begin by establishing an approximate loss and then develop a comprehensive explanation based on training dynamics, function approximation, and optimality analysis. Our results go beyond the core argument in Cui et al. (2024) regarding the superiority of multi-head over single-head attention: we not only compare the testing loss, but also explicitly demonstrate that single-head attention learns a nonparametric kernel regressor, while multi-head attention learns a more powerful parametric gradient descent predictor.

## A.2. Interpretation of Transformer Architecture

**Interpretation of Attention Model.** Note that (2.2) is a standard multi-head attention layer with a residual link. This function can be regarded as a mapping from a sequence of $L + 1$ vectors in $\mathbb{R}^{d+1}$, i.e., $\{z_\ell\}_{\ell \in [L]} \cup \{z_q\}$, to a new sequence

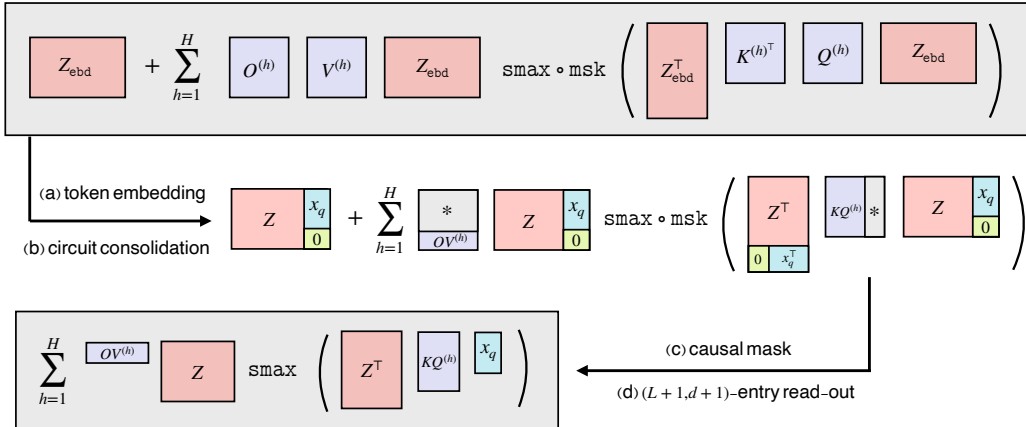

*Figure 6.* Illustration of the derivation from the full multi-head attention architecture in (2.2) to the simplified model. In the graph, we show the specified token embedding and read-out function with consolidated KQ and OV circuits. We use $*$ to denote the ineffective parameters due to the read-out function or the 0 embedding for the query token.

of $L + 1$ vectors in $\mathbb{R}^{d+1}$. In particular, at each token position $j$, the query, key, and value in the $h$-th head are given by $Q^{(h)} z_j$, $K^{(h)} z_j$, and $V^{(h)} z_j$, respectively. To get the output at position $j$, we compute the similarity between the query $Q^{(h)} z_j$ and all the previous keys to get $\{\langle K^{(h)} z_i, Q^{(h)} z_j \rangle\}_{i<j}$. These similarity scores are passed through the softmax function $\texttt{smax}(\cdot)$ to form a probability distribution over all previous token positions $[j-1]$. This distribution weights the values $\{V^{(h)} z_i\}_{i<j}$ to compute the attention output. The outputs of all $H$ heads are aggregated by the output matrices $\{O^{(h)}\}_{h \in [H]}$. Combining with the input $Z_{\text{ebd}}$ from the residual link, we get the output in (2.2).

Note that when computing the output at each token position $j$, we only use the key $Q^{(h)} z_j$ to look at the queries and values before position $j$. Thus, the softmax output dimension varies with $j$, forming a probability distribution over $[j-1]$ for each $j \in [L+1]$. In the model (2.2), this is enforced by the causal mask $\texttt{msk}(\cdot)$ before the softmax function. With the causal mask, only the first $j-1$ entries in the output vector of the softmax function are nonzero. Compared to the standard decoder-only transformer model (e.g., Vaswani, 2017), we omit layer normalization and positional embeddings. Layer normalization is unnecessary because a single-layer architecture does not suffer from the issue of vanishing or exploding gradients. In addition, the regression problem is permutation invariant in $\{(x_\ell, y_\ell)\}_{\ell \in [L]}$, making positional information unnecessary.

## B. Discussions and Extensions

The previous sections demonstrated that when training a multi-head attention model on linear ICL data, the learned weight matrices exhibit significant patterns. These structured weights enable the model to implement a debiased GD estimator. As we theoretically established, these patterns emerge from the interaction between the transformer architecture and the underlying data distribution. In this section, we further investigate how different components contribute to this phenomenon. Specifically, we examine the role of the softmax function in the attention model and the impact of isotropic covariate distribution. Additionally, we extend to the multi-task in-context regression setting and study how the interplay between the number of tasks and the number of heads affects model behavior in our empirical study.

### B.1. Comparison: Linear vs. Softmax Transformer

In this section, we study the connection between linear and softmax transformers. Proposed by Von Oswald et al. (2023), linear transformer simplifies the architecture by removing the nonlinear activation and the causal mask, given by

$$\texttt{LinTF}_\theta(Z_{\text{ebd}}) = Z_{\text{ebd}} + \frac{1}{L} \sum_{h=1}^{H} O^{(h)} V^{(h)} Z_{\text{ebd}} \cdot Z_{\text{ebd}}^\top K^{(h)\top} Q^{(h)} Z_{\text{ebd}}. \tag{B.1}$$

Recent research has focused on understanding the inner mechanisms of transformers through the simplified linear trans-

former(e.g., Von Oswald et al., 2023). For the linear ICL task, Zhang et al. (2024) show that linear attention can be trained to implement the one-step GD estimator with just one head. This raises the following question:

*In linear ICL tasks, does softmax attention offer advantages over linear attention?*

We argue that the softmax attention models are more advantageous than linear attention due to their enhanced expressive power. In fact, any $H$-head linear attention can be approximately implemented by a multi-head softmax attention with $2H$ heads, when the token embeddings are centralized. Specifically, consider a $2H$-head softmax attention model parameterized as

$$\mathtt{TF}_\theta(Z_{\mathsf{ebd}}) = Z_{\mathsf{ebd}} + \sum_{h=1}^{H} \sum_{j\in\{0,1\}} \frac{(-1)^j}{2\gamma} \cdot O^{(h)} V^{(h)} Z_{\mathsf{ebd}} \cdot \mathtt{smax}\big((-1)^j \gamma \cdot Z_{\mathsf{ebd}}^\top K^{(h)^\top} Q^{(h)} Z_{\mathsf{ebd}}\big),$$

where we use the same parameters as in $\mathtt{LinTF}_\theta(\cdot)$ and set $\gamma$ to a small rescaling constant. We define $\bar{Z}_{\mathsf{ebd}} = L^{-1} \cdot Z_{\mathsf{ebd}} \mathbf{1}_L$, which averages the token embedding across the $L$ token positions. Using a similar approximation when $\gamma$ is close to zero, we have

$$\mathtt{TF}_\theta(Z_{\mathsf{ebd}}) \approx \mathtt{LinTF}_\theta(Z_{\mathsf{ebd}}) - \sum_{h=1}^{H} O^{(h)} V^{(h)} \bar{Z}_{\mathsf{ebd}} \cdot \bar{Z}_{\mathsf{ebd}}^\top K^{(h)^\top} Q^{(h)} Z_{\mathsf{ebd}}. \tag{B.2}$$

From (B.2), the output of softmax attention closely matches its linear counterpart when the second term is small, e.g., token embeddings are centralized such that $\bar{Z}_{\mathsf{ebd}} \approx \mathbf{0}_d$, which is the case in in-context linear regression.

**Length Generalization.** While softmax attentions use more heads, they benefit from dynamic normalization. Notice that the linear attention in (B.1) requires a fixed normalization factor $1/L$ which depends on the sequence length $L$. In contrast, the parameters of softmax attention do not involve $L$ explicitly, but it effectively produces a $1/L$ normalization factor when $\gamma$ is sufficiently small, as shown in (B.2). The property of dynamic normalization enables softmax attention trained on linear ICL data to naturally generalize to different sequence lengths. During testing, a trained softmax model can process inputs with many more demonstration examples than it saw during training, and the ICL prediction error decreases as more examples are provided. In contrast, a trained linear attention model cannot generalize to longer sequences easily because its weights explicitly contain the normalization factor $1/L$. When the sequence length changes, linear attention would require different model parameters.

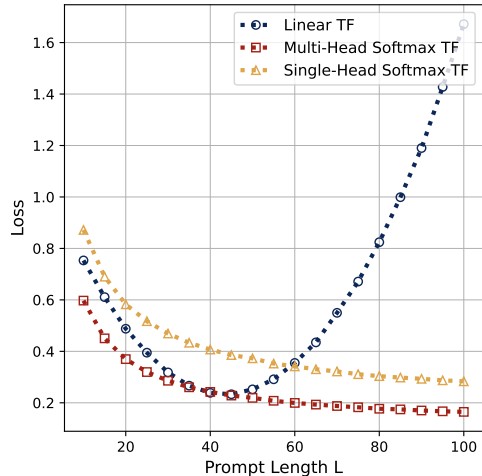

*Figure 7.* ICL prediction errors of linear attention, single-head and multi-head softmax attention trained on $L = 40$ but evaluated over different lengths. Note that softmax attention can generalize in length because the prediction error decays with more examples. In contrast, linear attention cannot generalize in length and the prediction error starts to increase when the number of examples exceeds $L$.

We conduct an experiment to test the length generalization ability of both model types. We train these models with $L = 40$ demonstration examples and test them with $L'$ examples, where $L'$ ranges from 10 to 100. The ICL prediction errors in Figure 7 show that both single- and multi-head softmax attention models generalize effectively to longer inputs, while the linear attention fails to do so. In particular, the prediction errors further decrease when the number of demonstration examples $L'$ increases beyond $L = 40$. In contrast, the prediction error of linear attention starts to increase after $L' > 40$.

## B.2. Ablation Study: Alternative Activations Beyond Softmax

Building on our comparison between linear and softmax transformers, we now further examine the role played by the softmax activation function itself. Consider a normalized activation $\sigma : \mathbb{R}^d \mapsto \mathbb{R}^d$ defined by letting $\sigma(\nu)_i = f(\nu_i)/\sum_{j=1}^d f(\nu_j)$ for all $i \in [d]$, where $\nu \in \mathbb{R}^d$ is the input vector, $\nu_i$ is the $i$-th entry of $\nu$, and $\sigma(\nu)_i$ is the $i$-th entry of the output $\sigma(\nu)$. The function $f : \mathbb{R} \mapsto \mathbb{R}$ can be any suitable univariate function, with softmax being the special case where $f(\cdot) = \exp(\cdot)$.

In §4.3, we identified specific patterns in the KQ and OV circuits of trained softmax attention models. A key property promotinh these patterns is the exponential function underlying softmax. These patterns allow the learned transformer to implement a sum of kernel regressors where the exponential function serves as the kernel. Using the first-order approximation $\exp(x) \approx 1 + x$, we proved that the model approximately implements the debiased GD. This raises questions:

*(a) Suppose we use a different nonlinear activation in the attention, do we expect similar patterns in the attention weights?*
*(b) Does this transformer also implement debiased GD?*

To answer these questions, we conduct additional experiments on the two-head attention model—using the same setup as §3—by replacing softmax with other normalized activations $f$ that satisfy the first order approximation $f(x) \approx 1 + C_f \cdot x$ for some constant $C_f > 0$. We test $f_1(x) = 1 + Cx$, $f_2(x) = (1 + Cx)^2$ and $f_3(x) = 1 + \tanh(x)$ with $C \in \{0.5, 0.8, 1\}$. Their first-order coefficients $C_f$ are given by $C_{f_1} = C$, $C_{f_2} = 2C$ and $C_{f_3} = 1$ respectively.

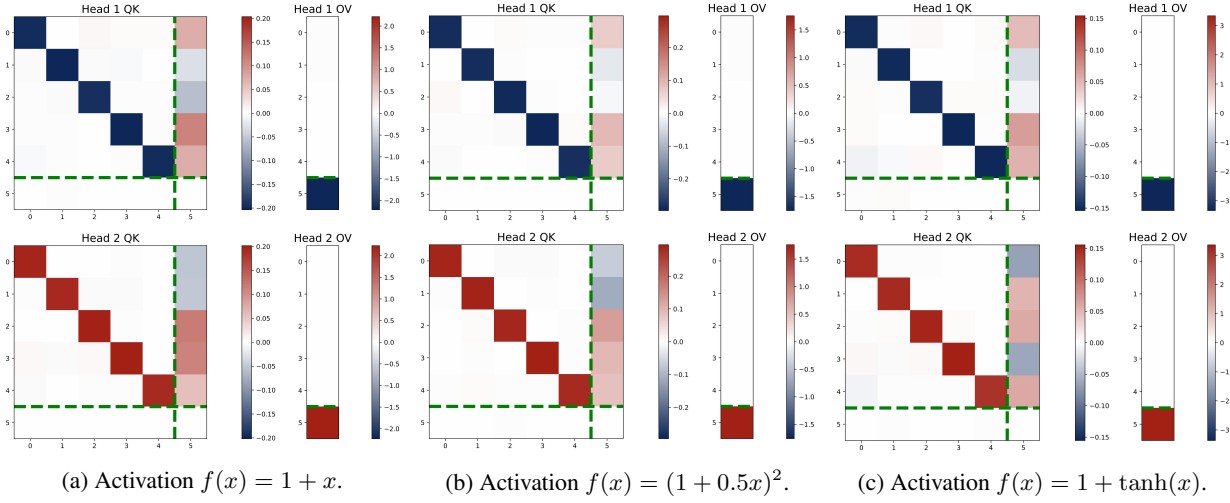

(a) Activation $f(x) = 1 + x$.      (b) Activation $f(x) = (1 + 0.5x)^2$.      (c) Activation $f(x) = 1 + \tanh(x)$.

*Figure 8.* Heatmap of KQ matrices and OV vectors of trained two-head attention using alternative activation functions. The trained models consistently exhibit the same patterns shared by softmax attention, though the parameters of different models converge to different values.

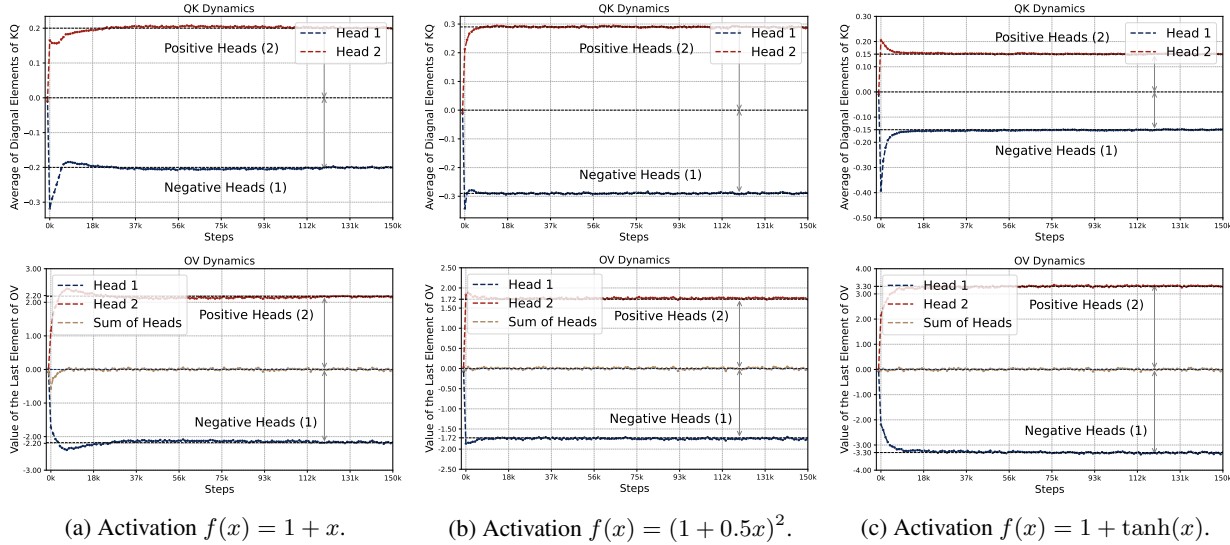

(a) Activation $f(x) = 1 + x$.      (b) Activation $f(x) = (1 + 0.5x)^2$.      (c) Activation $f(x) = 1 + \tanh(x)$.

*Figure 9.* Comparison of the training dynamics of KQ and OV parameters for a two-head attention model with various activation functions. In particular, Figure (c) illustrates the dynamics of the case $f(\cdot) = 1 + \tanh(\cdot)$, which are closer to those of the vanilla softmax attention model.

| Activation function | $\exp(x)$ | $1 + \tanh(x)$ | $1 + Cx$ | | | $(1 + Cx)^2$ | | |
|---|---|---|---|---|---|---|---|---|
| | | | $C = 0.5$ | $0.8$ | $1$ | $C = 0.5$ | $0.8$ | $1$ |
| $\lvert\omega^{(1)}\rvert$ | 0.1267 | 0.1504 | 0.3677 | 0.2411 | 0.1979 | 0.2882 | 0.1810 | 0.1452 |
| $\lvert\mu^{(1)}\rvert$ | 3.5006 | 3.3362 | 2.3963 | 2.2819 | 2.2221 | 1.7561 | 1.7487 | 1.7448 |
| $\eta_{\mathsf{eff}}$ | 0.8871 | 1.0035 | 0.8810 | 0.8802 | 0.8794 | 1.0122 | 1.0128 | 1.0132 |

*Table 1.* Comparison of learned parameters with different activationss for two-head attention model. The table reports the learned $\lvert\omega^{(1)}\rvert$, $\lvert\mu^{(1)}\rvert$, and the effective learning rate $\eta_{\mathsf{eff}} = 2C_f \cdot \lvert\omega^{(1)}\rvert \cdot \lvert\mu^{(1)}\rvert$, where $C_f$ denotes the coefficient in the first-order Taylor expansion of each activation function $f$. The exponential activation corresponds to the standard softmax attention. Despite different scaling patterns in the individual parameters, the effective learning rates $\eta_{\mathsf{eff}}$ remain remarkably consistent across all activation functions, demonstrating that trained models implement similar debiased GD predictors regardless of activation choice.

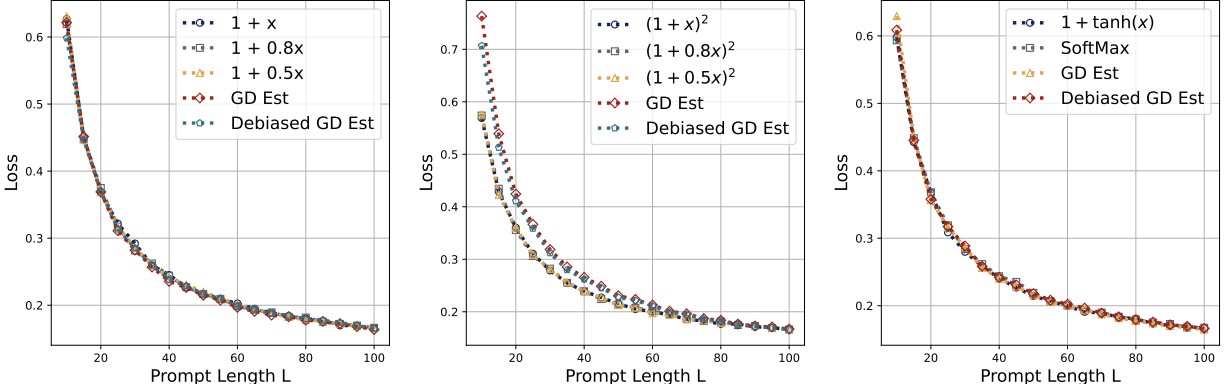

*Figure 10.* Comparison of two-head attention models with different activation functions and the corresponding canonical predictors—vanilla GD and debiased GD predictor—with effective learning rate $\eta_{\mathsf{eff}} = 2C_f \cdot \lvert\omega^{(1)}\rvert \cdot \lvert\mu^{(1)}\rvert$. As shown in the figure, the ICL prediction error of attention models with different activations closely matches that of the vanilla GD and debiased GD predictors. While the normalized quadratic activation exhibits slight deviations from the GD predictors for small $L$, the loss curves ultimately converge as $L$ increases.

In Figure 8, we plot the KQ and OV parameters for all these attention models. We observe the consistent positive-negative pattern in the attention weights. That is, the first two observations of §3 are still true. This implies that all of these learned attention models implement a sum of kernel regressors. Moreover, by examining the magnitude of the KQ and OV parameters, we observe that the magnitude of $\omega = (\omega^{(1)}, \omega^{(2)})$ is still relatively small while the magnitude of $\mu = (\mu^{(1)}, \mu^{(2)})$ is large. This is also consistent with the softmax attention. Mathematically, this means that the transformer predictors can be written as

$$\widehat{y}_q = \lvert\mu^{(1)}\rvert \cdot \left( \sum_{\ell=1}^{L} \frac{y_\ell \cdot f(\lvert\omega^{(1)}\rvert \cdot x_\ell^\top x_q)}{\sum_{\ell=1}^{L} f(\lvert\omega^{(1)}\rvert \cdot x_\ell^\top x_q)} - \sum_{\ell=1}^{L} \frac{y_\ell \cdot f(-\lvert\omega^{(1)}\rvert \cdot x_\ell^\top x_q)}{\sum_{\ell=1}^{L} f(-\lvert\omega^{(1)}\rvert \cdot x_\ell^\top x_q)} \right) \approx \frac{\eta_{\mathsf{eff}}}{L} \sum_{\ell=1}^{L} y_\ell \cdot \bar{x}_\ell^\top x_q,$$

where $\eta_{\mathsf{eff}} = 2C_f \cdot \lvert\omega^{(1)}\rvert \cdot \lvert\mu^{(1)}\rvert$, and $\bar{x}_\ell$ is the debiased covariate and the approximation can be similarly derived as in Remark F.1.

In Table 1, we report the limiting values of $\lvert\omega^{(1)}\rvert$, $\lvert\mu^{(1)}\rvert$ and $\eta_{\mathsf{eff}}$ of these models. We observe that while the values of $\lvert\omega^{(1)}\rvert$ and $\lvert\mu^{(1)}\rvert$ differ across different activations, the values of $\eta_{\mathsf{eff}}$ are all close to one. This suggests that all these models effectively learn the same debiased GD predictor $\widehat{y}_q^{\mathsf{gd}}(\eta^*)$. Here $\eta^* = (1 + (1 + \sigma^2) \cdot d/L)^{-1} \approx 1$ when $d/L \to 0$.

Moreover, in Figure 10 we report the ICL prediction errors of these learned multi-head attention models, together with the error of the debiased GD with corresponding effective learning rate. We observe that these error curves are highly consistent with debiased GD. Notice that all these models can readily generalize in length, which seems a benefit of the normalized activation in attention. Thus, we expect that Theorem 4.2 can be generalized to attention models based on normalized activations in general, as long as $f(x) \approx 1 + C_f \cdot x$ around $x = 0$. Finally, we remark the choice of the bias term 1 is just

for simplicity, and the argument can be readily generalized to other positive real numbers thanks to the normalization effect in softmax attention.

Finally, we plot the training dynamics of the KQ and OV parameters in Figure 9. Interestingly, the training dynamics across different activations exhibit slightly different behaviors. The behavior of $1 + \tanh(x)$ is similar to the softmax, where $|\omega^{(1)}|$ first increases and then decreases. The behavior of the other two cases in Figure 9 seems more complicated. We defer their study to future work.

In summary, we show that for the class of multi-head attention models normalized activations, as long as the activation $f$ satisfies the first-order condition $f(x) \approx 1 + C_f \cdot x$, the first three observations in §3 remain valid. That is, the attention weights share the patterns and the limiting attention models learn the same debiased GD predictor. However, the training dynamics are more subtle, which seem to rely on the choice of activation function.

### B.3. Extension to Anisotropic Covariates

In the following, we examine how the data distribution contributes to the observed patterns in the learned attention weights. To this end, we focus on the anisotropic case where the covariates are sampled from a centered Gaussian distribution with a general covariance structure. In particular, we investigate the following questions:

*(a) Can multi-head attention solve in-context linear regression with anisotropic covariates? (b) Do we observe the same patterns in the learned transformer? Does the learned transformer implement a GD algorithm approximately?*

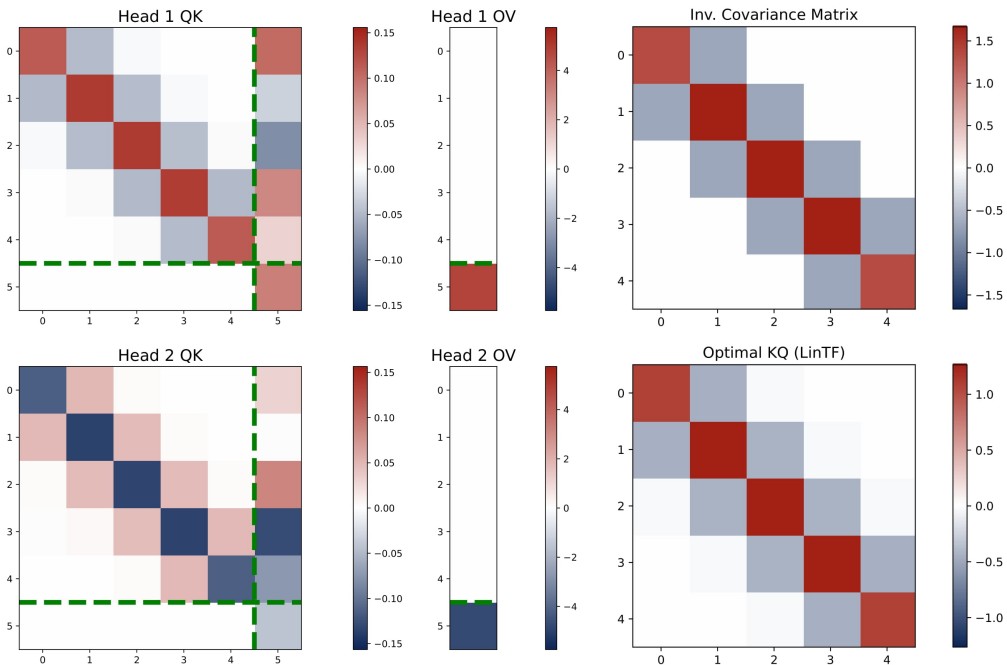

*Figure 11.* Heatmap of KQ matrices and OV vectors trained on anisotropic covariates. The left two Figures illustrate the patterns of trained two-head attention model, and the right Figures plot the inverse of covariance matrix $\Sigma^{-1}$ and the conjectured optimal KQ pattern $\widetilde{\Sigma}^{-1}$ in (B.6). The learned KQ matrices exhibit a dominant tridiagonal structure with small values in the $(i, i+2)$ and $(i+2, i)$ positions for all $i \in [d]$, suggesting the model deviates from a strict implementation of pre-conditioned GD using $\Sigma^{-1}$ since the inverse covariance matrix should be tridiagonal. However, this pattern aligns with the conjectured structure $\widetilde{\Sigma}^{-1}$, indicating the model implements a pre-conditioned GD predictor with structural adjustment.

In our experiment, we let the covariate distribution be $\mathsf{P}_x = \mathcal{N}(0, \Sigma)$, where $\Sigma$ is a Kac-Murdock-Szegö matrix (Fikioris, 2018) with parameter $\rho = 0.5$. That is, the $(i, j)$-th entry of $\Sigma$ is $\Sigma_{ij} = \rho^{|i-j|}$. We train a two-head attention model with $d = 5$ and $L = 40$.

**Transformer Learns Sum of Kernel Regressors.** We observe that the diagonal pattern of the KQ circuits disappears, but the pattern of the OV circuits as in Observation 1 in §3 persists. Moreover, as in Observation 2 in §3, the weight matrices of the two heads sum to zero, which means they also split into a positive and a negative head. In particular, the learned weight matrices can be written as

$$K^{(h)\top} Q^{(h)} = \begin{bmatrix} (-1)^{h+1} \cdot \gamma \cdot \Omega & * \\ \mathbf{0}_d^\top & * \end{bmatrix}, \quad O^{(h)} V^{(h)} = \begin{bmatrix} * & * \\ \mathbf{0}_d^\top & (-1)^{h+1} \cdot \mu_\gamma \end{bmatrix}. \tag{B.3}$$

Here $\gamma > 0$ is a small scaling parameter, $\mu_\gamma$ is defined in (**??**), and $\Omega \in \mathbb{R}^{d \times d}$ is a positive definite matrix. That is, the first head is positive and the second is negative. The properties of sign-matching and zero-sum OV still hold. Moreover, the magnitude of the OV circuits is roughly the same as in the isotropic case. While KQ matrices still have a small magnitude, they are no longer proportional to an identity matrix. See the first column of Figure 11 for details.

As a result, the learned transformer still implements a sum of two kernel regressors:

$$\widehat{y}_q = \mu_\gamma \cdot \left( \sum_{\ell=1}^{L} \frac{y_\ell \cdot \exp(\gamma \cdot x_\ell^\top \Omega x_q)}{\sum_{\ell=1}^{L} \exp(\gamma \cdot x_\ell^\top \Omega x_q)} - \sum_{\ell=1}^{L} \frac{y_\ell \cdot \exp(-\gamma \cdot x_\ell^\top \Omega x_q)}{\sum_{\ell=1}^{L} \exp(-\gamma \cdot x_\ell^\top \Omega x_q)} \right), \tag{B.4}$$

where the kernel function is induced by the bivariate function $F(x, x'; \gamma, \Omega) = 1/\gamma \cdot \exp(-\gamma \cdot x^\top \Omega x')$. Since $\gamma$ is small, we can similarly perform a first-order Taylor expansion to $\exp(\cdot)$ in (B.4), which implies that $\widehat{y}_q$ in (B.4) is close to a pre-conditioned version of the debiased GD:

$$\widehat{y}_q \approx \frac{2\gamma \cdot \mu_\gamma}{L} \sum_{\ell=1}^{L} y_\ell \cdot \bar{x}_\ell^\top \Omega x_q. \tag{B.5}$$

Thus, the effective learning rate, $\eta = 2\gamma \cdot \mu_\gamma$, is the same as in the isotropic case.

**Pre-Conditioning Matrix.** It remains to determine the pre-conditioning matrix $\Omega$. Recall that in the isotropic case, we prove that multi-head softmax attention recovers the estimator found by the trained linear attention. For the anisotropic case, it is proved that linear attention finds a pre-conditioned GD predictor (Ahn et al., 2023b), which is given by

$$\widehat{y}_q^{\mathsf{vgd}} := \frac{1}{L} \cdot \sum_{\ell=1}^{L} y_\ell \cdot x_\ell^\top \widetilde{\Sigma}^{-1} x_q, \quad \text{with } \widetilde{\Sigma} = \left(1 + \frac{1}{L}\right) \Sigma + \frac{\mathrm{tr}(\Sigma) + d\sigma^2}{L} \cdot I_d. \tag{B.6}$$

Besides, as we will show in §G.4, $\widetilde{\Sigma}$ corresponds to the optimal pre-conditioning matrix for the pre-conditioned GD, which minimizes the ICL risk. We conjecture that $\Omega$ is close to $\widetilde{\Sigma}^{-1}$, since (i) two-head softmax attention model and the linear attention when $\gamma$ is small, and (ii) $\widetilde{\Sigma}$ enjoys optimality over all pre-conditioning matrices. In the right column of Figure 11, we plot $\Sigma^{-1}$ and $\widetilde{\Sigma}^{-1}$. A closer examination of Figure 11 shows that $\Omega$ is closer to $\widetilde{\Sigma}^{-1}$ than $\Sigma^{-1}$.

Note when $L$ is sufficiently large, $\widetilde{\Sigma}$ is close $\Sigma$. Thus, while we cannot prove $\Omega = \widetilde{\Sigma}^{-1}$, we know that when $L$ is sufficiently large, the transformer predictor is equal to $L^{-1} \cdot \sum_{\ell=1}^{L} y_\ell \cdot x_\ell^\top \Sigma^{-1} x_q$. Also see Figure 14 in §C.2, which shows that $\Omega^{-1}$, $\Sigma$, and $\widetilde{\Sigma}$ are close.

### B.4. Extension to In-Context Multi-Task Regression

In the following, we consider the *multi-task* in-context regression, where the response variable is a vector. In this setting, as we will show below, depending on the number of heads and the number of tasks, the learned transformer may exhibit different patterns.

**Task Formulation.** We first introduce the data generation process of multi-task linear regression as follows, where each task has its own set of features.

**Definition B.1** (Multi-task Linear Model). *Given $d \in \mathbb{Z}^+$, we assume the covariate $x \in \mathbb{R}^d$ is independently sampled from $\mathsf{P}_x$, and let $\beta \in \mathbb{R}^d$ be a fixed signal parameter. Let $N \in \mathbb{Z}^+$ denote the number of tasks. For each task $n \in [N]$, let $\mathcal{S}_n \subseteq [d]$ denote a nonempty set of indices for task $n \in [N]$. Let $\beta_{\mathcal{S}_n}$ and $x_{\mathcal{S}_n}$ denote the subvectors of $\beta$ and $x$ indexed by $\mathcal{S}_n$. We define the response vector $y = [y_1, \ldots, y_N]^\top \in \mathbb{R}^N$ by letting $y_n = \beta_{\mathcal{S}_n}^\top x_{\mathcal{S}_n} + \epsilon_n$ for all $n \in [N]$, where $\epsilon_n \overset{i.i.d.}{\sim} \mathcal{N}(0, \sigma^2)$.*

In other words, each task $n$ only uses features in $\mathcal{S}_n$ and the response is generated by a linear model with the corresponding signal parameter $\beta_{\mathcal{S}_n}$ and covariate $x_{\mathcal{S}_n}$. Now we introduce the ICL problem for multi-task regression, which is almost the same as in the definition in §2.

**Definition B.2.** *Suppose that the signal set $\{\mathcal{S}_n\}_{n \in [N]}$ is fixed but unknown. To perform ICL, we first generate $\beta \sim \mathsf{P}_\beta$ and then generate $L$ demonstration examples $\{(x_\ell, y_\ell)\}_{\ell \in [L]}$ where $x_\ell \overset{i.i.d.}{\sim} \mathsf{P}_x$ and $y_\ell$ is generated by the linear model in Definition B.1 with parameter $\beta$. Moreover, we generate another covariate $x_q \sim \mathsf{P}_x$ and the goal is to predict the response $y_q \in \mathbb{R}^N$ for the query $x_q$.*

Note that for each sequence $Z_{\mathrm{ebd}} \in \mathbb{R}^{(d+1) \times (L+N)}$ defined as in (2.1), the signal set $\{\mathcal{S}_n\}_{n \in [N]}$ is shared while the signal parameter $\beta$ is randomly generated. Thus, we expect that the trained transformer model should be able to bake the shared structure $\{\mathcal{S}_n\}_{n \in [N]}$ into the model. Chen et al. (2024a) study a special case of multi-task scenario with $H = N$. They assume that, up to a rotation in $y$, each $\mathcal{S}_n$ does not overlap with each other, and $\bigcup_{n \in [N]} \mathcal{S}_n = [d]$. In this special case, each head learns to solve a separate linear task. That is, for each task $n$, there exists a head $h(n)$ such that the $h(n)$-th head solves the task $n$ using the single-head attention estimator given in (**??**). Moreover, $\{h(n)\}_{n \in [N]}$ is a permutation of $[N]$. Thus, in this special setup, the learned transformer model is a sum of $N$ independent kernel regressors, where the $n$-th regressor is based on the $\mathcal{S}_n$-subvector of $x_\ell$ and the $n$-th entry of $y_\ell$.

In contrast, we allow the signal sets $\{\mathcal{S}_n\}_{n \in [N]}$ to overlap with each other, which exhibits more complicated behaviors in the learned transformer model. In our experiment, we study a simple two-task scenario with $d = 6$, $L = 40$, $\sigma^2 = 0.1$, and $N = 2$. We set $\mathcal{S}_1 = \{1, 2, 3, 4\}$ and $\mathcal{S}_2 = \{3, 4, 5, 6\}$, which have two overlapping entries. For brevity, we defer the details of the experiment results and their interpretations to §C.3. We present the main findings as follows.

**Global Pattern.** In the multi-task setip, the KQ and OV matrices are $(N + d) \times (N + d)$ matrices. Similar to §3, we focus on the effective parameters that do not meet the zero vector in $Z_{\mathrm{ebd}}$, i.e., the top-left $(d + N) \times d$ submatrix of KQ matrix and the last $N$ rows of OV matrix:

$$KQ^{(h)} = \begin{bmatrix} KQ_{11}^{(h)} & * \\ KQ_{21}^{(h)} & * \end{bmatrix} \in \mathbb{R}^{(d+N) \times (d+N)}, \quad OV^{(h)} = \begin{bmatrix} * & * \\ OV_{21}^{(h)} & OV_{22}^{(h)} \end{bmatrix} \in \mathbb{R}^{(d+N) \times (d+N)}. \tag{B.7}$$

Here, $KQ_{11}^{(h)} \in \mathbb{R}^{d \times d}$ and $OV_{22}^{(h)} \in \mathbb{R}^{N \times N}$. Regardless of the number of heads, we observe the following consistent global pattern: There exists $\omega^{(h)} \in \mathbb{R}^d$ and $\mu^{(h)} \in \mathbb{R}^N$ such that

$$KQ_{11}^{(h)} = \mathrm{diag}(\omega^{(h)}), \qquad OV_{22}^{(h)} = \mathrm{diag}(\mu^{(h)}), \qquad KQ_{21}^{(h)} = OV_{21}^{(h)} = \mathbf{0}_{N \times d}, \qquad \forall h \in [H]. \tag{B.8}$$

Therefore, the KQ circuit exhibits a diagonal-only pattern in the top $d \times d$ submatrix with other entries equal to zero, and the OV circuit has non-zero values exclusively in the last $N \times N$ matrix. Following this, the trained transformer essentially behaves like a single-task model, where each head uses a softmax function to compute similarity scores $p^{(h)} = \mathtt{smax}(X \cdot \omega^{(h)} \odot x_q) \in \mathbb{R}^L$, then produces a weighted response $\mu^{(h)} \cdot \sum_{\ell=1}^L p_\ell^{(h)} \cdot y_\ell$, where $p_\ell^{(h)}$ is the $\ell$-th entry of $p^{(h)}$. In other words, the learned transformer model is a sum of $H$ kernel regressors, but the weights computed by the kernel are used to aggregated the vector-valued responses.

**Local Patterns.** However, different from the single task case, here each $\omega^{(h)}$ does not corresponds to an all-one vector. Rather, $\omega^{(h)}$ can have different magnitudes in different entries, which reflects the tension of feature selections across different tasks. Intuitively, focusing on each attention head $h$, the value of $\omega^{(h)}$ determines the similarity scores $p^{(h)}$. If we only use it to solve the first task, we expect that $\omega^{(h)}$ only has non-zero entries in $\mathcal{S}_1$. Similarly, if we only use it to solve the second task, we expect that $\omega^{(h)}$ only has non-zero entries in $\mathcal{S}_2$. But when it contributes to solving both tasks, the problem is tricky. How much effort does this head put into solving task one and task two? This will be reflected in the magnitude of $\omega^{(h)}$ in the overlapping entries.

We run experiment with multi-head attention models with $H$ ranging from $1$ to $4$. Depending on the number of heads, the trained transformer exhibit four different local patterns:

(i) $H = 1$: Single-head attention learns one weighted kernel regressor, assigning higher weights to the informative entries, i.e., the entries that are shared by more $\mathcal{S}_n$'s. On our simple case, these entries are $\mathcal{S}_1 \cap \mathcal{S}_2 = \{3, 4\}$.

(ii) $H = 2$: The two-head attention learns to act as one weighted pre-conditioned GD predictor with also higher weight assigned to the informative entries.

(iii) $H \geq 2N$: The attention heads are clustered into $N$ groups corresponding to $N$ tasks, where each group contains at least 2 heads. Within each group, the attention heads learn to solve a single task, say, task $n$, via the same mechanism as in the single-task setting. But the main difference is that these heads learns to use the true support $\mathcal{S}_n$ to solve the task. Thus, this group of heads learns to implement a debiased GD predictor based on data $\{x_{\ell, \mathcal{S}_n}, y_{\ell, n}\}_{\ell \in [L]}$, where $x_{\ell, \mathcal{S}_n}$ is the subvector of $x_\ell$ indexed by $\mathcal{S}_n$ and $y_{\ell, n}$ is the $n$-th entry of $y_\ell$. In particular, the knowledge of $\{\mathcal{S}_n\}$ is encoded in the KQ circuit, which is reflected in the support of parameter $\omega^{(h)}$. Overall, the transformer learns to implement $N$ independent debiased GD predictors at the same time.

(iv) $2 < H < 2N$: In this case, the learned pattern varies in repeated runs. In our experiment with $H = 3$ and $N = 2$, we observe that an interesting head that simultaneously serves as a positive head for task one and a negative head for task two. Interestingly, in this case, the model effectively implements $N$ independent debiased GD predictors as in the previous case. But it uses a complicated superposition mechanism (Elhage et al., 2022) to achieve this.

## C. Auxiliary Experiment Results

In this section, we provide additional experiment results to support the discussions in §3 and §B.

### C.1. Auxiliary Experiment Results for Section 3

**Experiment Setups in §3.** For all experiments in the main paper, we set $L = 40$, $d = 5$, $\sigma^2 = 0.1$ for data generation and employ the Adam optimizer with a learning rate $\eta = 10^{-3}$, a batch size $n_{\text{batch}} = 256$, and update for $T = 5 \times 10^5$ training steps. The weight matrices are initialized using the default random initialization in PyTorch. Each experiment takes about 15 minutes on a single NVIDIA A100 GPU. Furthermore, we found that the results are not sensitive to the choice of optimization algorithm, hyperparameters such as batch size and learning rate, or the configuration of the ICL data-generating process.

To back up the argument that multi-head attentions implement the same predictor, we consider the same setting as in §3 with various number of heads and visualize the training dynamics of the KQ and OV parameters. In particular, we consider $H \in \{2, 3, 4\}$, and set $d = 5$, $L = 40$, and $\sigma^2 = 0.1$.

In Figure 12, we plot the training dynamics of the KQ and OV parameters for different numbers of heads. The first column shows $\{\omega^{(h)}\}_{h \in [H]}$ and the second column shows $\{\mu^{(h)}\}_{h \in [H]}$ for different multi-head attention models. Our findings are consistent with theoretical results in §B.1, and can be summarized as follows:

(i) The patterns of the KQ and OV matrices are consistent across different numbers of heads, and are the same as in those described in Observation 1 and Observation 2 in §3. In particular, the property of homogeneous KQ scaling and zero-sum OV hold for all models.

(ii) More importantly, across different models, the limiting value of $\omega^{(h)}$'s and $\mu_+ = \sum_{h \in \mathcal{H}_+} \mu^{(h)}$ are the same. Besides, $\mu_- = \sum_{h \in \mathcal{H}_-} \mu^{(h)} = -\mu_+$. In particular, for all non-dummy heads, the magnitude of $|\omega^{(h)}| \sim 0.13$ for each model, and $\mu_+$ and $|\mu_-|$ are both $\sim 3.5$.

(iii) When $H > 2$, we observe dummy heads, which corresponds to a head $h \in [H]$ such that $|\omega^{(h)}| \approx |\mu^{(h)}| \approx 0$.

By the observation (ii) above, we know that when $H$ ranges from 2 to 4, the trained multi-head attention models all implements the same predictor as given in (??), which is a sum of two kernel regressors. Moreover, since the magnitude of the KQ parameters are small, as we show in (??), such a predictor is approximately equivalent to a debiased gradient descent predictor. Therefore, the experimental results support our theoretical findings in §B.1 that multi-head attentions learn to implement the same predictor.

**Training Dynamics of Dummy Head.** As a supplementary to Figure 4, which only plots the two-head cases, we provide the evolution of KQ and OV circuits for the four-head cases. Different from the two-head scenarios where the dummy heads will not occur due to the constraint of solution manifold $\mathscr{S}^*$, i.e., there should be at least one positive head and one negative head, there may occur dummy heads when $H \geq 3$. In our experiment, we observe a dummy head in the four-head model.

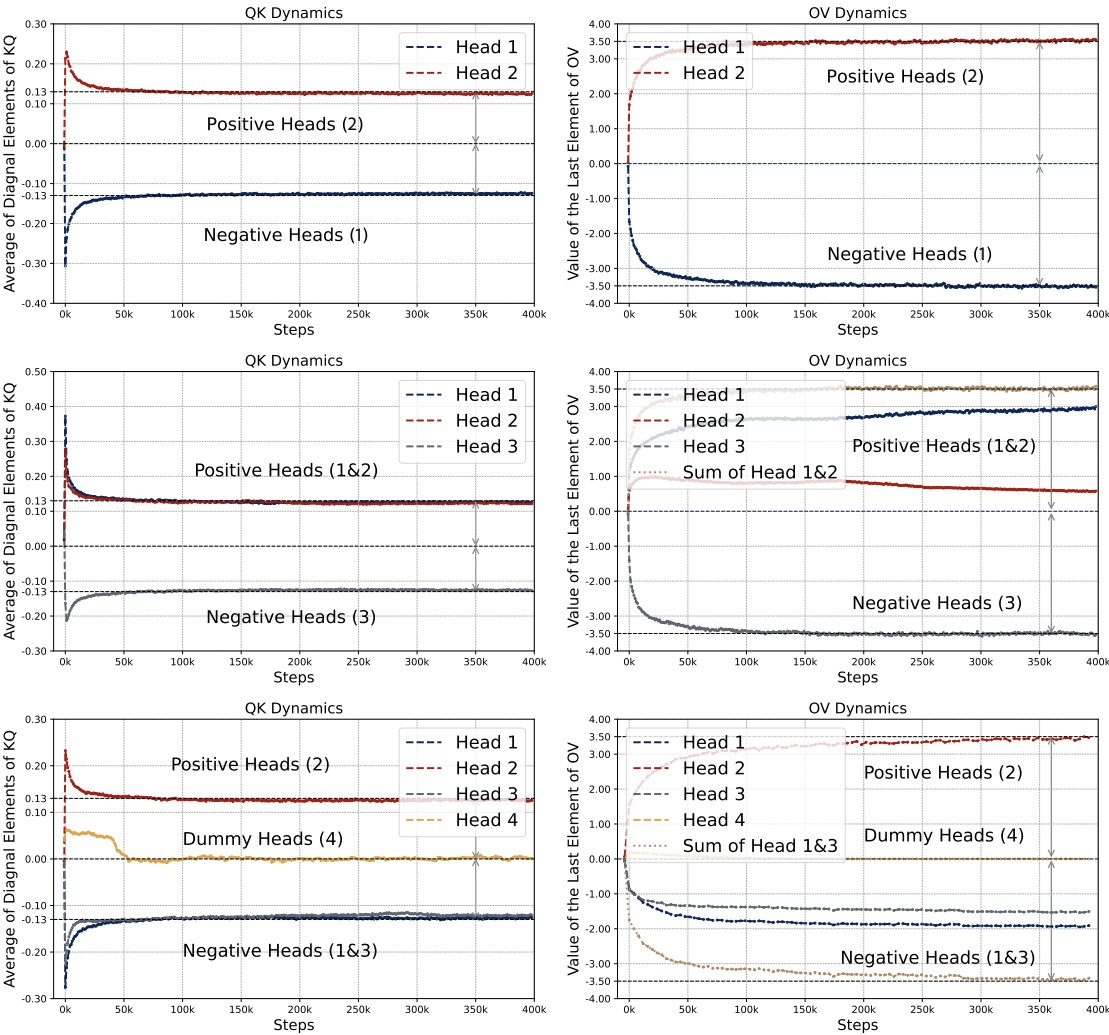

*Figure 12.* Comparison of training dynamics under the same setup with the number of heads ranging from 2 to 4. The magnitude of the KQ parameters is the same for all non-dummy heads in all models. The values of $\mu_+$ and $\mu_-$ are also consistent across different models. This shows that all these multi-head attention models learn to implement the same predictor. Furthermore, in this experiment, we observe an dummy head when $H = 4$. Dummy heads will appear when $H > 2$.

We note that whether the dummy head appears and the number of dummy heads are random and seem to depend on the initialization and the optimization algorithm. We also observe dummy heads for $H = 3$ with different random seeds.

Furthermore, as shown in Figure 13, the KQ matrix of Head 4 demonstrates a chaotic pattern at the early stage and converges to a zero-matrix. In correspondence, the OV vectors of the dummy heads converge to zero vectors, ensuring the dummy heads will not contribute to the output. For positive and negative heads, the behavior is similar to two-head cases. These observations can also be seen from the last row of Figure 12.

### C.2. Additional Results for Anisotropic Covariates in Section B.3

In the following, we provide additional details of the experiments on anisotropic covariates, which is introduced in §B.3.

**Kac-Murdock-Szegö matrix** A key property of the Kac-Murdock-Szegö matrix is that its inverse matrix is a tridiagonal matrix. Let $\Lambda = (1 - \rho^2) \cdot \Sigma^{-1}$, then the entries of $\Lambda$ are given by

$$\Lambda_{ii} = (1 + \rho^2) \cdot \mathbb{1}(i \neq 1 \text{ and } d) + \mathbb{1}(i = 1 \text{ or } d), \quad \Lambda_{i,i+1} = \Lambda_{i+1,i} = -\rho, \quad \text{ for all } i \in [d].$$

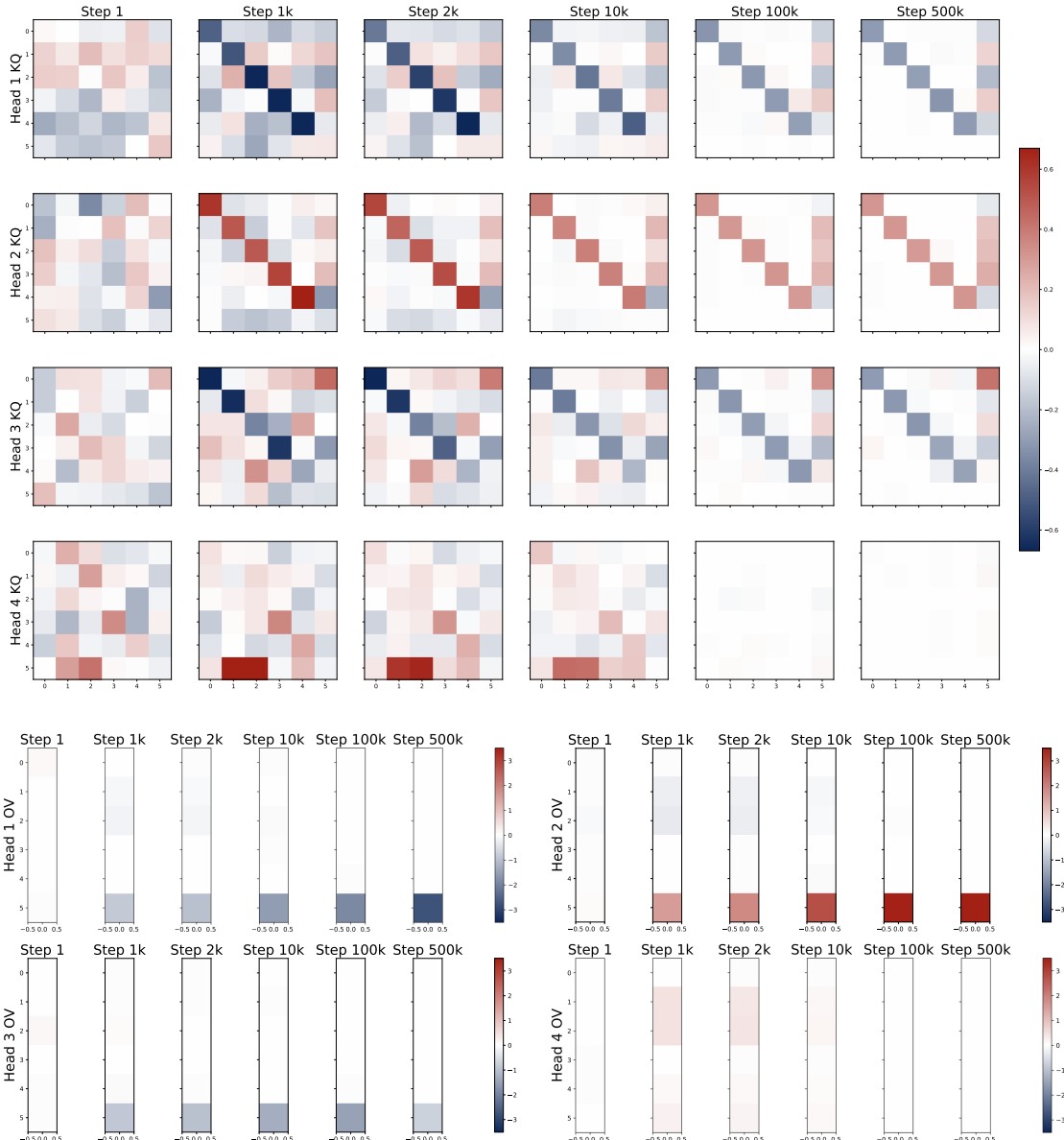

Figure 13. Heatmaps of the KQ matrices and OV vectors along training epochs with $H = 4$, $d = 5$, and $L = 40$, trained over $T = 5 \times 10^5$ steps with random initialization. The behaviors of the first three heads are similar to the two-head case as shown in Figure 4. The KQ matrix of Head 4 (dummy head) demonstrates a chaotic pattern at the early stage and converges to a zero-matrix. The OV vectors of the dummy heads converge to zero vectors.

The rest of the entries are zero. See, e.g., Theorem 2.1 in Fikioris (2018) for more details. We use this property to examine the value of the KQ matrices learned by the attention model.

**KQ Implements Pre-Conditioned GD with $\Omega = \widetilde{\Sigma}^{-1}$.** We show in (B.5) that multi-head attention learns to implement a pre-conditioned version of the debaised GD, where $\Omega$ is the pre-conditioning matrix. Here $\gamma \cdot \Omega$ appears in the KQ matrix of the positive head. In Figure 14, we plot the inverse of the top-left $d \times d$ submatrices of the KQ matrices, and compare them with $\Sigma$ and $\widetilde{\Sigma}$. This figures shows that these matrices are all very close, which is the case when $L$ is large.

In addition, since the $\Sigma^{-1}$ is a tridiagonal matrix, its $(i, i+2)$- and $(i+2, i)$-entries are zero. As shown in Figure 11, $\Omega$ not only have significant values in the tridiagonal entries but also exhibit very small values in the $(i, i+2)$- and $(i+2, i)$-entries. This indicates that the trained model does not strictly implement pre-conditioned GD using $\Sigma^{-1}$. In comparison, the adjusted

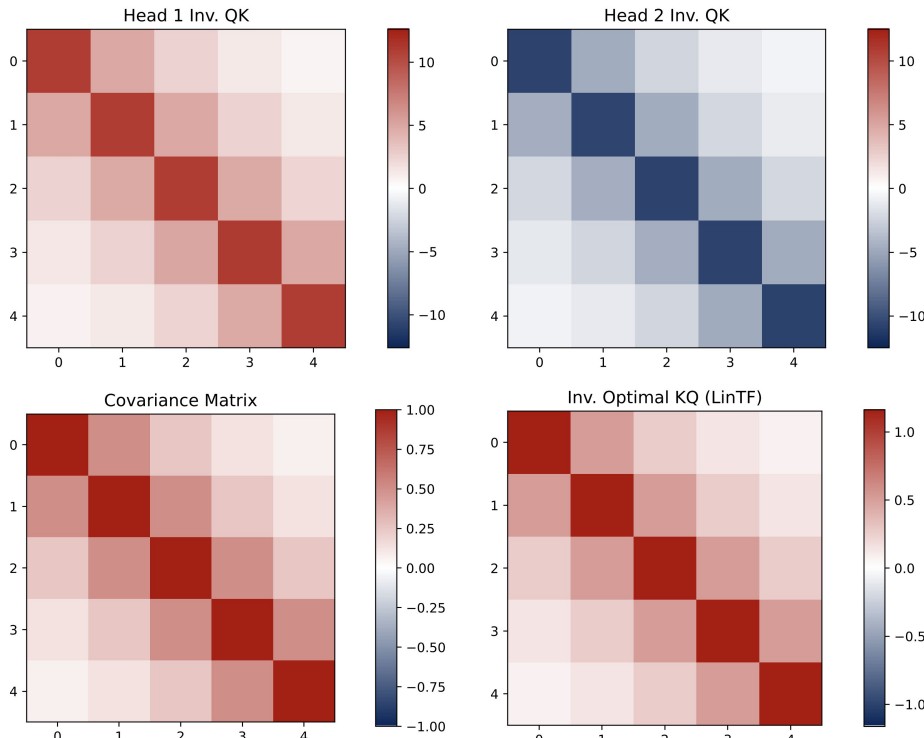

*Figure 14.* Comparison of inverse KQ matrices trained on anisotropic covariates with the ground-truth covariance matrix $\Sigma$ and the inverse matrix of conjectured optimal KQ pattern $\widetilde{\Sigma}$ in (B.6). The learned KQ matrices exhibit an positive-negative pattern, similar to the isotropic case in §3. Additionally, the inverse KQ matrix is nearly proportional to the ground-truth covariance matrix and $\widetilde{\Sigma}$, suggesting that the model implements a pre-conditioned GD predictor.

$\widetilde{\Sigma}^{-1}$ (see bottom-right of Figure 11), also exhibits small values beyond the tridiagonal entries due to the perturbation of $\mathrm{tr}(\Sigma)/L \cdot I_d$. By comparing KQ matrices and $\widetilde{\Sigma}^{-1}$, we can show that the KQ matrices of two heads take the form $\gamma \cdot \widetilde{\Sigma}$ and $-\gamma \cdot \widetilde{\Sigma}$ with a small scaling $\gamma$, aligning with the conjecture in §B.1. Meanwhile, note that the difference between $\Sigma^{-1}$ and $\widetilde{\Sigma}^{-1}$ is roughly $O(1/L)$, which is negligible when $L$ is large. Thus, it suffices to identify the estimator in (B.6) with simple pre-conditioned GD predictor $L^{-1} \sum_{\ell=1}^{L} y_\ell \cdot x_\ell^\top \Sigma^{-1} x_q$ in the low-dimensional regime where $d/L \to 0$.

### C.3. Additional Results for Multi-Task Linear Regression in Section §B.4

In the following, we introduce the details of the experiments on multi-task linear regression, which is discussed in §B.4.

**Experimental Setups.** Recall that the data generation process is defined in Definitions B.1 and (B.2). We let $\{(x_\ell, y_\ell)\}_{\ell \in [L]}$ denote the ICL examples, where $x_\ell \in \mathbb{R}^d$ and $y_\ell \in \mathbb{R}^N$. We consider the isotropic case with $\mathsf{P}_x = \mathcal{N}(0, I_d)$ and $\mathsf{P}_\beta = \mathcal{N}(0, I_d/d)$. Moreover, in our experiment, we focus on the two-task case with $N = 2$, $d = 6$, $L = 40$, $\sigma^2 = 0.1$. Moreover, we set $\mathcal{S}_1 = \{1, 2, 3, 4\}$ and $\mathcal{S}_2 = \{3, 4, 5, 6\}$, i.e., the features of the tasks have overlap $\{3, 4\}$. We train multi-head attention models with $H \in \{1, 2, 3, 4\}$. The rest of the experimental setup is the same as that in §3.

To simplify the notation, we let $\mathcal{S}^* = \mathcal{S}_1 \cap \mathcal{S}_2$ and $\mathcal{S}^c = [N] \backslash \mathcal{S}^*$. Besides, for each $n \in [N]$, we let $y_{\ell,n}$ denote the $n$-th entry of the response vector $y_\ell$. For any subset $\mathcal{S} \subseteq [d]$, we let $x_{\ell,\mathcal{S}}$ denote the subvector of $x_\ell$ with indices in $\mathcal{S}$. As discussed in §B.3, the trained transformer exhibits global patterns in the KQ and OV matrices. In specific, by writing the KQ and OV matrices as block matrices as in (B.7), we have (B.8) for all $h \in [H]$. In other words, each head $h$ is captured by parameters $\omega^{(h)} \in \mathbb{R}^d$ and $\mu^{(h)} \in \mathbb{R}^N$. As we will show as follows, each four attention models illustrate one of the four possible modes of behavior respectively.

**Mode I ($H = 1$): Single Weighted Kernel Regressor.** In Figure 15, we visualize the learned pattern of single-head attention. Upon closer examination, the third and fourth entries, corresponding to $\mathcal{S}_1 \cap \mathcal{S}_2$, have a larger value ($\sim 0.5$) than other diagonal entries ($\sim 0.42$). That is, $\omega_j^{(1)} \approx 0.42$ for $j \notin \{3, 4\}$ and $\omega_3^{(1)} \approx \omega_4^{(1)} \approx 0.5$. In the general case, we have

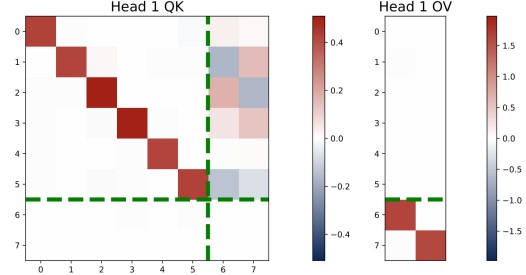

$$\omega_{\mathcal{S}^*}^{(1)} = \breve{\omega}^* \cdot \mathbf{1}_{|\mathcal{S}^*|}, \quad \omega_{\mathcal{S}^c}^{(1)} = \breve{\omega}^c \cdot \mathbf{1}_{|\mathcal{S}^c|}, \quad \mu^{(1)} = \breve{\mu} \cdot \mathbf{1}_N,$$

where $\breve{\omega}^*$, $\breve{\omega}^c$, and $\breve{\mu}$ are three scaling factors.

*Figure 15.* Heatmap of KQ and OV matrices trained on multiple tasks with $H = 1$. The trained attention model learns to implement a single weighted kernel regressor.

This indicates that, the trained single-attention model learns a weighted kernel regressor. Specifically, for each task $n \in [N]$, the predictor is

$$\widehat{y}_{q,n} = \breve{\mu} \cdot \sum_{\ell=1}^{L} \frac{\exp(\breve{\omega}^* \cdot \langle x_{\ell,\mathcal{S}^*}, x_{q,\mathcal{S}^*} \rangle + \breve{\omega}^c \cdot \langle x_{\ell,\mathcal{S}^c}, x_{q,\mathcal{S}^c} \rangle) \cdot y_{\ell,n}}{\sum_{\ell=1}^{L} \exp(\breve{\omega}^* \cdot \langle x_{\ell,\mathcal{S}^*}, x_{q,\mathcal{S}^*} \rangle + \breve{\omega}^c \cdot \langle x_{\ell,\mathcal{S}^c}, x_{q,\mathcal{S}^c} \rangle)},$$

which applies uses a same set of weights to aggregate responses for all tasks.

**Mode II ($H = 2$): Single Weighted Pre-Conditioned GD Predictor.** As shown in Figure 16, the two-head attention learns an opposing positive-negative pattern which is nearly identical to the single-task we observed in §3. However, similar to Mode I, the attention model assigns more weight on the informative entries $\mathcal{S}^*$ ($\sim 0.13$ vs. $\sim 0.12$). In addition, the learned scale is much smaller than that in the previous single-head case, which is also observed in the single-task setting. The pattern learned by the attention model can be summarized as below:

$$\omega_{\mathcal{S}^*}^{(1)} = -\omega_{\mathcal{S}^*}^{(2)} = \breve{\omega}^* \cdot \mathbf{1}_{|\mathcal{S}_*|}, \quad \omega_{\mathcal{S}^c}^{(1)} = -\omega_{\mathcal{S}^c}^{(2)} = \breve{\omega}^c \cdot \mathbf{1}_{|\mathcal{S}^c|}, \quad \mu^{(1)} = -\mu^{(2)} = \breve{\mu} \cdot \mathbf{1}_N,$$

where $\breve{\omega}^*$, $\breve{\omega}^c$, and $\breve{\mu}$ are three scaling factors. Therefore, when $H = 2$, for each task $n \in [N]$, the predictor approximately takes the following form:

$$\widehat{y}_{q,n} \approx \frac{2\breve{\mu}\breve{\omega}^*}{L} \cdot \sum_{\ell=1}^{L} \langle \bar{x}_{\ell,\mathcal{S}^*}, x_{q,\mathcal{S}^*} \rangle \cdot y_{\ell,n} + \frac{2\breve{\mu}\breve{\omega}^c}{L} \cdot \sum_{\ell=1}^{L} \langle \bar{x}_{\ell,\mathcal{S}^c}, x_{q,\mathcal{S}^c} \rangle \cdot y_{\ell,n}.$$

Here we use the fact that $\breve{\omega}$ is small. Thus, the trained attention model solves the multi-task linear regression using weighted debiased GD predictor, which assigns a slightly larger weight to $\mathcal{S}^*$.

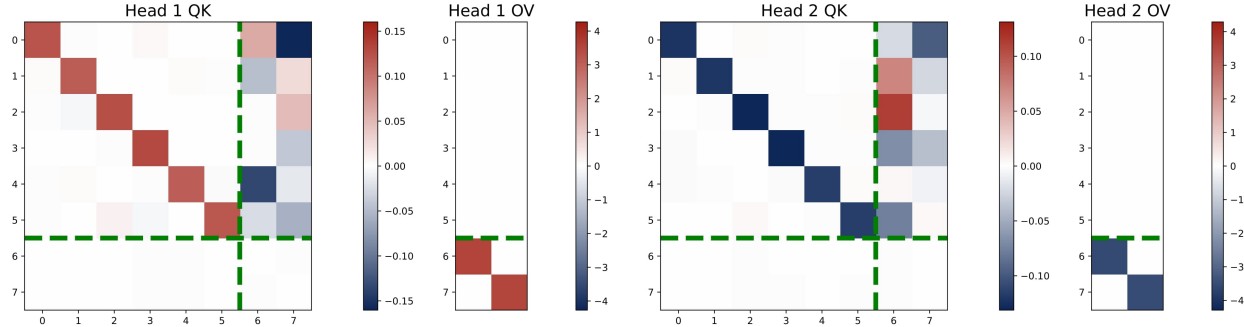

*Figure 16.* Heatmap of KQ and OV matrices trained on multi-task problem with $H = 2$. In the trained model, the KQ matrices exhibits a diagonal-only pattern and the diagonal values have two magnitudes. The magnitude of $\{\omega_j^{(h)}\}_{j \in \{3,4\}}$ is slightly larger than the rests of the diagonal entries. Besides, $\mu^{(1)} = -\mu^{(2)}$ and they are proportional to an all-one vector.

**Mode III** ($H \geq 2N$)**:** $N$ **Independent Debiased GD Predictors.** Recall for a single task, at least two heads are required to implement a debiased GD predictor. Here, we explore a regime where $H \geq 2N$. Thus, the transformer has the expressive power to use two attention heads to solve each task independently, via implementing debiased GD predictors

As shown in Figure 17, the experiment result on a four-head model supports this hypothesis. In the trained model, Heads 1 and 2 are coupled to solve Task 1, focusing exclusively on $\mathcal{S}_1$ in KQ while contributing only to the prediction of $\widehat{y}_{q,1}$. This can be seen by noting that $\omega_{\mathcal{S}_1}^{(1)} = -\omega_{\mathcal{S}_1}^{(2)} = \breve{\omega} \cdot \mathbf{1}_{|\mathcal{S}_1|}$, $\omega_{\mathcal{S}_1^c}^{(1)} = -\omega_{\mathcal{S}_1^c}^{(2)} = \mathbf{0}_{|\mathcal{S}_1^c|}$, $\mu_1^{(1)} = -\mu_1^{(2)} = \breve{\mu}$, and $\mu_2^{(1)} = \mu_2^{(2)} = 0$. Here $\breve{\omega}$ and $\breve{\mu}$ are two positive scaling parameters and $\mu_n^{(h)}$ denotes the $n$-th entry of $\mu^{(h)}$. Similarly, Heads 3 and 4 are coupled to solve Task 2, following the same structure. In the general case, consider $H = 2N$ with $(2n-1)$-th and $2n$-th heads solving the $n$-th task together. For all $n \in [N]$, the learned pattern satisfies

$$\omega_{\mathcal{S}_n}^{(2n-1)} = -\omega_{\mathcal{S}_n}^{(2n)} = \breve{\omega} \cdot \mathbf{1}_{|\mathcal{S}_n|}, \quad \omega_{\mathcal{S}_n^c}^{(2n-1)} = -\omega_{\mathcal{S}_n^c}^{(2n)} = \mathbf{0}_{|\mathcal{S}_n^c|}, \quad \mu^{(2n-1)} = -\mu^{(2n)} = \breve{\mu} \cdot \mathbf{e}_n,$$

where we use $\mathbf{e}_n \in \mathbb{R}^N$ to denote a canonical basis whose $n$-th entry is one and the rest of the entries are all zero. With this pattern, using the fact that $\breve{\omega}$ is small, for each task $n \in [N]$, the predictor approximately takes the following form:

$$\widehat{y}_{q,n} \approx \frac{2\breve{\mu}\breve{\omega}}{L} \cdot \sum_{\ell=1}^{L} \langle \bar{x}_{\ell,\mathcal{S}_n}, x_{q,\mathcal{S}_n} \rangle \cdot y_{\ell,n}, \tag{C.1}$$

where $2\breve{\mu}\breve{\omega} \approx 1$. This implements an independent GD predictor for each task $n$, using $\{x_{\ell,\mathcal{S}_n}, y_{\ell,n}\}_{\ell \in [L]}$. In other words, the model bakes the true supports $\{\mathcal{S}_n\}_{n \in [H]}$ into the learned model weights.

Note that in Mode I and Mode II, the same weights are applied across different tasks, relying on all covariate entries. In contrast, when the model has sufficient expressive power, it learns to allocate dedicated heads for each task and efficiently solve each task by using a debiased gradient descent predictor that is restricted to the true support $\mathcal{S}_n$ of the task.

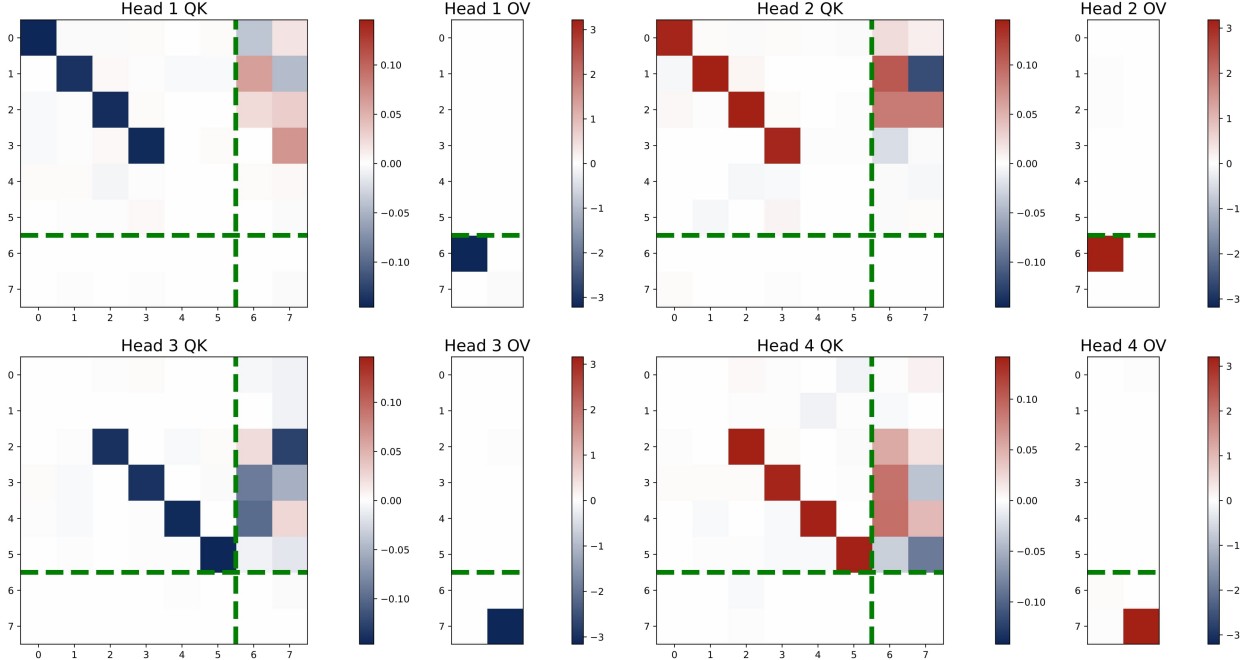

*Figure 17.* Heatmap of KQ and OV matrices trained on the multi-task problem with $H = 4$. The trained model learns to implement $N$ independent pre-conditioned GD predictors by assigning each task to a unique pair of heads, ensuring that each head is exclusively used for a single task. Task-relevant entry information is encoded in the KQ circuit and OV controls the output task.

**Mode IV** ($2 < H < 2N$)**: Pre-Conditioned GD Predictors with Superposition.** Unlike Mode I and Mode II, where attention employs shared weights across all tasks, or Mode III, where each task is assigned distinct, non-overlapping heads,

when $2 < H < 2N$, we observe an intriguing superposition phenomenon. Intuitively, superposition requires requires some heads to solve move than one task simultaneously, which means that we may observe $\omega^{(h)}$ that have both positive and negative values. Superposition enables the transformer model to approximate the performance of the $N$ debiased GD predictors despite the limited head capacity.

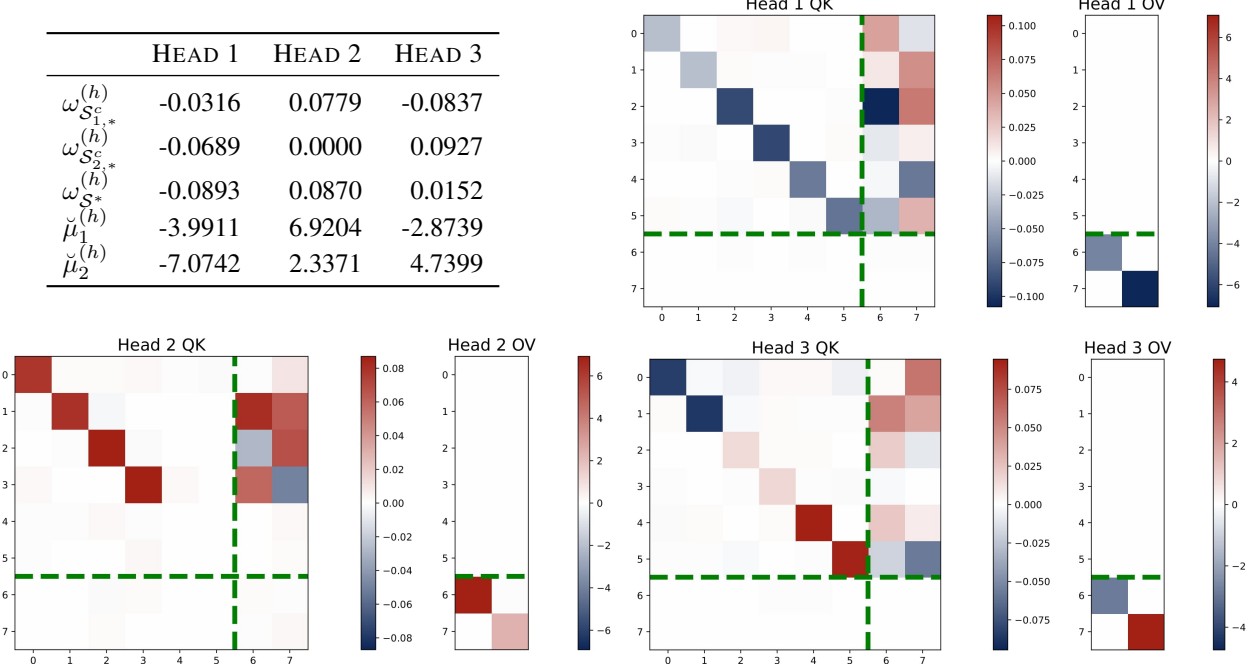

| | HEAD 1 | HEAD 2 | HEAD 3 |
|---|---|---|---|
| $\omega_{\mathcal{S}_{1,*}^c}^{(h)}$ | -0.0316 | 0.0779 | -0.0837 |
| $\omega_{\mathcal{S}_{2,*}^c}^{(h)}$ | -0.0689 | 0.0000 | 0.0927 |
| $\omega_{\mathcal{S}^*}^{(h)}$ | -0.0893 | 0.0870 | 0.0152 |
| $\breve{\mu}_1^{(h)}$ | -3.9911 | 6.9204 | -2.8739 |
| $\breve{\mu}_2^{(h)}$ | -7.0742 | 2.3371 | 4.7399 |

*Figure 18.* (Top left) Learned parameters of three-head attention model trained on multi-task problem. By direct calculation, we can show that the learned parameter follow the patterns described in (C.5) and (C.6), indicating that the model effectively emulates the pre-conditioned GD predictor despite its limited expressive power via superposition. (Remaining) Heatmap of KQ and OV matrices trained on multi-task problem with $H = 3$. In this case, each head contributes to both tasks, and in particular, Head 3 is a positive head for Task 2 and a negative head for Task 1. The trained model learns to implement the $N$ independent debiased GD predictors with superposition.

Figure 18 shows the learned pattern of a three-head model, trained on the two-task in-context linear regression problem. We have the following observations:

(i) All the three heads contribute to solving both regression tasks. This can be seen from the fact that $\mu^{(h)}$ does not have zero entries for all $h$.

(ii) Head 1 is a negative head and Head 2 is a positive head, $\mu^{(1)}, \omega^{(1)}$ are negative, while $\mu^{(2)}, \omega^{(2)}$ are positive.

(iii) Head 3 has both positive and negative components, illustrating the superposition phenomenon. Head 3 is a negative head for Task 1 and a positive head for Task 2. The first two entries of $\omega^{(3)}$ are negative, the last two entries are positive, and the middle two entries are close to $0$.

(iv) Head 1 put more effort in solving Task 2, as $\mu_2^{(1)}$ has a larger magnitude than $\mu_1^{(1)}$. Similarly, Head 2 focus more on Task 1. Head 3 is rather more balanced.

These observations lead to a mathematical depiction of $\{\omega^{(h)}, \mu^{(h)}\}_{h \in [H]}$ as follows. Let $\mathscr{D}$ denote the set of all possible grouped divisions $\{\mathcal{S}^*, \mathcal{S}_{1,*}^c, \mathcal{S}_{2,*}^c\}$ with $\mathcal{S}_{n,*}^c = \mathcal{S}_n \backslash \mathcal{S}^*$ for $n \in \{1, 2\}$. Under our setup, we have $\mathcal{S}_{1,*}^c = \{1, 2\}$, $\mathcal{S}_{2,*}^c = \{5, 6\}$, and $\mathcal{S}^* = \{3, 4\}$. The parameters of the three-head model can be summarized as follows:

$$\sum_{h=1}^{H} \mu^{(h)} = \mathbf{0}_N, \quad \omega_{\mathcal{S}}^{(h)} = \breve{\omega}_{\mathcal{S}}^{(h)} \cdot \mathbf{1}_{|\mathcal{S}|}, \quad \forall (h, \mathcal{S}) \in [H] \times \mathscr{D}.$$

In other words, the supports of the two tasks split $[d]$ into non-overlapping groups, and the value of the KQ parameters are the same within each group. Here, $\omega_{\mathcal{S}}^{(h)}$ denotes the subvector of $\omega^{(h)}$ with entries in $\mathcal{S}$ and $\breve{\omega}_{\mathcal{S}}^{(h)}$ is a scaling factor. Moreover, the OV parameters sum to a zero vector, generalizes single-task setting. Let $\breve{\mu}^{(h)}$ denote the learned value of $\mu^{(h)}$. Then Head $h$ contributes to the prediction of Task $n$ via the parameter $\breve{\mu}_n^{(h)}$, and we have $\sum_{h=1}^{H} \breve{\mu}_n^{(h)} = 0$ for all $n$.

By plugging the values of $\{\omega^{(h)}, \mu^{(h)}\}_{h \in [H]}$ into the transformer model, for each task $n \in [N]$, its output can be written as

$$\widehat{y}_{q,n} = \sum_{h=1}^{H} \breve{\mu}_n^{(h)} \cdot \sum_{\ell=1}^{L} \frac{\exp(\langle x_\ell, \breve{\omega}^{(h)} \odot x_q \rangle) \cdot y_{\ell,n}}{\sum_{\ell=1}^{L} \exp(\langle x_\ell, \breve{\omega}^{(h)} \odot x_q \rangle)}, \tag{C.2}$$

where $\omega^{(h)} \odot x_q$ denotes the Hadamard product. By the construction of $\mathscr{D}$, we have

$$\langle x_\ell, \breve{\omega}^{(h)} \odot x_q \rangle = \sum_{\mathcal{S} \in \mathscr{D}} \breve{\omega}_{\mathcal{S}}^{(h)} \cdot \langle x_{\ell,\mathcal{S}}, x_{q,\mathcal{S}} \rangle. \tag{C.3}$$

When $L$ is sufficiently large, we combine (C.3) and apply Taylor expansion in (C.2) to obtain that

$$\begin{aligned} \widehat{y}_{q,n} &\approx \sum_{h=1}^{H} \breve{\mu}_n^{(h)} \cdot \sum_{\ell=1}^{L} \frac{y_{\ell,n}}{L} \cdot \left(1 + \sum_{\mathcal{S} \in \mathscr{D}} \breve{\omega}_{\mathcal{S}}^{(h)} \cdot \langle \bar{x}_{\ell,\mathcal{S}}, x_{q,\mathcal{S}} \rangle \right) \\ &= \sum_{h=1}^{H} \sum_{\mathcal{S} \in \mathscr{D}} \frac{\breve{\mu}_n^{(h)} \breve{\omega}_{\mathcal{S}}^{(h)}}{L} \cdot \sum_{\ell=1}^{L} \langle \bar{x}_{\ell,\mathcal{S}}, x_{q,\mathcal{S}} \rangle \cdot y_{\ell,n}, \end{aligned} \tag{C.4}$$

the last equality follows from the fact that $\sum_{h=1}^{H} \breve{\mu}_n^{(h)} = 0$. That is, $\widehat{y}_{q,n}$ can be viewed as a sum of $|\mathscr{D}|$ independent debiased GD predictors, each of which is restricted to the support $\mathcal{S}$.

Furthermore, we carefully examine the learned values of $\{\omega^{(h)}, \mu^{(h)}\}_{h \in [H]}$, which are listed in Table 18. We observe that the following conditions hold:

$$\sum_{h=1}^{H} \breve{\mu}_1^{(h)} \breve{\omega}_{\mathcal{S}_{1,*}^c}^{(h)} = \sum_{h=1}^{H} \breve{\mu}_1^{(h)} \breve{\omega}_{\mathcal{S}^*}^{(h)} \approx 1, \qquad \sum_{h=1}^{H} \breve{\mu}_1^{(h)} \breve{\omega}_{\mathcal{S}_{2,*}^c}^{(h)} = 0, \tag{C.5}$$

$$\sum_{h=1}^{H} \breve{\mu}_2^{(h)} \breve{\omega}_{\mathcal{S}_{2,*}^c}^{(h)} = \sum_{h=1}^{H} \breve{\mu}_2^{(h)} \breve{\omega}_{\mathcal{S}^*}^{(h)} \approx 1, \qquad \sum_{h=1}^{H} \breve{\mu}_2^{(h)} \breve{\omega}_{\mathcal{S}_{1,*}^c}^{(h)} = 0. \tag{C.6}$$

As a result, $\widehat{y}_{q,n}$ in (C.4) can be simplified as

$$\widehat{y}_{q,n} \approx \sum_{h=1}^{H} \frac{1}{L} \cdot \sum_{\ell=1}^{L} \langle \bar{x}_{\ell,\mathcal{S}_n}, x_{q,\mathcal{S}_n} \rangle \cdot y_{\ell,n}, \qquad \forall n \in [N].$$

This estimator is identical to that in (C.1). This means that, when $H = 3$ and $N = 2$, the trained transformer model learns the same estimator – sum of two independent debiased GD estimator – as the case when $H = 2N$. But here the transformer model achieves by exploiting the superposition. That is, each head is used to solve both tasks, and the aggregated effect is that the model can approximate the performance of the $N$ independent debiased GD predictors. Here we only study a toy case with $H = 3$ and $N = 2$. We believe that the superposition phenomenon is a general phenomenon that occurs when $2 < H < 2N$. But more a larger $N$, the superposition phenomenon becomes more complex and entangled. For the general multi-task setting, characterizing the training dynamics and solution manifold of the multi-head attention with superposition is an important direction for future work.

## C.4. Experimental Results for Loss Approximation

In the following, we examine the difference of the true population loss of ICL and the approximate loss in (4.1) in terms of the training dynamics.

**Intuitions of Approximation and Scaling in Proposition E.1.** In the proof of loss approximation, the main rationale is that the high-order terms are negligible when the prompt length $L$ is sufficiently large under certain conditions. For a fixed $d$, the intuition is that $p = \mathtt{smax}(\omega \cdot X x_q)$ should be distributed relatively "uniformly" across the entries, resulting in a quite small norm $\|p\|_2^k$ for $k \geq 2$. As shown in Figure 19a, with a fixed $d$, the norm concentrates around $0$ as the length $L$ increases. However, this property may not hold in the high-dimensional proportional regime, i.e., $d/L \to \xi$ (see Figure 19b). To address this issue, we impose scaling conditions $\|\omega\|_\infty^2 \lesssim \log L/d$ to offset the effects of high dimensionality and ensure that the norm remains small (see Figure 19c).

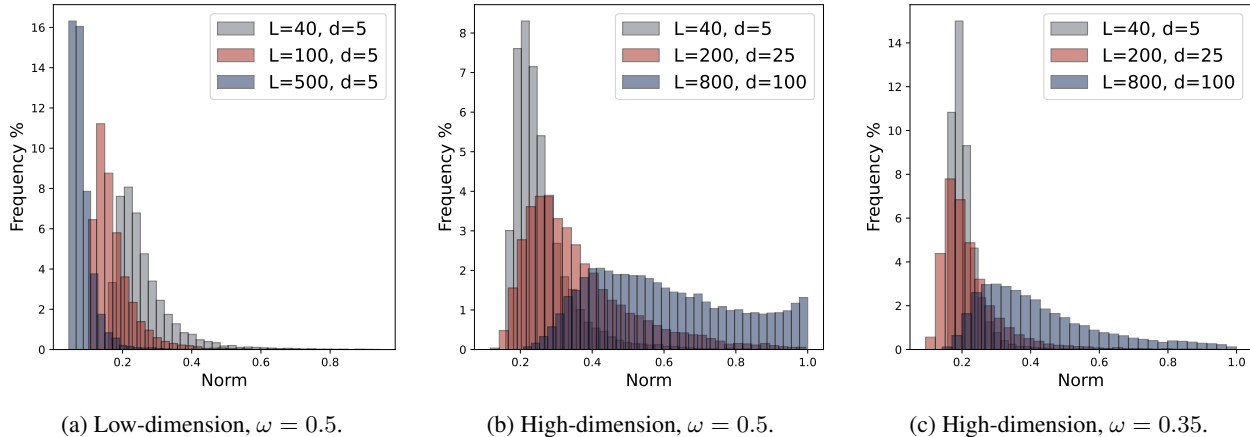

(a) Low-dimension, $\omega = 0.5$.      (b) High-dimension, $\omega = 0.5$.      (c) High-dimension, $\omega = 0.35$.

*Figure 19.* Distribution of norm $\|p\|_2^2$. Figure (a) depicts the low-dimensional regime with a fixed $d = 5$. As the sequence length $L$ increases, the distribution of $\|p\|_2^2$ converges to singleton at zero. Figure (b) and (c) illustrate the proportional regime $d/L = 1/8$. As $L$ increases and the scaling factor $\omega$ decreases, the norm $\|p\|_2^2$ becomes increasingly concentrated around zero.

**Training Dynamics of True Loss and Approximate Loss.** To further evaluate the performance of approximate loss in (4.1), we compare the training dynamics given by these two different losses. Consider the four-head attention model with $d = 5$, $L = 40$, $\sigma^2 = 0.1$. By comparing Figures 20a and 20b, we observe that the approximate loss effectively captures the evolution of parameters seen in the full model. Additionally, we note that the actual training dynamics appear less "regular" due to the finite batch size (as opposed to the population risk used in the approximate loss) and internal perturbations arising from parameters at non-target positions in the full model.

## D. Analysis on Training Dynamics of Two-head Attention

In this section, we offer a more comprehensive analysis of the gradient flow on the approximate loss defined in (4.1) with a two-head attention model. This analysis provides justification for the evolution of the parameters $\{\omega^{(h)}, \mu^{(h)}\}_{h \in [2]}$ in Figure 1b. For simplicity, we consider the gradient flow of $\widetilde{\mathcal{L}}/2$ with a two-head attention, where $\widetilde{\mathcal{L}}$ is the approximate loss function. We let $\theta$ denote the vector of all parameters, i.e., $\theta = (\mu, \omega) \in \mathbb{R}^4$, where $\mu = (\mu^{(1)}, \mu^{(2)}) \in \mathbb{R}^2$ and $\omega = (\omega^{(1)}, \omega^{(2)}) \in \mathbb{R}^2$. Consider the gradient flow algorithm. We let $\theta_t$ denote the parameter at time $t$. The gradient flow dynamics is characterized by an ordinary differential equation (ODE):

$$\partial_t \theta_t = -\nabla_\theta \mathcal{L}(\theta_t), \qquad \text{where } \mathcal{L}(\theta) := \widetilde{\mathcal{L}}(\theta)/2.$$

Here we slightly abuse the notation by writing $\widetilde{\mathcal{L}}/2$ as $\mathcal{L}$, which is the case only in this section. For simplification, we consider the following initialization.

**Definition D.1** (Initialization). *Consider a two-head attention model with symmetric initialization such that $\mu = \omega = (\alpha, -\alpha)$ where $\alpha > 0$ is a sufficiently small constant defining the initial scale.*

Under initialization in Definition D.1, gradient flow starts from a point satisfying sign-matching and homogeneous scaling (see §4.2). In the following, we show that these patterns, once emerged, are preserved along the gradient flow trajectory. This aligns with the experimental results shown in Figure 1b, where we train the full parameters of the attention model from random initialization.

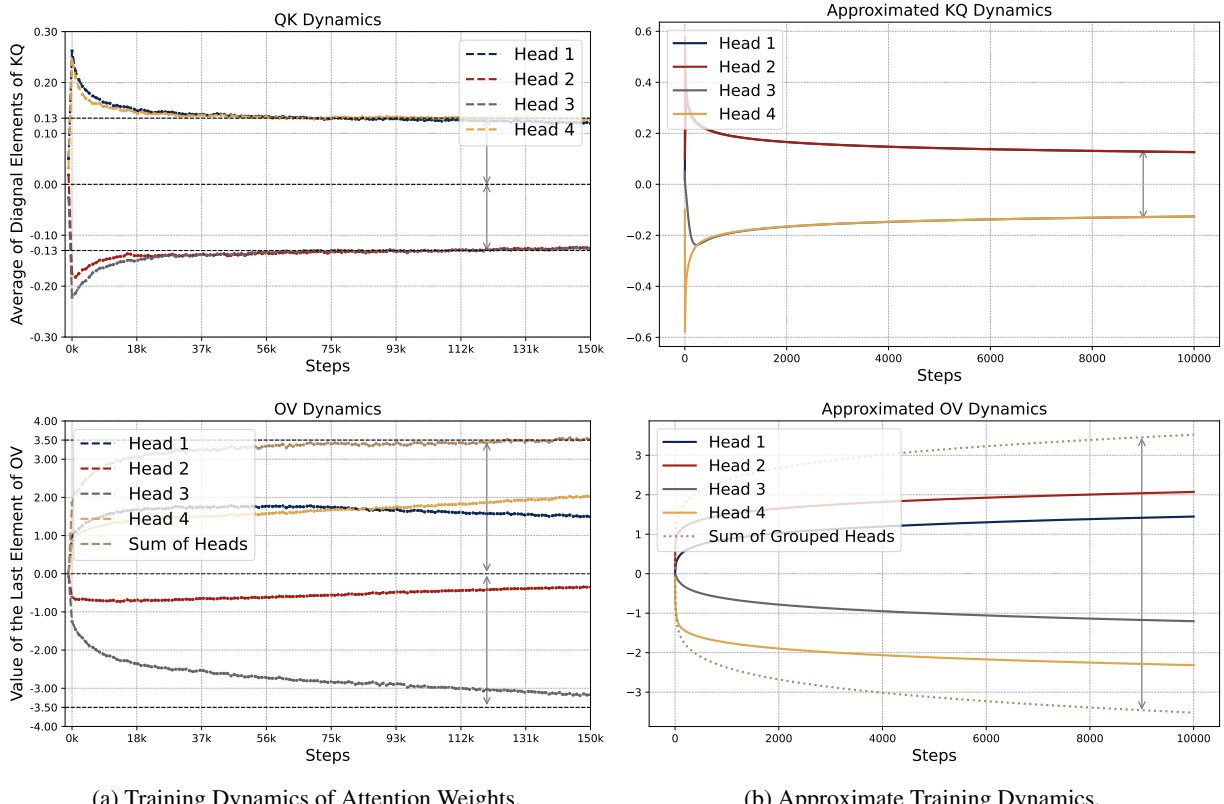

(a) Training Dynamics of Attention Weights.

(b) Approximate Training Dynamics.

*Figure 20.* Comparison of training dynamics between the true loss and the approximate loss for four-head attention model. The approximate training dynamics closely capture the behavior of the true dynamics, providing an effective characterization of the whole training process.

**Preservation of Symmetry and Homogeneous Scaling.** Based on gradient calculations and the definition of gradient flow, the time derivative of parameters satisfies that

$$\partial_t \mu_t = \omega_t - \mu_t^\top \omega_t \cdot \omega_t - \lambda \cdot \exp(d\omega_t \omega_t^\top) \cdot \mu_t,$$
$$\partial_t \omega_t = \mu_t - \mu_t^\top \omega_t \cdot \mu_t - d\lambda \cdot \left( \exp(d\omega_t \omega_t^\top) \odot (\mu_t \mu_t^\top) \right) \cdot \omega_t,$$

where we define $\lambda = (1 + \sigma^2)/L$ for notational simplicity. Let $s_t = \exp(d \cdot (\omega_t^{(1)})^2) - \exp(d \cdot (\omega_t^{(2)})^2)$ and $u_t = (\langle \mu_t, \mathbf{1} \rangle, \langle \omega_t, \mathbf{1} \rangle, s_t) \in \mathbb{R}^3$. By direct calculation and careful decomposition, we have

$$
\begin{aligned}
\partial_t \langle \mu_t, \mathbf{1} \rangle &= (1 - \mu_t^\top \omega_t) \cdot \langle \omega_t, \mathbf{1} \rangle - \lambda \cdot \exp(d \cdot \omega_t^{(1)} \omega_t^{(2)}) \cdot \langle \mu_t, \mathbf{1} \rangle \\
&\quad + \lambda \cdot \exp(d \cdot (\omega_t^{(1)})^2) \cdot \langle \mu_t, \mathbf{1} \rangle + \lambda \cdot \mu_t^{(2)} \cdot s_t := \langle g_{1,t}, u_t \rangle,
\end{aligned}
\tag{D.1}
$$

$$
\begin{aligned}
\partial_t \langle \omega_t, \mathbf{1} \rangle &= (1 - \mu_t^\top \omega_t) \cdot \langle \mu_t, \mathbf{1} \rangle - d\lambda \cdot \mu_t^{(1)} \mu_t^{(2)} \cdot \exp(d \cdot \omega_t^{(1)} \omega_t^{(2)}) \cdot \langle \omega_t, \mathbf{1} \rangle \\
&\quad - d\lambda \cdot \omega_t^{(1)} \cdot (\mu_t^{(1)})^2 \cdot s_t - d\lambda \cdot (\mu_t^{(1)})^2 \cdot \exp(d \cdot (\omega_t^{(2)})^2) \cdot \langle \omega_t, \mathbf{1} \rangle \\
&\quad - d\lambda \cdot \omega_t^{(2)} \cdot \exp(d \cdot (\omega_t^{(2)})^2) \cdot (\mu_t^{(2)} - \mu_t^{(1)}) \cdot \langle \mu_t, \mathbf{1} \rangle := \langle g_{2,t}, u_t \rangle.
\end{aligned}
\tag{D.2}
$$

Here we observe that the time-derivatives of both $\langle \mu_t, \mathbf{1} \rangle$ and $\langle \omega_t, \mathbf{1} \rangle$ are linear functions of $u_t$, and we let $g_{1,t}$ and $g_{2,t}$ denote the linear coefficients, respectively. Furthermore, it is easy to see that

$$
\begin{aligned}
\partial_t s_t &= 2d \cdot \exp\left(d \cdot (\omega_t^{(1)})^2\right) \cdot \omega_t^{(1)} \cdot \partial_t \omega_t^{(1)} - 2d \cdot \exp\left(d \cdot (\omega_t^{(2)})^2\right) \cdot \omega_t^{(2)} \cdot \partial_t \omega_t^{(2)} \\
&= 2d \cdot \omega_t^{(1)} \cdot \partial_t \omega_t^{(1)} \cdot s_t + 2d \cdot \exp(d \cdot (\omega_t^{(2)})^2) \cdot \partial_t \omega_t^{(1)} \cdot \langle \omega_t, \mathbf{1} \rangle \\
&\quad - 2d \cdot \exp(d \cdot (\omega_t^{(2)})^2) \cdot \omega_t^{(2)} \cdot \partial_t \langle \omega_t, \mathbf{1} \rangle := \langle g_{3,t}, u_t \rangle.
\end{aligned}
\tag{D.3}
$$

In other words, the time-derivative of $s_t$ is also a linear function of $u_t$, with coefficient given by $g_{3,t}$. Here the last equality uses the fact that we can write $\partial_t \langle \omega_t, \mathbf{1} \rangle = \langle g_{2,t}, u_t \rangle$ as shown in (D.2). We note that $g_{1,t}$, $g_{2,t}$, and $g_{3,t}$ are functions of $u_t$.

To show that $\mu^{(1)} = -\mu^{(2)}$ and $\omega^{(1)} = -\omega^{(2)}$ are preserved along the gradient flow, note that based on (D.1), (D.2) and (D.3), we can characterize the dynamics of $u_t$ with a matrix ODE $\partial_t u_t = G_t u_t$ where $G_t = [g_{1,t}, g_{2,t}, g_{3,t}]^\top \in \mathbb{R}^{3\times 3}$. By solving the ODE, we have $u_t = \varpi \cdot \exp\left(\int_0^t G_s \mathrm{d}s\right)$, where $\varpi$ is a constant vector that reflects the initial condition, i.e., $u_0 = \varpi$. Note that the initialization in Definition D.1 indicates that $u_0 = \varpi = 0$. Therefore, $u_t = 0$ at any time $t$. In particular, this implies that $\mu_t^{(1)} + \mu_t^{(2)} = \omega_t^{(1)} + \omega_t^{(2)} = 0$. See Figure 21a for an illustration, which plot the full evolution of $\mu$ and $\omega$.

Hence, we can track the dynamics of $(\mu_t, \omega_t)$ by only focusing on $\mu_t^{(1)}$ and $\omega_t^{(1)}$. For notational clarity, we let $\varphi_t$ and $\varrho_t$ denote $\mu_t^{(1)}$ and $\omega_t^{(1)}$, respectively. Then the evolution of $(\varphi_t, \varrho_t)$ is given by the following ODE system:

$$\partial_t \varphi_t = \varrho_t - 2\varphi_t \varrho_t^2 - \lambda \varphi_t \cdot \left(\exp(d\varrho_t^2) - \exp(-d\varrho_t^2)\right),$$
$$\partial_t \varrho_t = \varphi_t - 2\varphi_t^2 \varrho_t - d\lambda \varphi_t^2 \varrho_t \cdot \left(\exp(d\varrho_t^2) + \exp(-d\varrho_t^2)\right). \tag{D.4}$$

Such an ODE system does not admit a closed-form solution, but can be solved numerically. To gain some insight, in Figures 21a and 21b, we plot the full dynamics of $\mu$ and $\omega$ with $L = 40$, $d = 5$ and $\sigma = 0.1$, as well as the dynamics in the early stage of gradient flow. In addition, in Figure 21c we plot the evolution of the loss $\widetilde{L}(\theta_t)$ as a function of $t$, as well as the ratio $\varphi_t / \varrho_t$. In Figure 21d, we plot the dynamics of $\varphi_t$, together with $\varphi_t \cdot \varrho_t$ and $\varphi^*(\varrho_t)$. Here $\varphi^*(\varrho_t)$ denotes the optimal value of $\varphi_t$ that minimizes the approximate loss function, when $\varrho_t$ is fixed. See (D.17) for its definition. In addition, we additionally plot the dynamics of $\mu$ and $\omega$ in the case with $d = 10$ in Figure 21. We observe almost identical behaviors as in the case with $d = 5$.

By examining these figures, we observe that the evolution of the ODE system in (D.4) exhibits the following three phases:

- **Phase I (Exponential and Synchronous Growth).** From a small initialization, both $\varphi_t$ and $\varrho_t$ grow exponentially fast in $t$ at nearly the same rate. This leads to a rapid reduction in the loss $\widetilde{\mathcal{L}}$. Moreover, during this stage, $\varphi_t$ stays roughly equal to $\varrho_t$, as shown in Figure 21b. In particular, their ratio $\varphi_t / \varrho_t$ stays close to one, which is shown in Figure 21c.

- **Phase II (Slowed Growth and Peak Formation).** As $\varphi_t$ and $\varrho_t$ increase, their growth rate in time decreases, i.e., the exponential growth in both $\varphi_t$ and $\varrho_t$ stops. Moreover, $\varrho_t$ eventually reaches its maximum value $\widetilde{O}(d^{-1/2})$. Although the parameters no longer grow exponentially, the loss decreases sharply until $\varrho_t$ begins to decline. Moreover, the ratio $\varphi_t / \varrho_t$ does not grow much from one. This phase ends when $\varrho_t$ attains its maximum value, which corresponds a critical of the ODE in (D.4), under the condition that $\varphi_t \approx \varrho_t$. As shown in Figure 21b, by the end of this phase, $\varphi_t$ and $\varrho_t$ part ways, with $\varphi_t$ keeping increasing while $\varrho_t$ begins to decrease. Throughout this phase, the ratio $\varphi_t / \varrho_t$ does not increase much from one, as shown in Figure 21c. In addition to the sharp drop of the loss, the product $\varphi_t \cdot \varrho_t$ increases rapidly, as shown in Figure 21d.

- **Phase III (Convergence).** After $\varrho_t$ reaches its peak, it decreases to zero while $\varphi_t$ continues increasing. Their ratio $\varphi_t / \varrho_t$ thus keeps increasing. In addition, the loss converges to the minimum value and the value of the product $\varphi_t \cdot \varrho_t$ increases and converges. Moreover, the last phase exhibits an interesting phenomenon: the difference between $\varphi_t$ and $\varphi^*(\varrho_t)$ gradually becomes negligible. This motivates us to use $\varphi^*(\varrho_t)$ as a surrogate of $\varphi_t$ for the analysis of the limiting behavior of the ODE system.

In the following, we analyze the ODE system defined in (D.4) under the high-dimensional regime where $d/L \to \xi$ and $d, L \to \infty$. Recall that we define $\lambda = (1 + \sigma^2)/L$, and thus $d\lambda \to (1 + \sigma^2) \cdot \xi$. Here we additionally assume that $\xi$ is a small constant such that the limit of $d\lambda$ is less than one.

**Phase I: Exponential and Synchronous Growth.** At the beginning of gradient flow, both $\varphi_t$ and $\varrho_t$ quickly escape from the starting point, maintaining an almost identical growth rate while keeping their magnitude small. During the first stage, $\varphi_t \approx \varrho_t$ holds due to the same small initialization (see Figure 21b for an illustration). In this case, we have $\exp(d\varrho_t^2) - \exp(-d\varrho_t^2) = O(\varrho_t^2)$ and $\exp(d\varrho_t^2) + \exp(-d\varrho_t^2) \approx 2$. Ignoring the high-order terms of $\varphi_t$ and $\varrho_t$, we can approximate (D.4) by a much simpler system of differential equations:

$$\partial_t \varphi_t \approx \varrho_t, \qquad \partial_t \varrho_t \approx \varrho_t, \tag{D.5}$$

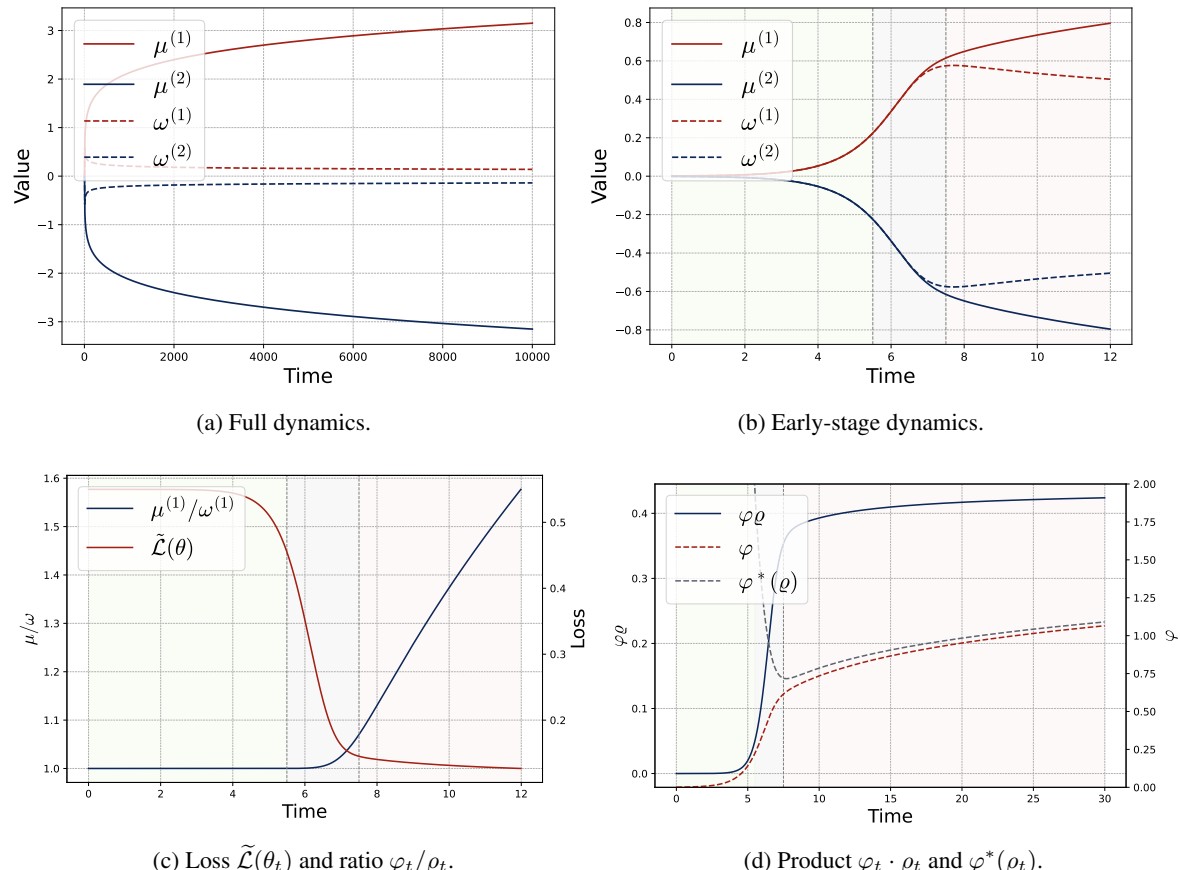

(a) Full dynamics.

(b) Early-stage dynamics.

(c) Loss $\widetilde{\mathcal{L}}(\theta_t)$ and ratio $\varphi_t/\varrho_t$.

(d) Product $\varphi_t \cdot \varrho_t$ and $\varphi^*(\varrho_t)$.

*Figure 21.* Gradient flow with respect to the approximated loss $\widetilde{\mathcal{L}}/2$ with $L = 40$, $d = 5$, $H = 2$ and $\sigma^2 = 0.1$ under initialization in Definition D.1 with initialization $\alpha = 0.001$. Figure (a) shows the full evolution of the four-dimensional dynamic system in terms of $\mu$ and $\omega$ along the gradient flow. Figure (b) provides a closer view of the dynamics during the early stage, with areas shaded in different colors highlighting the three phases. Specifically, $\varrho_t$ first reaches its maximum ($\sim 0.57$) which is on the order of $O(\sqrt{\log d/d})$ ($\sim 0.56$) and $\varphi_t$ keeps increasing. Figure (c) and (d) present the evolution of loss $\widetilde{\mathcal{L}}(\theta_t)$, $\varphi_t/\varrho_t$, $\varphi_t\varrho_t$, and $\varphi^*(\varrho_t)$ to track the relative behavior of parameters. Here $\varphi^*(\varrho_t)$ is defined in (D.17).

whose solution is given by

$$\varrho_t \approx \alpha \exp(t), \qquad \varphi_t \approx \alpha \int_0^t \varrho_s \mathrm{d}s \approx \alpha \cdot \exp(t). \tag{D.6}$$

Thus, both $\varphi_t$ and $\varrho_t$ grows exponentially in $t$.

Moreover, despite the exponential growth, the ratio between $\varphi_t$ and $\varrho_t$ remains at the initial value, i.e., $\varphi_0/\varrho_0 = 1$ for a pretty long time, resulting in a synchronous growth at the initial stage. To see this, note that we have

$$\partial_t \log(\varphi_t/\varrho_t) = \partial_t \log \varphi_t - \partial_t \log \varrho_t = 1/\varphi_t \cdot \partial_t \varrho_t - 1/\varrho_t \cdot \partial_t \varrho_t$$

$$= \left( \frac{\varrho_t}{\varphi_t} - \frac{\varphi_t}{\varrho_t} \right) - 2(\varrho_t^2 - \varphi_t^2) - \lambda \cdot \left( \exp(d\varrho_t^2) - \exp(-d\varrho_t^2) \right)$$

$$+ d\lambda\varphi_t^2 \cdot \left( \exp(d\varrho_t^2) + \exp(-d\varrho_t^2) \right) \approx 2d\lambda \cdot \varphi_t^2 > 0. \tag{D.7}$$

Here the last approximation step is derived from the facts that $\varphi_t \approx \varrho_t$ and that $\exp(d\varrho_t^2) + \exp(-d\varrho_t^2) \approx 2$. As a result, the ratio $\varphi_t/\varrho_t$ increases in $t$. Moreover, recall that we define $\lambda = (1 + \sigma^2)/L$, which is small when $L$ is large. As a result, when $t$ is small, the ratio $\varphi_t/\varrho_t$ increases in $t$ at a rather slow rate. More specifically, by solving the ODE above, $\varphi_t/\varrho_t$ can

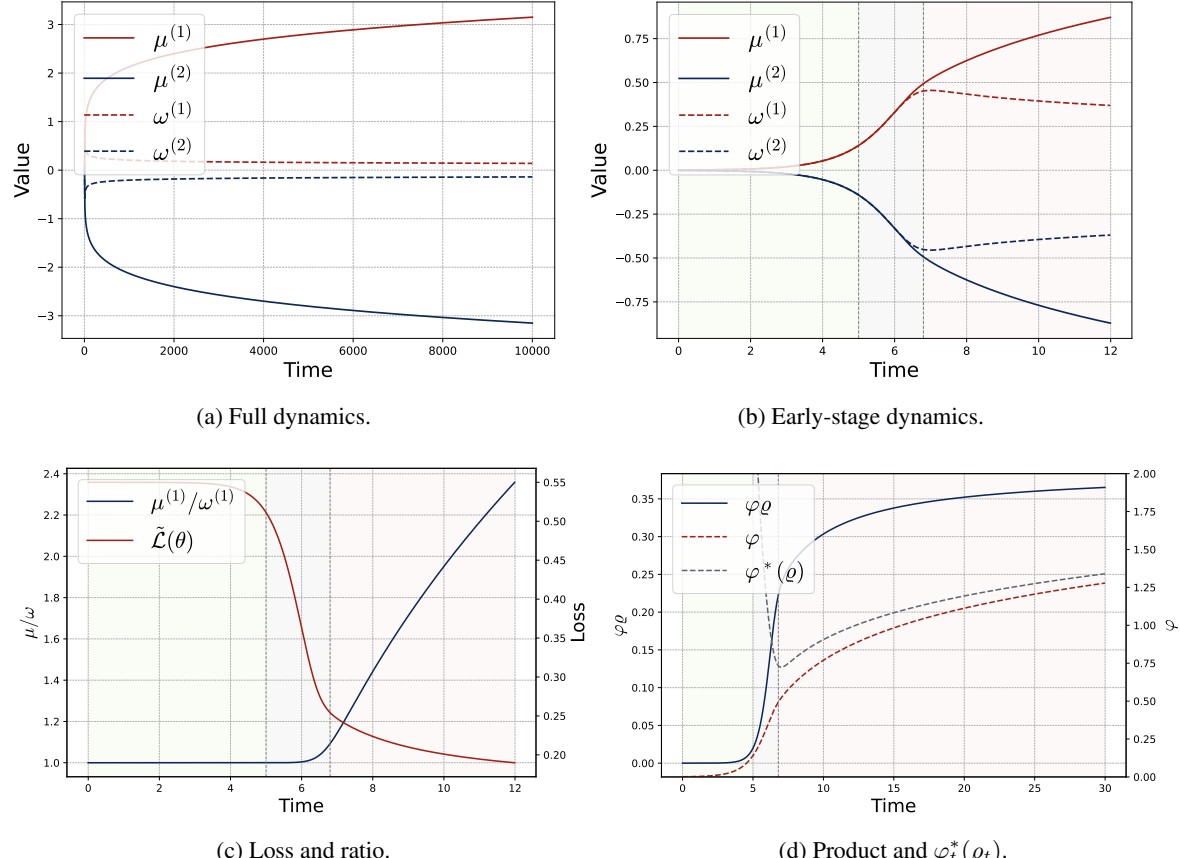

(a) Full dynamics.

(b) Early-stage dynamics.

(c) Loss and ratio.

(d) Product and $\varphi_t^*(\varrho_t)$.

*Figure 22.* Gradient flow with respect to the approximated loss $\widetilde{\mathcal{L}}/2$ with $L = 40$, $d = 10$, $H = 2$ and $\sigma^2 = 0.1$ under initialization in Definition D.1 with initialization $\alpha = 0.001$. Figure (a) shows the full evolution of the four-dimensional dynamic system in terms of $\mu$ and $\omega$ along the gradient flow. Figure (b) provides a closer view of the dynamics during the early stage, with areas shaded in different colors highlighting the three phases. Specifically, $\varrho_t$ first reaches its maximum ($\sim 0.45$) which is on the order of $O(\sqrt{\log d/d})$ ($\sim 0.47$) and $\varphi_t$ keeps increasing. Figure (c) and (d) present the evolution of loss, $\varphi_t/\varrho_t$, $\varphi_t \varrho_t$, and $\varphi_t^*$ to track the relative behavior of parameters.

be approximately written as a function of $\varrho_t$, i.e.,

$$\frac{\varphi_t}{\varrho_t} \approx \exp\left(2d\lambda \int_0^t \varphi_s^2 \mathrm{d}s\right) \approx \exp\left(2d\lambda\alpha^2 \cdot \int_0^t \exp(2s)\mathrm{d}s\right)$$
$$= \exp\left(d\lambda\alpha^2 \cdot \exp(2t)\right) \approx 1 + d\lambda\alpha^2 \cdot \exp(2t) \approx 1 + d\lambda \cdot \varrho_t^2. \tag{D.8}$$

Here, the second and the last approximation step uses (D.6), and the third approximation step uses the fact that $t$ is small. As shown in Figures 21b and 21c, in the first phase, while both $\varphi_t$ and $\varrho_t$ increase in $t$, their ratio remains substantially close to one when $\varrho_t$.

In addition, Figure 21c also shows that in the first phase, the loss started to decrease alongside the rapid growth of parameters. To show this rigorously, we compute the time-derivative of the loss function. Let $\mathcal{L}_t$ denote $\mathcal{L}(\theta_t)$, which is given by

$$\mathcal{L}_t := \mathcal{L}(\theta_t) = (1 + \sigma^2)/2 - 2\varphi_t\varrho_t + 2\varphi_t^2\varrho_t^2 + \lambda\varphi_t^2 \cdot \left(\exp(d\varrho_t^2) - \exp(-d\varrho_t^2)\right).$$

Then we have

$$\partial_t \mathcal{L}_t = -2(1 - 2\varphi_t\varrho_t) \cdot (\varphi_t \cdot \partial_t \varrho_t + \varrho_t \cdot \partial_t \varphi_t) + 2\lambda\varphi_t \cdot \partial_t \varphi_t \cdot \left(\exp(d\varrho_t^2) - \exp(-d\varrho_t^2)\right)$$
$$+ 4\lambda d\varphi_t^2 \varrho_t \cdot \partial_t \varrho_t \cdot \left(\exp(d\varrho_t^2) + \exp(-d\varrho_t^2)\right) \tag{D.9}$$
$$\approx 4\varrho_t^2 \cdot (2\varphi_t\varrho_t - 1) + 8\lambda d \cdot \varphi_t^2 \varrho_t^2 \approx -4\varrho_t^2 = -4\alpha^2 \cdot \exp(2t).$$

Here, the first approximation is based on (D.5) and the facts that $\varphi_t \approx \varrho_t$ and that $\exp(d\varrho_t^2) + \exp(-d\varrho_t^2) \approx 2$. In the second approximation, we only keep the dominating term, $-4\varrho_t^2$, and remove the high-order terms that are negligible. In the last equality, we plug in the closed-form of $\rho_t$ in (D.6). Hence, solving this differential equation, we have

$$\mathcal{L}_t \approx \mathcal{L}_0 - 4\int_0^t \alpha^2 \exp(2s)\mathrm{d}s = \mathcal{L}_0 - 2\varrho_t^2,$$

where $\mathcal{L}_0 = \mathcal{L}(\theta_0)$ is the initial value of the loss. That is, the loss decreases exponentially in $t$. However, since the magnitude of $\varrho_t$ remains small despite its exponential growth, the loss function does not decrease significantly during this phase.

In summary, during Phase I, $\varphi_t$ and $\varrho_t$ grow exponentially while keeping their ratio close to one. In addition, the change in both the loss $\mathcal{L}_t$ and the ratio $\varphi_t/\varrho_t$ is proportional to $\varrho_t^2$, indicating exponential changes in both quantities. The end of Phase I is defined by

$$\tau_1 = \max\{t \in \mathbb{R}^+ : \min\{\varphi_t, \varrho_t\} \leq d^{-1/2}/2\}, \tag{D.10}$$

when $\varphi_t$ and $\varrho_t$ is sufficiently large such that approximations for $\exp(d\varrho_t^2)$, $\exp(-d\varrho_t^2)$, and the high-order terms are no longer valid. Hence, the loss has decreased at least by $(2d)^{-1}$ and $\varphi_t/\varrho_t$ is approximately bounded by $1 + \lambda/4 \asymp 1 + 1/L$ during the first phase.

**Phase II: Slowed Growth and Peak Formation.** In the second phase, we focus on the evolution after time $t \geq \tau_1$, which is defined in (D.10). In this phase, we cannot ignore the high-order terms of $\varphi_t$ and $\varrho_t$. In particular, we use the first-order approximation $\exp(d\varrho_t^2) - \exp(-d\varrho_t^2) \approx 2d\varrho_t^2$. Moreover, We still adopt the approximation that $\varphi_t \approx \varrho_t$. To see this, note that (D.7) now becomes

$$\partial_t \log(\varphi_t/\varrho_t) \approx -2d\lambda \cdot \varrho_t^2 + d\lambda\varphi_t^2 \cdot (2 + d^2\varrho_t^4) = d\lambda \cdot \varphi_t^6. \tag{D.11}$$

Here we plug in $\varrho_t \approx \varphi_t$ and the second-order approximation that $\exp(d\varrho_t^2) + \exp(-d\varrho_t^2) \approx 2 + d^2\varrho_t^4$. Similar to (D.8), we can solve (D.11) and conclude that the ratio $\varphi_t/\varrho_t$ remains close to one, as long as $\varphi_t$ and $\varrho_t$ remains small.

Now we focus on the dynamics of $\varphi_t$ and $\varrho_t$. By (D.4), the dynamics of $\varphi_t$ can be simplified to

$$\partial_t\varphi_t \approx \varrho_t - 2\varphi_t\varrho_t^2 - \lambda\varphi_t \cdot 2d\varrho_t^2 \approx \varphi_t - 2(d\lambda + 1) \cdot \varphi_t^3, \tag{D.12}$$

where the approximations follow from $\exp(d\varrho_t^2) - \exp(-d\varrho_t^2) \approx 2d\varrho_t^2$ and $\varphi_t \approx \varrho_t$. Solving this differential equation, for all for all time $t$ during this phase ($t \geq \tau_1$), we have

$$\varphi_t \approx \{2(d\lambda + 1) + \zeta(\tau_1) \cdot \exp(-2(t - \tau_1))\}^{-1/2}, \quad \text{with } \zeta(t) = \alpha^{-2}\exp(-2t) - 2(d\lambda + 1). \tag{D.13}$$

Note that here we directly solve the differential equation $\partial_t\varphi_t = \varphi_t - 2(d\lambda + 1) \cdot \varphi_t^3$ with initialization $\varphi_{\tau_1} = \alpha \cdot \exp(\tau_1)$ in closed-form. Based on (D.13), we see that the growth rate of $\varphi_t$ is decelerated compared with the exponential growth in Phase I. But the increase in the magnitude of $\varphi_t$ is much faster thanks to the much larger initial value $\alpha \cdot \exp(\tau_1) = d^{-1/2}/2 \gg \alpha$.

Note that $\varrho_t$ shares a similar behavior since their ratio remains around one given its small growing rate. In particular, the dynamics of $\varrho_t$ can be approximately characterized by

$$\partial_t\varrho_t \approx \varphi_t - 2\varphi_t^2\varrho_t - d\lambda\varphi_t^2\varrho_t \cdot \left(2 + d^2\varrho_t^4\right) \approx \varrho_t - 2(d\lambda + 1) \cdot \varrho_t^3 - d^3\lambda \cdot \varrho_t^7, \tag{D.14}$$

where we use the second-order approximation $\exp(d\varrho_t^2) + \exp(-d\varrho_t^2) \approx 2 + d^2\varrho_t^4$ and $\varphi_t \approx \varrho_t$. Furthermore, to better understand the relative behavior between $\varphi_t$ and $\varrho_t$, we can calculate the time derivative of their difference $\varphi_t - \varrho_t$. In particular, combining (D.12) and (D.14), we have

$$\partial_t\varphi_t - \partial_t\varrho_t \approx d^3\lambda \cdot \varrho_t^7 > 0, \tag{D.15}$$

where we use the fact that $\varphi_t \approx \varrho_t$. Hence, the gap between $\varphi_t$ and $\varrho_t$ increase in $t$ and thus approximation $\varphi_t \approx \varrho_t$ would be violated by the end of this phase. Note that with the increasing of $\varrho_t$, $\partial_t\varphi_t$ and $\partial_t\varrho_t$ are quickly decreasing due to the exponential growth of $\exp(d\varrho_t^2)$ in (D.4). Based on (D.15), we can see that $\partial_t\varrho_t < \partial_t\varphi_t$ such that $\varrho_t$ will first reach the critical point, i.e., $\partial_t\varrho_t = 0$, which marks the end of Phase II. Define

$$\tau_2 = \max\{t \in \mathbb{R}^+ : \varphi_t\varrho_t \cdot (d\lambda \cdot \exp(d\varrho_t^2) + d\lambda \cdot \exp(-d\varrho_t^2) + 2) \leq 1\}.$$

To characterize the scale of $\varrho_t$ by the end of the second phase, given $\varrho_t \approx \varphi_t$, we have

$$1 \approx \varrho_{\tau_2}^2 \cdot (d\lambda \cdot \exp(d\varrho_{\tau_2}^2) + d\lambda \cdot \exp(-d\varrho_{\tau_2}^2) + 2) \gtrsim \varrho_{\tau_2}^2 \cdot \exp(d\varrho_{\tau_2}^2), \tag{D.16}$$

since we assume $d\lambda = (1 + \sigma^2) \cdot \xi$ is of constant order. Moreover, note that $\tau_2$ characterizes the time when $\varrho_t$ hit its peak value since $\varrho_t$ increases monotonically during the first two stages. And as we will show later, $\varrho_t$ continues to decrease thereafter. Thus, we conclude that

$$\max_{t \in \mathbb{R}^+} \varrho_t = \varrho_{\tau_2} = O\left(\sqrt{(\log d - \log\log d)/d}\right) = \widetilde{O}(1/\sqrt{d}),$$

by solving the inequality in (D.16) given sufficiently large $d$. More specifically, we solve the equation $d\lambda \cdot \varrho_{\tau_2}^2 \cdot \exp(d\varrho_{\tau_2}^2) = 1$ and use the expansion of Lambert-$W$ function. Here $\widetilde{O}(\cdot)$ omits logarithmic factors. Therefore, we prove that the value of $\varrho_t$ forms a peak, with magnitude of $\widetilde{O}(1/\sqrt{d})$. This aligns with the observation in Figure 21b.

**Phase III: Convergence**   Next, we provide a heuristic analysis of the last phase showing that $\varrho_t$ decays to zero asymptotically. In particular, we employ a two-timescale analysis by first pushing $\varphi_t$ to its limit, enabling us to eliminate $\varphi_t$ by treating it as a function of $\varrho_t$. Recall that

$$\mathcal{L}_t = (1 + \sigma^2)/2 - 2\varphi_t\varrho_t + 2\varphi_t^2\varrho_t^2 + \lambda\varphi_t^2 \cdot \left(\exp(d\varrho_t^2) - \exp(-d\varrho_t^2)\right).$$

Note that the loss is quadratic with respect to $\varphi_t$. By finding the critical point of $\mathcal{L}_t$ with respect to $\varphi_t$, we know that the limiting value of $\varphi_t$ for a fixed $\varrho_t$ is given by

$$\varphi_t^* := \varphi^*(\varrho_t) = \frac{\varrho_t}{2\varrho_t^2 + \lambda\left(\exp(d\varrho_t^2) - \exp(-d\varrho_t^2)\right)}. \tag{D.17}$$

Moreover, as shown in Figure 21c, in this phase, the evolution of $\varphi_t^*$ and $\varphi_t$ are almost identical. Motivated by this observation, to characterize the limiting behavior of $\varrho_t$, we replace $\varphi_t$ by $\varphi_t^*$ in the ODE in (D.4). This leads to an ODE only involving $\varrho_t$:

$$\partial_t\varrho_t = \varphi_t^* \cdot \left\{1 - \left(2 + d\lambda \cdot \left(\exp(d\varrho_t^2) + \exp(-d\varrho_t^2)\right)\right) \cdot \varphi_t^*\varrho_t\right\}. \tag{D.18}$$

Using first-order approximations

$$\exp(d\varrho_t^2) - \exp(-d\varrho_t^2) \approx 2d\varrho_t^2, \qquad \text{and} \qquad \exp(d\varrho_t^2) + \exp(-d\varrho_t^2) \approx 2 + d^2\varrho_t^4$$

in (D.17) and (D.18), we have $\varphi_t^* \approx (2(d\lambda + 1) \cdot \varrho_t)^{-1}$ and

$$\partial_t\varrho_t \approx \frac{1}{2(d\lambda + 1) \cdot \varrho_t} \cdot \left\{1 - \frac{2 + d\lambda \cdot \left(2 + d^2\varrho_t^4\right)}{2(d\lambda + 1)}\right\} = -\frac{d^3\lambda}{4(d\lambda + 1)^2} \cdot \varrho_t^3.$$

Solving this ODE in closed form, we can characterize the evolution of $\varrho_t$ and $\varphi_t^* = \varphi^*(\varrho_t)$ by

$$\varrho_t \approx \left(\frac{1}{\varrho_{\tau_2}^2} + \frac{d^3\lambda \cdot (t - \tau_2)}{2(d\lambda + 1)^2}\right)^{-1/2}, \quad \varphi_t^* \approx \frac{1}{2(d\lambda + 1)} \cdot \left(\frac{1}{\varrho_{\tau_2}^2} + \frac{d^3\lambda \cdot (t - \tau_2)}{2(d\lambda + 1)^2}\right)^{1/2}, \quad \forall t \in [\tau_2, \infty).$$

As mentioned above, when $t$ is sufficiently large, we regard that $\varphi_t^*$ is almost equal to $\varphi_t$. Furthermore, we know that $\varrho_\tau^2 \approx \log d/d$. Under the high-dimensional regime where $d/L \to \xi$, we know that $d\lambda \to (1 + \sigma^2) \cdot \xi < 1$, which can be regarded as a constant. Thus, $\varrho_t$ decreases and converges to $0^+$ at a rate of $\Theta((d\sqrt{t})^{-1})$, i.e., $\varrho_\infty \to 0^+$. While $\varphi_t$ increases at a rate of $\Theta(d\sqrt{t})$ when $t$ increases, and thus grows to infinity. Moreover, while $t$ increases, the product $2\varphi_t\varrho_t$ remains almost constant and converges to $1/(1 + (1 + \sigma^2) \cdot \xi)$ during this phase.

In summary, through a heuristic two-timescale analysis, we prove that $\varrho_t$ decreases to zero after reaching its peak. And $\varphi_t$ increases to infinity while maintaining $\varphi_t\varrho_t$ at a constant level.

# E. Implications of Learned Pattern of QK and OV Circuits

## E.1. Transformer Model as a Sum of Kernel Regressors

Recall that we use $\mathtt{smax}(\cdot)$ denote the softmax function and use $Z_{\mathsf{ebd}} \in \mathbb{R}^{(d+1) \times (L+1)}$ denote $\begin{bmatrix} Z & z_q \end{bmatrix}$, where $Z$ contains the first $L$ demonstration samples and $z_q = (x_q^\top, 0)^\top$ contains the test input. Here the $L$ demonstration samples are generated from a noisy linear model with parameter $\beta$, which itself is drawn from a distribution $\mathsf{P}_\beta$. The transformer output $\mathtt{TF}_\theta(Z_{\mathsf{ebd}})$ is specified in (2.2), which is a $(d+1) \times (L+1)$-dimensional matrix. Finally, the prediction head outputs the $(d+1, L+1)$-th entry of $\mathtt{TF}_\theta(Z_{\mathsf{ebd}})$, denoted as $\widehat{y}_q$, which is used to predict the desired response $\beta^\top x_q$. Notice that the $(d+1, L+1)$-th entry of $Z_{\mathsf{ebd}}$ is the $(d+1)$-th entry of $z_q$, which is equal to zero. Due to causal masking, $z$ does not attend to itself and only attend to the previous tokens, namely $Z$. Thus, we can simplify the $(L+1)$-th column of $\mathtt{TF}_\theta(Z_{\mathsf{ebd}})$ to

$$\sum_{h=1}^H O^{(h)} V^{(h)} Z \cdot \mathtt{smax}(Z^\top K^{(h)\top} Q^{(h)} z_q) \in \mathbb{R}^{(d+1)}. \tag{E.1}$$

Compared with (2.2), we replace $Z_{\mathsf{ebd}}$ with $Z$ in the value part of attention, because causal masking ensures that the output of the softmax function is a probability distribution over $[L]$. Then, $\widehat{y}_q$ is the last entry of the vector in (E.1). Under the simplified parametrization given in (3.1), we have

$$Z^\top K^{(h)\top} Q^{(h)} z_q = \omega^{(h)} \cdot X x_q \in \mathbb{R}^L, \qquad O^{(h)} V^{(h)} Z = \mu^{(h)} y \in \mathbb{R}^L, \qquad \forall h \in [H].$$

Thus, we can simplify $\widehat{y}_q$ to the following form:

$$\widehat{y}_q = \sum_{h=1}^H \mu^{(h)} \cdot \left\langle y, \mathtt{smax}\left(\omega^{(h)} \cdot X x_q\right) \right\rangle := \sum_{h=1}^H \mu^{(h)} \cdot y^\top p^{(h)}, \tag{E.2}$$

where we define $p^{(h)} = \mathtt{smax}\left(\omega^{(h)} \cdot X x_q\right)$ for all $h \in [H]$. Here, $p^{(h)}$ and $\mu^{(h)} \cdot y$ are the attention score and value of the $h$-th head at position $(L+1)$, respectively.

**Interpretation of** (E.2) **as a Sum of Kernel Regressors.** Consider the $h$-th head, where $p^{(h)}$ is a probability distribution over $[L]$, and the probability assigned to each $\ell$ is proportional to $\exp(\omega^{(h)} \cdot x_\ell^\top x_q)$. Then, the output of the $h$-th head is equal to

$$\mu^{(h)} \cdot \sum_{\ell=1}^L y_\ell \cdot \mathfrak{K}(x_q, x_\ell; \omega^{(h)}), \qquad \text{where} \quad \mathfrak{K}(x_q, x_\ell; \omega) = \frac{\exp(\omega \cdot x_\ell^\top x_q)}{\sum_{i=1}^L \exp(\omega \cdot x_i^\top x_q)}. \tag{E.3}$$

Here $\mathfrak{K}(x_q, x_\ell; \omega)$ is a kernel function that captures the similarity of $x_q$ and $x_\ell$ and the parameter $1/\omega$ plays the role of bandwidth of the kernel. Notice that the kernel $\mathfrak{K}$ in (E.3) is slightly different from the standard Gaussian radial basis function (RBF) kernel, which is defined as $\exp(-\omega \cdot \|x_q - x_\ell\|_2^2)$, where $\omega$ is a parameter. These two kernels coincide when when $x_q$ and $x_\ell$ are on the unit sphere. However, in our case, $\mathsf{P}_x$ is supported on $\mathbb{R}^d$ and thus these two kernels are different. Nevertheless, each term as in (E.3) is still a Nadaraya-Watson kernel regression predictor. Moreover, the intuition of such an estimator is clear: the output in (E.3) is a weighted sum of the responses $\{y_\ell\}_{\ell \in [L]}$ in the demonstration data, and the weights are determined by the similarity between the test input $x_q$ and the demonstration input $x_\ell$. In summary, when the parameters of the transformer are given by (3.1), the prediction $\widehat{y}_q$ is a sum of $H$ kernel regressors.

**Interpretation of Multi-Head Attention Predictor with** $(\omega, \mu) \subseteq \mathscr{S}_\gamma$**.** As shown in the empirical observations in §3 and the solution manifold in §4.2, parameters $\{(\omega^{(h)}, \mu^{(h)})\}_{h \in [H]}$ found by the transformer model are in the solution manifold $\mathscr{S}_\gamma$ defined in (4.5) for some $\gamma > 0$. Thus, $\widehat{y}_q$ in (E.2) can be simplified as a sum of two kernel regressors:

$$\widehat{y}_q = \sum_{\ell=1}^L \mu_\gamma \cdot y_i \cdot \left(\mathfrak{K}(x_q, x_\ell; \gamma) - \mathfrak{K}(x_q, x_\ell; -\gamma)\right). \tag{E.4}$$

The intuition behind such an estimator is as follows. The kernel $\mathfrak{K}(x_q, x_\ell; \gamma)$ assigns a larger weight to $x_\ell$ when $x_\ell^\top x_q$ is large, i.e., when $x_q$ and $x_\ell$ are similar. The kernel $\mathfrak{K}(x_q, x_\ell; -\gamma)$ is large when $x_\ell^\top x_q$ is small, i.e., when $x_q$ and $x_\ell$ are dissimilar. Thus, the predictor in (E.4) is a sum of two kernel regression predictors based on datasets $\{x_\ell, y_\ell\}_{\ell \in [L]}$ and $\{-x_\ell, -y_\ell\}_{\ell \in [L]}$ respectively. It is also reasonable why both these two kernel regressors appear. Specifically, $(x_\ell, y_\ell)$ and $(-x_\ell, -y_\ell)$ have the same distribution when $x_\ell \sim \mathcal{N}(0, I_d)$ and $y_\ell$ is generated from a linear model. Thus, we can use both $\{x_\ell, y_\ell\}_{\ell \in [L]}$ and $\{-x_\ell, -y_\ell\}_{\ell \in [L]}$ to learn the parameter $\beta$.

**Difference between Single-Head and Multi-Head Softmax Attention.** As shown in Chen et al. (2024a), when $H = 1$, the learned transformer implements a Nadaraya-Watson kernel regressor as in (E.3), with $|\omega^{(1)}| \asymp 1/\sqrt{d}$. Moreover, the KQ and OV matrices of the trained single-head model exhibit a positive or negative only pattern. In contrast, when $H \geq 2$, the multi-head attention mechanism learns to behave as a sum of kernel regressors, achieving better performance due to its greater expressive power. Furthermore, the trained multi-head model displays a coupled positive-negative pattern and works in a regime with a significantly smaller magnitudes of KQ parameters. Therefore, despite the form of the sum of kernel regressors, the multi-head attention indeed learns to approximate a variant of the debiased GD predictor, which better captures the linear structure of the problem compared to the single-head case (see §4.3 for details).

### E.2. Proof of Proposition 4.1: Loss Approximation

For better understanding the approximation and scaling conditions in Proposition E.1, we provide a more detailed discussion alongside the simulation results in §C.4.

**Proposition E.1** (Formal Statement of Proposition 4.1). *Under the parametrization in* (3.1), *consider the $H$-head attention model with dimension $d \in \mathbb{Z}^+$ and sample size $L \in \mathbb{Z}^+$. Suppose that the parameters $(\omega, \mu) \subseteq \mathbb{R}^{2H}$ satisfy one of the following conditions:*

$$\|\omega\|_\infty \leq 0.1\sqrt{\log L / \max\{d, \log L\}}, \qquad \max\{\|\mu\|_\infty, \|\mu\|_\infty^2\} \lesssim L^{-\lambda/2+3/10},$$

*where $\lambda > 0$ is an absolute constant defining the error level. Then, it holds that*

$$\mathcal{L}(\omega, \mu) = 1 + \sigma^2 - 2\mu^\top \omega + \mu^\top \big(\omega\omega^\top + (1 + \sigma^2) \cdot L^{-1} \cdot \exp(d\omega\omega^\top)\big)\mu + O(dH^2 \cdot L^{-\lambda}).$$

*Here $O(\cdot)$ omits some universal constant.*

The informal statement in Proposition 4.1 follows by assuming $d > \log L$ and setting $\lambda = 1/5$. A more refined trade-off between the scales of $\omega$ and $\mu$ can be achieved by carefully selecting $\kappa$, the scaling factor for $\omega$, i.e., $\|\omega\|_\infty \leq \kappa\sqrt{\log L / \max\{d, \log L\}}$, where $\kappa$ is now set to a constant 0.1.

*Proof of Proposition E.1.* We prove this proposition in three steps:

- In **Part 1**, we first expand the population loss $\mathcal{L}(\omega, \mu)$ into multiple terms, where each term involves expectations over the randomness of both coefficient $\beta \sim \mathsf{P}_\beta$ and ICL samples $Z_{\mathsf{ebd}}$.
- In **Part 2**, we simplify these terms by conditioning on $x_q$ and taking an expectation over the randomness of the rest $\beta$ and $\{(x_\ell, y_\ell)\}_{\ell \in [L]}$ within the context sequence, which gives us a reformulation of the loss in terms of $\|x_q\|_2^2$ and *moments* of the attention probabilities.
- In **Part 3**, we isolate the high-order moments from the low-order moments in the reformulated loss. We show that low-order terms can be well approximated by a polynomial of OV weights and exponential QK weights up to $O(dH \cdot L^{-\lambda})$ error, and high-order terms can also be uniformly bounded by $O(dH^2 \cdot L^{-\lambda})$ under appropriate scaling conditions.

STEP 1. EXPAND POPULATION LOSS INTO MULTIPLE TERMS.

Recall that under the simplified parametrization given in (3.1), the prediction of the transformer model, $\widehat{y}_q$, is given by (E.2), where we let $p^{(h)} = \mathtt{smax}(\omega^{(h)} \cdot Xx_q)$ be the attention score of the $h$-th head. Then, the population loss defined in (2.3) can be expanded as follows:

$$\mathcal{L}(\omega, \mu) = \mathbb{E}\left[\left(\beta^\top x_q - \sum_{h=1}^H \mu^{(h)} \cdot (X\beta + \epsilon)^\top p^{(h)}\right)^2\right] + \sigma^2$$

$$= \underbrace{\mathbb{E}\left[(\beta^\top x_q)^2\right]}_{\textbf{(i)}} - 2\underbrace{\mathbb{E}\left[\beta^\top x_q \cdot \sum_{h=1}^H \mu^{(h)}(X\beta + \epsilon)^\top p^{(h)}\right]}_{\textbf{(ii)}}$$

$$+ \underbrace{\mathbb{E}\left[\sum_{h=1}^H \sum_{h'=1}^H \mu^{(h)}\mu^{(h')} \cdot (X\beta + \epsilon)^\top p^{(h)} \cdot (X\beta + \epsilon)^\top p^{(h')}\right]}_{\textbf{(iii)}} + \sigma^2. \tag{E.5}$$

We let $\epsilon = (\epsilon_1, \ldots, \epsilon_L) \in \mathbb{R}^L$ with $\epsilon_i \overset{\text{i.i.d.}}{\sim} \mathcal{N}(0, \sigma^2)$ be the noise terms in $\{y_\ell\}_{\ell \in [L]}$. Notice that $p^{(h)} \in \mathbb{R}^L$ and $X \in \mathbb{R}^{L \times d}$. Recall that $\beta \sim \mathcal{N}(0, I_d/d)$ and $x_q \sim \mathcal{N}(0, I_d)$, then we have

$$\textbf{(i)} = \mathbb{E}\left[(\beta^\top x_q)^2\right] = \mathbb{E}\left[x_q^\top \beta\beta^\top x_q\right] = \mathbb{E}\|x_q\|_2^2/d = 1. \tag{E.6}$$

For the remaining two terms, we can simplify using a similar argument to marginalize $\beta \sim \mathcal{N}(0, I_d/d)$. Specifically, for term **(ii)**, we have

$$\textbf{(ii)} = \mathbb{E}\left[\beta^\top x_q \sum_{h=1}^H \mu^{(h)} \beta^\top X^\top p^{(h)}\right]$$

$$= \sum_{h=1}^H \mu^{(h)} \cdot \mathbb{E}\left[x_q^\top \beta\beta^\top X^\top p^{(h)}\right] = \sum_{h=1}^H \frac{\mu^{(h)}}{d} \cdot \mathbb{E}\left[x_q^\top X^\top p^{(h)}\right]. \tag{E.7}$$

For term **(iii)**, using the independence between $\epsilon$ and $(X, x_q, \beta)$, we have

$$\textbf{(iii)} = \sum_{h=1}^H \sum_{h'=1}^H \mu^{(h)} \mu^{(h')} \cdot \mathbb{E}\left[p^{(h')^\top}(X\beta + \epsilon)(X\beta + \epsilon)^\top p^{(h)}\right]$$

$$= \sum_{h=1}^H \sum_{h'=1}^H \mu^{(h)} \mu^{(h')} \cdot \mathbb{E}\left[p^{(h')^\top} X\beta\beta^\top X^\top p^{(h)}\right] + \sum_{h=1}^H \sum_{h'=1}^H \mu^{(h)} \mu^{(h')} \cdot \mathbb{E}\left[p^{(h')^\top} \epsilon\epsilon^\top p^{(h)}\right]$$

$$= \sum_{h=1}^H \sum_{h'=1}^H \frac{\mu^{(h)} \mu^{(h')}}{d} \cdot \mathbb{E}\left[p^{(h')^\top} XX^\top p^{(h)}\right] + \sigma^2 \cdot \sum_{h=1}^H \sum_{h'=1}^H \mu^{(h)} \mu^{(h')} \cdot \mathbb{E}\left[p^{(h')^\top} p^{(h)}\right], \tag{E.8}$$

where the last equality holds because $\mathbb{E}[\epsilon\epsilon^\top] = \sigma^2 \cdot I_L$ and $\mathbb{E}[\beta\beta^\top] = I_d/d$.

In the following, to further simplify the expectation terms in (E.7) and (E.8), we decouple the randomness of $X \in \mathbb{R}^{L \times d}$ and $x_q \in \mathbb{R}^d$, where $\{x_\ell\}_{\ell \in [L]}$ and $x_q$ are i.i.d. random vectors sampled from $\mathcal{N}(0, I_d)$. To this end, we take conditional expectations with respect to $X$ given $x_q$.

STEP 2. TAKE CONDITIONAL EXPECTATIONS GIVEN $x_q$.

Recall that $p^{(h)} = \texttt{smax}(\omega^{(h)} \cdot Xx_q)$. In the following, for any $\ell \in [L]$, we let $p_\ell^{(h)}$ and $[\texttt{smax}(\omega^{(h)} \cdot Xx_q)]_\ell$ denote the $\ell$-th entry of the attention probability $p^{(h)}$. Also note that $X = [x_1, \ldots, x_L]^\top$ with $x_\ell \overset{\text{i.i.d.}}{\sim} \mathcal{N}(0, I_d)$. For any random vector $x \sim \mathcal{N}(0, I_d)$ and any differentiable function $g$, Stein's Lemma states that $\mathbb{E}[xg(x)] = \mathbb{E}[\nabla g(x)]$. By this lemma, for all $h \in [H]$, it holds that

$$\mathbb{E}[X^\top p^{(h)} \,|\, x_q] = \sum_{\ell=1}^L \mathbb{E}\left[x_\ell \cdot [\texttt{smax}(\omega^{(h)} \cdot Xx_q)]_\ell \,|\, x_q\right] = \sum_{\ell=1}^L \mathbb{E}\left[\nabla_{x_\ell} [\texttt{smax}(\omega^{(h)} \cdot Xx_q)]_\ell \,|\, x_q\right]$$

$$= \sum_{\ell=1}^L \omega^{(h)} \cdot x_q \cdot \mathbb{E}\left[p_\ell^{(h)} - (p_\ell^{(h)})^2 \,|\, x_q\right] = \omega^{(h)} \cdot x_q - \omega^{(h)} \cdot x_q \cdot \mathbb{E}\left[\|p^{(h)}\|_2^2 \,|\, x_q\right]. \tag{E.9}$$

Here, the second equality follows from Stein's Lemma, the third equality follows from the derivative of the softmax function, and the last equality follows from the fact that $\sum_{\ell=1}^L p_\ell^{(h)} = 1$. Combining (E.9) and (E.7), conditioning on the value of $x_q$, we can write the second term in (E.5) as

$$\textbf{(ii)}_{\,|\, x_q} = \sum_{h=1}^H \mu^{(h)} \omega^{(h)}/d \cdot \|x_q\|_2^2 - \sum_{h=1}^H \mu^{(h)} \omega^{(h)} \cdot \|x_q\|_2^2/d \cdot \mathbb{E}[\|p^{(h)}\|_2^2 \,|\, x_q], \tag{E.10}$$

where we use $\textbf{(ii)}_{\,|\, x_q}$ to denote the counterpart of **(ii)** with $x_q$ fixed. It remains to simplify the third term in (E.5). In particular, we need to handle terms of the form $\mathbb{E}[p^{(h')^\top} XX^\top p^{(h)} \,|\, x_q]$. To this end, we introduce the following lemma.

**Lemma E.2.** *Consider random variable $X = [x_1, \ldots, x_L]^\top \in \mathbb{R}^{L \times d}$ with $x_\ell \overset{i.i.d.}{\sim} \mathcal{N}(0, I_d)$ for all $\ell \in [L]$. Let $\omega, \widetilde{\omega} \in \mathbb{R}$ and $\upsilon \in \mathbb{R}^d$ be fixed parameters. We define $p = \mathtt{smax}(\omega \cdot X\upsilon)$ and $\widetilde{p} = \mathtt{smax}(\widetilde{\omega} \cdot X\upsilon)$. Then, it holds*

$$
\begin{aligned}
\mathbb{E}[\widetilde{p}^\top X X^\top p] &= d \cdot \mathbb{E}[p^\top \widetilde{p}] + \omega \cdot \widetilde{\omega} \cdot \|\upsilon\|_2^2 \cdot \mathbb{E}[(1 - \|p\|_2^2) \cdot (1 - \|\widetilde{p}\|_2^2)] \\
&\quad + 2\omega^2 \cdot \|\upsilon\|_2^2 \cdot \mathbb{E}[-\widetilde{p}^\top p^{\odot 2} + \widetilde{p}^\top p \cdot \|p\|_2^2] + 2\widetilde{\omega}^2 \cdot \|\upsilon\|_2^2 \cdot \mathbb{E}[-p^\top \widetilde{p}^{\odot 2} + p^\top \widetilde{p} \cdot \|\widetilde{p}\|_2^2] \\
&\quad + \omega \cdot \widetilde{\omega} \cdot \|\upsilon\|_2^2 \cdot \mathbb{E}[p^\top \widetilde{p} - p^\top \widetilde{p}^{\odot 2} - \widetilde{p}^\top p^{\odot 2} + (p^\top \widetilde{p})^2].
\end{aligned}
$$

*Here $p^{\odot 2} = p \odot p$ denotes the element-wise square of $p \in \mathbb{R}^L$.*

*Proof of Lemma E.2.* See §G.1 for a detailed proof. $\qquad\square$

To compute $\mathbb{E}[p^{(h')^\top} X X^\top p^{(h)} \mid x_q]$, we apply Lemma E.2 with $\omega = \omega^{(h)}$, $\widetilde{\omega} = \omega^{(h')}$, and $\upsilon = x_q$. Then, for any $h, h' \in [H]$, we obtain that

$$
\begin{aligned}
&\mathbb{E}[p^{(h')^\top} X X^\top p^{(h)} \mid x_q] \\
&= d \cdot \mathbb{E}[p^{(h)^\top} p^{(h')} \mid x_q] + \omega^{(h)} \omega^{(h')} \cdot \|x_q\|_2^2 \cdot \mathbb{E}[(1 - \|p^{(h)}\|_2^2)(1 - \|p^{(h')}\|_2^2) \mid x_q] \\
&\quad + 2(\omega^{(h)})^2 \cdot \|x_q\|_2^2 \cdot \mathbb{E}[-p^{(h')^\top} p^{(h) \odot 2} + p^{(h')^\top} p^{(h)} \cdot \|p^{(h)}\|_2^2 \mid x_q] \\
&\quad + 2(\omega^{(h')})^2 \cdot \|x_q\|_2^2 \cdot \mathbb{E}[-p^{(h)^\top} p^{(h') \odot 2} + p^{(h)^\top} p^{(h')} \cdot \|p^{(h')}\|_2^2 \mid x_q] \\
&\quad + \omega^{(h)} \cdot \omega^{(h')} \cdot \|x_q\|_2^2 \cdot \mathbb{E}[p^{(h)^\top} p^{(h')} - p^{(h)^\top} p^{(h') \odot 2} - p^{(h')^\top} p^{(h) \odot 2} + (p^{(h)^\top} p^{(h')})^2 \mid x_q] \\
&:= d \cdot \mathbb{E}[p^{(h)^\top} p^{(h')} \mid x_q] + \omega^{(h)} \omega^{(h')} \cdot \|x_q\|_2^2 + \|x_q\|_2^2 \cdot \mathcal{T}_{h,h'}(x_q),
\end{aligned} \tag{E.11}
$$

where we use $\mathcal{T}_{h,h'}(x_q)$ to denote the high-order terms in (E.11). Here the high-order terms contain the product of $\|x_q\|_2^2$ and expectations of polynomials of $p^{(h)}$ and $p^{(h')}$ with degrees at least two.

Combine (E.5), (E.6), (E.7) and (E.8) and the form of conditional expectations in (E.9), (E.10), and (E.11), we can write the conditional expectation of the loss function given $x_q$ into a polynomial of $\{p^{(h)}\}_{h \in [H]}$. Specifically, we have

$$
\begin{aligned}
\mathcal{L}_{|x_q}(\omega, \mu) &:= \mathbb{E}\left[\left(y_q - \sum_{h=1}^H \mu^{(h)} y^\top p^{(h)}\right)^2 \,\middle|\, x_q\right] \\
&= 1 + \sigma^2 - \frac{2\|x_q\|_2^2}{d} \cdot \sum_{h=1}^H \mu^{(h)} \omega^{(h)} \\
&\quad + \frac{\|x_q\|_2^2}{d} \cdot \sum_{h=1}^H \sum_{h'=1}^H \mu^{(h)} \mu^{(h')} \omega^{(h)} \omega^{(h')} + (1 + \sigma^2) \cdot \sum_{h=1}^H \sum_{h'=1}^H \mu^{(h)} \mu^{(h')} \cdot \mathbb{E}[p^{(h)^\top} p^{(h')} \mid x_q] \\
&\quad + \frac{2\|x_q\|_2^2}{d} \cdot \sum_{h=1}^H \mu^{(h)} \omega^{(h)} \cdot \mathbb{E}[\|p^{(h)}\|_2^2 \mid x_q] + \frac{\|x_q\|_2^2}{d} \cdot \sum_{h=1}^H \sum_{h'=1}^H \mu^{(h)} \mu^{(h')} \cdot \mathcal{T}_{h,h'}(x_q).
\end{aligned} \tag{E.12}
$$

Notice that $\mathcal{L}_{|x_q}(\omega, \mu)$ above is a function of $x_q$. In the following, we analyze the expected value of each term in (E.12).

STEP 3. MARGINALIZE $x_q$ AND ELIMINATIONS OF HIGH-ORDER TERMS.

Let $\mu = (\mu^{(1)}, \ldots, \mu^{(H)})^\top \in \mathbb{R}^H$ and $\omega = (\omega^{(1)}, \ldots, \omega^{(H)})^\top \in \mathbb{R}^H$. Since $x_q \sim \mathcal{N}(0, I_d)$, we have $\mathbb{E}[\|x_q\|_2^2] = d$. By direct calculation, we have

$$
\mathbb{E}\left[\frac{\|x_q\|_2^2}{d} \cdot \sum_{h=1}^H \mu^{(h)} \omega^{(h)}\right] = \mu^\top \omega, \quad \mathbb{E}\left[\frac{\|x_q\|_2^2}{d} \cdot \sum_{h=1}^H \sum_{h'=1}^H \mu^{(h)} \mu^{(h')} \omega^{(h)} \omega^{(h')}\right] = (\mu^\top \omega)^2. \tag{E.13}
$$

In the following, we show that $\mathbb{E}[p^{(h)^\top} p^{(h')}]$ with large $L$ can be approximated by $\exp(d \cdot \omega^{(h)} \omega^{(h')})/L$ (**Step 3.1**), and the remaining terms are mostly high-order and thus can be ignored (**Step 3.2**).

**Step 3.1. Approximate** $\mathbb{E}[p^{(h)^\top} p^{(h')}]$ **under Large** $L$**.** Next, we approximate $\mathbb{E}[p^{(h)^\top} p^{(h')}]$ by $\exp(d \cdot \omega^{(h)}\omega^{(h')})/L$ under large $L$. This enables us to control the term

$$(1 + \sigma^2) \cdot \sum_{h=1}^{H} \sum_{h'=1}^{H} \mu^{(h)} \mu^{(h')} \cdot \mathbb{E}[p^{(h)^\top} p^{(h')}],$$

of the loss $L(\omega, \mu)$, as given in (E.12). Our derivation is based on the following insight. When $\omega^{(h)}$ is small, for $p^{(h)}$, we can approximate the denominator in the softmax function by $L$. Viewing $p^{(h)}$ as a function of $\omega^{(h)}$, the first two terms in the Taylor expansion is:

$$p_\ell^{(h)} \approx L^{-1} \cdot (1 + \omega^{(h)} \cdot x_\ell^\top x_q), \qquad \forall \ell \in [L].$$

Here, we approximate the denominator $\sum_{\ell=1}^{L} \exp(\omega^{(h)} \cdot x_\ell^\top x_q)$ in the softmax function by $L$. We can similarly write down the Taylor expansion for $p^{(h')}$ in terms of $\omega^{(h')}$. Thus, we have

$$\mathbb{E}[p^{(h)^\top} p^{(h')} \,|\, x_q] \approx \frac{1}{L^2} \cdot \sum_{\ell=1}^{L} \mathbb{E}\big[(1 + \omega^{(h)} \cdot x_\ell^\top x_q) \cdot (1 + \omega^{(h')} \cdot x_\ell^\top x_q)\big| \, x_q\big]$$

$$= L^{-1} \cdot (1 + \omega^{(h)}\omega^{(h')} \cdot \|x_q\|_2^2).$$

This implies the following approximation scheme:

$$\mathbb{E}[p^{(h)^\top} p^{(h')} \,|\, x_q] \approx L^{-1} \cdot (1 + \omega^{(h)}\omega^{(h')} \cdot x_q^\top x_q) \approx L^{-1} \cdot \exp(\omega^{(h)}\omega^{(h')} \cdot x_q^\top x_q). \tag{E.14}$$

Similarly, by taking expectation with respect to $x_q$ in (E.14), we have

$$\mathbb{E}[p^{(h)^\top} p^{(h')}] \approx L^{-1} \cdot \big(1 + \omega^{(h)}\omega^{(h')} \cdot \mathbb{E}[x_q^\top x_q]\big) = L^{-1} \cdot (1 + d \cdot \omega^{(h)}\omega^{(h')}) \approx L^{-1} \cdot \exp(d \cdot \omega^{(h)}\omega^{(h')}).$$

In the following, we rigorously prove the above approximation. We prove by leveraging a lemma from Chen et al. (2024a) to quantify the accuracy of the approximation in (E.14) and the large deviation of $x_q$. We first present the lemma below.

**Lemma E.3.** *Let* $x_q \in \mathbb{R}^d$ *be any fixed vector that satisfies* $\max\{\|W x_q\|_2^2, \|\widetilde{W} x_q\|_2^2\} \leq \tau \log L$*, where* $\tau > 0$ *is a constant and* $W \in \mathbb{R}^{d \times d}$, $\widetilde{W} \in \mathbb{R}^{d \times d}$ *are two parameter matrices. Let* $p = \mathtt{smax}(X W x_q)$ *and* $\widetilde{p} = \mathtt{smax}(X \widetilde{W} x_q)$*, where* $X = [x_1, \ldots, x_L]^\top \in \mathbb{R}^{L \times d}$ *with* $x_\ell \overset{i.i.d.}{\sim} \mathcal{N}(0, I_d)$*, it holds that*

*(i).* $\left| \mathbb{E}[p^\top \widetilde{p} \,|\, x_q] - L^{-1} \cdot \exp\left(x_q^\top \widetilde{W}^\top W x_q\right) \right| = O(L^{-2(1-\epsilon)});$

*(ii).* $\max\left\{ \mathbb{E}[p^\top \widetilde{p}^{\odot 2} \,|\, x_q], \mathbb{E}[\|p\|_2^2 \cdot p^\top \widetilde{p} \,|\, x_q], \mathbb{E}[(p^\top \widetilde{p})^2 \,|\, x_q] \right\} = O(L^{-2(1-\epsilon)}),$

*where* $\epsilon = \sqrt{\tau/2} + 3/(1 + \sqrt{1 + 1/\tau})$*. When* $x_q \sim \mathcal{N}(0, I_d)$ *and* $\|\omega\|_\infty \lesssim \tau \log L / \max\{d, \log L\}$*, we further have*

*(iii).* $\mathbb{E}\left[\left(\mathbb{E}[p^\top \widetilde{p} \,|\, x_q] - L^{-1} \cdot \exp\left(x_q^\top \widetilde{W}^\top W x_q\right)\right)^2\right] = O(L^{-(3-\epsilon)}).$

*Note all these three arguments are at population level and thus hold with probability one.*

*Proof of Lemma E.3.* See Lemma B.2 and Lemma B.3 in Chen et al. (2024a) for detailed proofs. The first two items are in Lemma B.2, and the third item is in Lemma B.3. □

The first item in Lemma E.3 quantifies the error of the approximation in (E.14). The second item shows that the high-order terms of $p$ and $\widetilde{p}$ are small. Additionally, the third item quantifies the squared error of the approximation given $x_q \sim \mathcal{N}(0, I_d)$. Furthermore, we note that the approximation errors in Lemma E.3 depends on $\epsilon$, which depends on the parameter $\tau$. To make the approximation errors small, we need to choose a small $\tau$ such that the resulting $\epsilon$ is small.

To apply this lemma to $p^{(h)}$ and $p^{(h')}$, we set $W = \omega^{(h)} \cdot I_d$ and $\widetilde{W} = \omega^{(h')} \cdot I_d$. To satisfies the requirement that $x_q$ is bounded in the lemma, we control the tail behavior of $\|x_q\|_2^2$ using the concentration results for $\chi^2$-distribution. By setting $t = \log L$ in Lemma G.1, we have

$$\mathbb{P}\left(\|x_q\|_2^2 > 5\max\{d, \log L\}\right) \leq \mathbb{P}\left(\|x_q\|_2^2 > d + 2\sqrt{d\log L} + 2\log L\right) \leq L^{-1}. \tag{E.15}$$

In the following, we assume that $\omega \in \mathbb{R}^H$ satisfies the following condition:

$$\text{(C1)} \qquad \|\omega\|_\infty \leq \kappa\sqrt{\log L / \max\{d, \log L\}},$$

where $\kappa$ denotes a parameter to be determined later. Based on any $\omega$ satisfying (C1), we define the following good event where $\|\omega^{(h)} x_q\|_2^2$ are all bounded:

$$\mathcal{E}_{\text{good}} = \left\{\forall h \in [H], \ \|\omega^{(h)} \cdot x_q\|_2^2 \leq 5\kappa^2 \cdot \log L\right\}. \tag{E.16}$$

Then, when $\omega$ satisfies (C1), by (E.15) we have

$$\mathbb{P}\left(\mathcal{E}_{\text{good}}^c\right) \leq \mathbb{P}\left(\|\omega\|_\infty^2 \cdot \|x_q\|_2^2 > 5\kappa^2 \log L\right) \leq \mathbb{P}\left(\|x_q\|_2^2 > 5\max\{d, \log L\}\right) \leq L^{-1}.$$

Besides, when $\mathcal{E}_{\text{good}}$ holds, the premise of Lemma E.3 holds with $\tau = 5\kappa^2$. Now we are ready to work on $\mathbb{E}[p^{(h)\top} p^{(h')}]$. We decompose the $\mathbb{E}[p^{(h)\top} p^{(h')}]$ as follows:

$$\mathbb{E}[p^{(h)\top} p^{(h')}] = \mathbb{E}\left[p^{(h)\top} p^{(h')} \cdot \mathbb{1}\left(\mathcal{E}_{\text{good}}\right)\right] + \mathbb{E}\left[p^{(h)\top} p^{(h')} \cdot \mathbb{1}\left(\mathcal{E}_{\text{good}}^c\right)\right]$$
$$\leq \mathbb{E}\left[p^{(h)\top} p^{(h')} \cdot \mathbb{1}\left(\mathcal{E}_{\text{good}}\right)\right] + L^{-1}, \tag{E.17}$$

where the last inequality follows from the fact that $p^{(h)\top}$ and $p^{(h')}$ are probability distributions over $[L]$, and thus $p^{(h)\top} p^{(h')} \leq 1$. Furthermore, we decompose the first term on the RHS of (E.17) as

$$\mathbb{E}\left[p^{(h)\top} p^{(h')} \cdot \mathbb{1}\left(\mathcal{E}_{\text{good}}\right)\right] = \underbrace{\mathbb{E}\left[\left(p^{(h)\top} p^{(h')} - L^{-1}\exp(\omega^{(h)}\omega^{(h')} \cdot \|x_q\|_2^2|)\right) \cdot \mathbb{1}\left(\mathcal{E}_{\text{good}}\right)\right]}_{\textbf{(iv)}}$$
$$+ \underbrace{L^{-1} \cdot \left(\mathbb{E}\left[\exp(\omega^{(h)}\omega^{(h')} \cdot \|x_q\|_2^2) \cdot \mathbb{1}\left(\mathcal{E}_{\text{good}}\right)\right] - \exp(d \cdot \omega^{(h)}\omega^{(h')})\right)}_{\textbf{(v)}}$$
$$+ L^{-1} \cdot \exp(d \cdot \omega^{(h)}\omega^{(h')}) \tag{E.18}$$

We control term **(iv)** by applying Lemma E.3-(i) with $\tau = 5\kappa^2$, which implies that there exists and absolute constant $C > 0$ such that

$$\textbf{(iv)} \leq \left|\mathbb{E}\left[\mathbb{E}\left[(p^{(h)\top} p^{(h')} - L^{-1} \cdot \exp(\omega^{(h)}\omega^{(h')}\|x_q\|_2^2|)) \cdot \mathbb{1}\left(\mathcal{E}_{\text{good}}\right) \,\Big|\, x_q\right]\right]\right| \leq C_1 \cdot L^{-2(1-\epsilon)}. \tag{E.19}$$

Moreover, here $\epsilon$ is a function of specified constant $\kappa$ that controls the scaling of $\|\omega\|_\infty$, defined as

$$\epsilon = \kappa\sqrt{5/2} + 3/(1 + \sqrt{1 + (5\kappa^2)^{-1}}). \tag{E.20}$$

To handle term **(v)**, we write it as

$$\textbf{(v)} = L^{-1} \cdot \exp(d \cdot \omega^{(h)}\omega^{(h')}) \cdot \left(\mathbb{E}\left[\exp(\omega^{(h)}\omega^{(h')} \cdot (\|x_q\|_2^2 - d)) \cdot \mathbb{1}\left(\mathcal{E}_{\text{good}}\right)\right] - 1\right).$$

Let $\Upsilon$ denote the random variable $\exp(\omega^{(h)}\omega^{(h')} \cdot (\|x_q\|_2^2 - d)) \cdot \mathbb{1}\left(\mathcal{E}_{\text{good}}\right)$, which depends on $x_q$ only. Motivated by (E.15), we consider the cases where $\mathbb{1}(\|x_q\|_2^2 - d \leq 2\sqrt{d\log L} + 2\log L)$ is true or not separately. Note that when this event is true, we have

$$\Upsilon \leq \exp(|\omega^{(h)}\omega^{(h')}| \cdot (2\sqrt{d\log L} + 2\log L)). \tag{E.21}$$

When this event does not hold, we have

$$\exp(d \cdot \omega^{(h)} \omega^{(h')}) \cdot \Upsilon = \exp(\omega^{(h)} \omega^{(h')} \cdot \|x_q\|_2^2) \cdot \mathbb{1}\left(\mathcal{E}_{\text{good}}\right) \le \exp(5\kappa^2 \log L), \tag{E.22}$$

thanks to the event $\mathcal{E}_{\text{good}}(\omega)$ defined in (E.16). Therefore, we conclude that

$$\begin{aligned}
\text{(v)} &= L^{-1} \cdot \exp(d \cdot \omega^{(h)} \omega^{(h')}) \cdot \mathbb{E}[\Upsilon - 1] \\
&= L^{-1} \cdot \exp(d \cdot \omega^{(h)} \omega^{(h')}) \cdot \mathbb{E}[\Upsilon \cdot \mathbb{1}(\|x_q\|_2^2 - d \le 2\sqrt{d \log L} + 2\log L) - 1] \\
&\quad + L^{-1} \cdot \exp(d \cdot \omega^{(h)} \omega^{(h')}) \cdot \mathbb{E}[\Upsilon \cdot \mathbb{1}(\|x_q\|_2^2 - d > 2\sqrt{d \log L} + 2\log L)] \\
&\le L^{-1} \cdot \exp(d \cdot \omega^{(h)} \omega^{(h')}) \cdot \left(\exp\left(|\omega^{(h)} \omega^{(h')}| \cdot (2\sqrt{d \log L} + 2\log L)\right) - 1\right) \\
&\quad + L^{-2} \cdot \exp\left(5\kappa^2 \cdot \log L\right). 
\end{aligned} \tag{E.23}$$

Here, the last inequality follows from (E.21) and (E.22). Note that when (C1) is true, we have

$$\begin{aligned}
|\omega^{(h)} \omega^{(h')}| \cdot (2\sqrt{d \log L} + 2\log L) &\le \kappa^2 \cdot \log L / \max\{d, \log L\} \cdot (2\sqrt{d \log L} + 2\log L) \le 4\kappa^2 \cdot \log L, \\
\exp(d \cdot \omega^{(h)} \omega^{(h')}) &\le \exp\left(\kappa^2 \cdot \log L / \max\{d, \log L\} \cdot d\right) \le L^{\kappa^2}.
\end{aligned}$$

Thus, in this case, we have By substituting the arguments above back into (E.23), we obtain that

$$\text{(v)} \le L^{-1+5\kappa^2} + L^{-2+5\kappa^2} \le 2L^{-1+5\kappa^2}. \tag{E.24}$$

Combining (E.17), (E.18), (E.19) and (E.24), we have

$$\begin{aligned}
\mathbb{E}[p^{(h)^\top} p^{(h')}] &= L^{-1} \cdot \exp(d \cdot \omega^{(h)} \omega^{(h')}) + C_1 \cdot L^{-2(1-\epsilon)} + 2L^{-1+5\kappa^2} + L^{-1} \\
&= L^{-1} \cdot \exp(d \cdot \omega^{(h)} \omega^{(h')}) + C_1 \cdot L^{-2(1-\epsilon)} + 3L^{-1+5\kappa^2},
\end{aligned}$$

where we define $\epsilon$ as $\epsilon = \kappa\sqrt{5/2} + 3/(1 + \sqrt{1 + 1/(5\kappa^2)})$, which is a function of the specified constant $\kappa$, and $C_1$ is an absolute constant coming from (E.19). Thus, we conclude that

$$\begin{aligned}
&\sum_{h=1}^{H} \sum_{h'=1}^{H} \mu^{(h)} \mu^{(h')} \cdot \mathbb{E}[p^{(h)^\top} p^{(h')}] \\
&\le \frac{1}{L} \sum_{h=1}^{H} \sum_{h'=1}^{H} \mu^{(h)} \mu^{(h')} \cdot \exp(d \cdot \omega^{(h)} \omega^{(h')}) + \|\mu\|_\infty^2 \cdot H^2 \cdot \left(C_1 \cdot L^{-2(1-\epsilon)} + 3L^{-1+5\kappa^2}\right). 
\end{aligned} \tag{E.25}$$

This enables us to bound the corresponding term in the loss function $L(\omega, \mu)$, as given in (E.12).

**Step 3.2. Bound** $\mathbb{E}\left[\|x_q\|_2^2 \cdot \mathbb{E}[\|p^{(h)}\|_2^2 \mid x_q]\right]$ **under Large** $L$. Next, we bound the term

$$\mathbb{E}\left[\frac{2\|x_q\|_2^2}{d} \cdot \sum_{h=1}^{H} \mu^{(h)} \omega^{(h)} \cdot \mathbb{E}[\|p^{(h)}\|_2^2 \mid x_q]\right],$$

in the decomposition of the loss $\widetilde{\mathcal{L}}(\omega, \mu)$. Recall that $\|x_q\|_2^2 \sim \chi_d^2$. Thus, using the second moment of $\chi_d^2$, we have $\mathbb{E}\|x_q\|_2^4 = d \cdot (d+2)$. Now, applying the Cauchy-Schwartz inequality, we have

$$\begin{aligned}
&\sum_{h=1}^{H} \mu^{(h)} \omega^{(h)} \cdot \mathbb{E}\left[\|x_q\|_2^2 \cdot \mathbb{E}[\|p^{(h)}\|_2^2 \mid x_q]\right] \le \sum_{h=1}^{H} \mu^{(h)} \omega^{(h)} \cdot \sqrt{\mathbb{E}[\|x_q\|_2^4] \cdot \mathbb{E}\left[\left(\mathbb{E}[\|p^{(h)}\|_2^2 \mid x_q]\right)^2\right]} \\
&\lesssim d \cdot \sum_{h=1}^{H} \mu^{(h)} \omega^{(h)} \cdot \left(\sqrt{\mathbb{E}\left[\left(\mathbb{E}[\|p^{(h)}\|_2^2 \mid x_q] - L^{-1} \cdot \exp(d\omega^{(h),2})\right)^2\right]} + L^{-1} \cdot \exp(d\omega^{(h),2})\right).
\end{aligned}$$

Here, in the last inequality, we use $\sqrt{\mathbb{E}[X^2]} \leq \sqrt{\mathbb{E}[(X-a)^2]} + |a|$, where $X$ is any random variable and $a$ is a constant. Recall that we use $a \lesssim b$ to denote that $a \leq C \cdot b$ for some absolute constant $C > 0$. By applying Lemma E.3-(iii) with $\tau = 5\kappa^2$, we obtain the following inequality:

$$\mathbb{E}\big[\big(\mathbb{E}[\|p^{(h)}\|_2^2 \mid x_q] - L^{-1} \cdot \exp(d\omega^{(h),2})\big)^2\big] \leq C_2 \cdot L^{-3(1-\epsilon)},$$

where $C_2$ is an absolute constant and $\epsilon$ is defined in (E.20). In addition, under the scaling condition (C1), for all $h \in [H]$, we have $\exp(d\omega^{(h)^2}) \leq L^{\kappa^2}$. Therefore, we conclude that

$$\sum_{h=1}^H \mu^{(h)}\omega^{(h)} \cdot \mathbb{E}\big[\|x_q\|_2^2 \cdot \mathbb{E}[\|p^{(h)}\|_2^2 \mid x_q]\big] \lesssim \|\mu\|_\infty \cdot \|\omega\|_\infty \cdot dH \cdot \big(L^{-(3-\epsilon)/2} + L^{-1+\kappa^2}\big)$$

$$\leq \|\mu\|_\infty \cdot dH \cdot \big(L^{-(3-\epsilon)/2} + L^{-1+\kappa^2}\big), \tag{E.26}$$

where the last inequality follows from the fact that $\kappa \leq 1$.

**Step 3.3. Bound High-order Terms $\mathbb{E}[\|x_q\|_2^2 \cdot \mathcal{T}_{h,h'}(x_q)]$ under large $L$.**   Recall that we define high-order terms $\mathcal{T}_{h,h'}(x_q)$ as in (E.11), which comes from the decomposition of $\mathbb{E}[p^{(h')^\top} XX^\top p^{(h)} \mid x_q]$. As shown in (E.11), $\mathcal{T}_{h,h'}(x_q)$ is composed of terms such as

$$\mathbb{E}[p^\top \widetilde{p}^{\odot 2} \mid x_q], \qquad \mathbb{E}[\|p\|_2^2 \cdot p^\top \widetilde{p} \mid x_q], \qquad \text{and} \qquad \mathbb{E}[(p^\top \widetilde{p})^2 \mid x_q],$$

where we denote $\{p, \widetilde{p}\} = \{p^{(h)}, p^{(h')}\}$. Also notice that, by (E.11), each term in $\mathcal{T}_{h,h'}(x_q)$ involves has multiplicative factors $(\omega^{(h)})^2$, or $(\omega^{(h')})^2$, or $\omega^{(h)}\omega^{(h')}$. Following a similar argument as in **Step 3.2**, we apply Cauchy-Schwartz inequality to bound the high-order terms as

$$\sum_{h=1}^H \sum_{h'=1}^H \mu^{(h)}\mu^{(h')} \cdot \mathbb{E}[\|x_q\|_2^2 \cdot \mathcal{T}_{h,h'}(x_q)] \lesssim d \cdot \sum_{h=1}^H \mu^{(h)}\mu^{(h')} \sqrt{\mathbb{E}\big[\big(\big[\mathbb{E}[\mathcal{T}_{h,h'}(x_q) \mid x_q]\big)^2\big]}. \tag{E.27}$$

Then we apply Lemma E.3-(ii) with $\tau = 5\kappa^2$ to the terms in (E.27), which implies that

$$[\mathbb{E}[\mathcal{T}_{h,h'}(x_q) \mid x_q] \leq C_3 \cdot \|\omega\|_\infty^2 \cdot L^{-2(1-\epsilon)} \lesssim L^{-2(1-\epsilon)},$$

where $\epsilon$ is defined in (E.20). By (E.27), we have

$$\sum_{h=1}^H \sum_{h'=1}^H \mu^{(h)}\mu^{(h')} \cdot \mathbb{E}[\|x_q\|_2^2 \cdot \mathcal{T}_{h,h'}(x_q)] \lesssim \|\mu\|_\infty^2 \cdot dH^2 \cdot L^{-2(1-\epsilon)}. \tag{E.28}$$

This inequality establishes an upper bound on the last term in (E.12).

STEP 4. COMBINE EVERYTHING AND CONCLUDE THE PROOF.

Now we take the expectation with respect to $x_q$ in (E.12) and get

$$\mathcal{L}(\omega, \mu) = 1 + \sigma^2 - \underbrace{\frac{2\mathbb{E}[\|x_q\|_2^2]}{d} \cdot \sum_{h=1}^H \mu^{(h)}\omega^{(h)}}_{\text{(E.13)}} + \underbrace{\frac{\mathbb{E}[\|x_q\|_2^2]}{d} \cdot \sum_{h=1}^H \sum_{h'=1}^H \mu^{(h)}\mu^{(h')}\omega^{(h)}\omega^{(h')}}_{\text{(E.13)}}$$

$$+ \underbrace{(1 + \sigma^2) \cdot \sum_{h=1}^H \sum_{h'=1}^H \mu^{(h)}\mu^{(h')} \cdot \mathbb{E}[p^{(h)^\top} p^{(h')}]}_{\text{(E.25)}} + \underbrace{\sum_{h=1}^H \mu^{(h)}\omega^{(h)} \cdot \mathbb{E}\left[\frac{2\|x_q\|_2^2}{d} \cdot \mathbb{E}[\|p^{(h)}\|_2^2 \mid x_q]\right]}_{\text{(E.26)}}$$

$$+ \underbrace{\sum_{h=1}^H \sum_{h'=1}^H \mu^{(h)}\mu^{(h')} \cdot \mathbb{E}\left[\frac{\|x_q\|_2^2}{d} \cdot \mathcal{T}_{h,h'}(x_q)\right]}_{\text{(E.28)}}.$$

Here we list the number of inequalities that bound each term. Combining (E.13), (E.25), (E.26) and (E.28), we have

$$\mathcal{L}(\omega, \mu) = 1 + \sigma^2 - 2\mu^\top \omega + (\mu^\top \omega^2) + \frac{1}{L} \sum_{h=1}^{H} \sum_{h'=1}^{H} \mu^{(h)} \mu^{(h')} \cdot \exp(d \cdot \omega^{(h)} \omega^{(h')}) + \mathsf{Err}_\kappa$$

$$= 1 + \sigma^2 - 2\mu^\top \omega + \mu^\top \Big( \omega \omega^\top + (1 + \sigma^2) \cdot L^{-1} \cdot \exp(d\omega\omega^\top) \Big) \mu + \mathsf{Err}_\kappa,$$

where we error term $\mathsf{Err}_\kappa$ is bounded via

$$\mathsf{Err}_\kappa \leq \|\mu\|_\infty^2 \cdot H^2 \cdot \big( L^{-2(1-\epsilon)} + L^{-1+5\kappa^2} \big) + \|\mu\|_\infty \cdot dH \cdot \big( L^{-(3-\epsilon)/2} + L^{-1+\kappa^2} \big)$$

$$+ \|\mu\|_\infty^2 \cdot dH^2 \cdot L^{-2(1-\epsilon)}$$

$$\lesssim \max\{\|\mu\|_\infty, \|\mu\|_\infty^2\} \cdot dH^2 \cdot \max\big\{ L^{-2(1-\epsilon)}, L^{-(3-\epsilon)/2}, L^{-(1-5\kappa^2)} \big\}. \tag{E.29}$$

Here $\epsilon$ is defined in (E.20) and $\kappa$ is a small constant. By taking $\kappa = 0.1$ such that $\epsilon \approx 0.69 \leq 0.7$, the approximation error can be further simplified as $\mathsf{Err}_\kappa \lesssim \max\{\|\mu\|_\infty, \|\mu\|_\infty^2\} \cdot dH^2 \cdot L^{-2(1-\epsilon)} \leq \max\{\|\mu\|_\infty, \|\mu\|_\infty^2\} \cdot dH^2 \cdot L^{-3/5}$. Hence, for a given error level $\lambda > 0$, if we assume $\mu \in \mathbb{R}^H$ satisfies

$$\textbf{(C2)} \qquad \max\{\|\mu\|_\infty, \|\mu\|_\infty^2\} \lesssim L^{-\lambda/2 + 3/10},$$

then we can control the error by $\mathsf{Err}_\kappa = O(dH^2 \cdot L^{-\lambda})$. This completes the proof. □

## F. Proof of Theorem 4.2

In this section, we prove Theorem 4.2, the main result of this paper. We present separate proofs for each part of the theorem in the following subsections. Recall that we use $\theta = (\omega, \mu) \in \mathbb{R}^{2H}$ to parameterize the multi-head attention model with $H$ heads, where the weight matrices satisfy (3.1). Furthermore, recall that we define the parameter spaces $\mathscr{S}^*$ and $\bar{\mathscr{S}}$ in (4.5) and (4.6), respectively, with $\mathscr{S}^* \subseteq \bar{\mathscr{S}}$. We define $\sum_{h \in \mathcal{H}_+} \mu^{(h)} = \mu_+$ and $\sum_{h \in \mathcal{H}_-} \mu^{(h)} = \mu_-$ to simplify the notation. Also recall that we define the debiased GD predictor with learning $\eta$ as

$$\widehat{y}_q^{\mathsf{gd}}(\eta) = \frac{\eta}{L} \cdot \sum_{\ell=1}^{L} y_\ell \cdot \bar{x}_\ell^\top x_q, \quad \text{with} \quad \bar{x}_\ell = x_\ell - \frac{1}{L} \sum_{\ell=1}^{L} x_\ell. \tag{F.1}$$

Note that here we use $\bar{x}_\ell$ to denote the centered covariate.

In the following, we present a proof of Theorem 4.2. In §F.1, we show that the multi-head attention model can approximate the debiased GD predictor with a small error. In §F.2, we show that minimizing the approximate loss in (4.1) over the parameter space $\bar{\mathscr{S}}$ leads to a multi-head attention model that converges to the debiased GD predictor. In §F.3, we show that the debiased GD predictor is asymptotically Bayes optimal up to a proportionality factor.

### F.1. Proof of Theorem 4.2-(i): Approximation

*Proof of Theorem 4.2-(i).* In this proof, we regard the learning rate $\eta$ as a constant. Let $\gamma > 0$ be a scaling parameter and let $\breve{\mu}$ be defined such that $2\breve{\mu}\gamma = \eta$. Since we assumed that $\mu_+ = -\mu_- = \breve{\mu}$ with $\breve{\mu} > 0$ and $(\omega, \mu) \in \bar{\mathscr{S}}$, the multi-head attention estimator takes the form

$$\widehat{y}_q(\theta) = \breve{\mu} \cdot \sum_{\ell=1}^{L} \frac{y_\ell \cdot \exp(\gamma \cdot x_\ell^\top x_q)}{\sum_{\ell=1}^{L} \exp(\gamma \cdot x_\ell^\top x_q)} - \breve{\mu} \cdot \sum_{\ell=1}^{L} \frac{y_\ell \cdot \exp(-\gamma \cdot x_\ell^\top x_q)}{\sum_{\ell=1}^{L} \exp(-\gamma \cdot x_\ell^\top x_q)}, \tag{F.2}$$

and the debiased GD predictor is defined in (F.1) with $\eta = 2\gamma\breve{\mu}$. Following this, we the decompose difference between transformer $\widehat{y}_q(\theta)$ and debiased GD $\widehat{y}_q^{\mathsf{gd}}(\eta)$ as

$$\breve{\mu}^{-1} \cdot \{\widehat{y}_q(\theta) - \widehat{y}_q^{\mathsf{gd}}(\eta)\}$$

$$= \frac{1}{L} \cdot \sum_{\ell=1}^{L} \big( \exp(\gamma \cdot x_\ell^\top x_q) - \exp(-\gamma \cdot x_\ell^\top x_q) - 2\gamma \cdot x_\ell^\top x_q \big) \cdot y_\ell$$

$$+ \frac{L - \sum_{\ell=1}^{L} \exp(\gamma \cdot x_\ell^\top x_q)}{L \cdot \sum_{\ell=1}^{L} \exp(\gamma \cdot x_\ell^\top x_q)} \cdot \sum_{\ell=1}^{L} \exp(\gamma \cdot x_\ell^\top x_q) \cdot y_\ell + \frac{\gamma}{L} \cdot \sum_{\ell=1}^{L} \bar{x}^\top x_q \cdot y_\ell$$

$$+ \frac{\sum_{\ell=1}^{L} \exp(-\gamma \cdot x_\ell^\top x_q) - L}{L \cdot \sum_{\ell=1}^{L} \exp(-\gamma \cdot x_\ell^\top x_q)} \cdot \sum_{\ell=1}^{L} \exp(-\gamma \cdot x_\ell^\top x_q) \cdot y_\ell + \frac{\gamma}{L} \cdot \sum_{\ell=1}^{L} \bar{x}^\top x_q \cdot y_\ell$$

$$= \text{(i)} + \text{(ii)} + \text{(iii)}. \tag{F.3}$$

Here we define the three terms as **(i)**, **(ii)** and **(iii)** to simplify the notation. Before delving into details, we first define three auxiliary functions over $(\gamma, x) \in \mathbb{R}^2$ as below:

$$\phi_1(\gamma, x) = \exp(\gamma x) - \exp(-\gamma x) - 2\gamma x, \quad \phi_2(\gamma, x) = \exp(\gamma x) - 1 - \gamma x, \quad \psi(\gamma, x) = \exp(\gamma x) - 1.$$

By Taylor expansion, we can obtain that $\phi_1(\gamma, x)$ and $\phi_2(\gamma, x)$ are both $O(\gamma^2 x^2)$ when $\gamma \cdot x$ is small, and $\psi(\gamma, x) = O(\gamma x)$. We use $\phi_1$ to bound term **(i)** in (F.3) as follows:

$$\left| \text{(i)} \right| = \left| \frac{1}{L} \cdot \sum_{\ell=1}^{L} \phi_1(\gamma, x_\ell^\top x_q) \cdot y_\ell \right| \leq \max_{\ell \in [L]} |\phi_1(\gamma, x_\ell^\top x_q)| \cdot \frac{\|y\|_2}{\sqrt{L}} = \phi_1\left( \gamma, \max_{\ell \in [L]} |x_\ell^\top x_q| \right) \cdot \frac{\|y\|_2}{\sqrt{L}}. \tag{F.4}$$

Here the inequality results from the Cauchy-Schwartz inequality and the last equality follows from the fact that $\phi_1(\gamma, \cdot)$ is an even function and monotonically increasing for $\gamma > 0$. Next we consider the term **(ii)**. To simplify the notation, we define $\mathcal{S}_{\exp} = \sum_{\ell=1}^{L} \exp(\gamma \cdot x_\ell^\top x_q)$ and then we have

$$\sum_{\ell=1}^{L} -\phi_2(\gamma, x_\ell^\top x_q) = \sum_{\ell=1}^{L} \left( 1 + \gamma \cdot x_\ell^\top x_q - \exp(\gamma \cdot x_\ell^\top x_q) \right) = L + \gamma \cdot \sum_{\ell=1}^{L} x_\ell^\top x_q - \mathcal{S}_{\exp}. \tag{F.5}$$

Based on the definition of term **(ii)** in (F.3), we have

$$\text{(ii)} = \frac{L - \mathcal{S}_{\exp}}{L \cdot \mathcal{S}_{\exp}} \cdot \left( \sum_{\ell=1}^{L} \exp(\gamma \cdot x_\ell^\top x_q) \cdot y_\ell \right) + \frac{\gamma}{L} \cdot \sum_{\ell=1}^{L} \bar{x}^\top x_q \cdot y_\ell$$

$$= \frac{1}{L \cdot \mathcal{S}_{\exp}} \cdot \left( \sum_{\ell=1}^{L} -\phi_2(\gamma, x_\ell^\top x_q) - \gamma \cdot \sum_{\ell=1}^{L} x_\ell^\top x_q \right) \cdot \left( \sum_{\ell=1}^{L} \exp(\gamma \cdot x_\ell^\top x_q) \cdot y_\ell \right) + \frac{\gamma}{L} \cdot \sum_{\ell=1}^{L} \bar{x}^\top x_q \cdot y_\ell$$

$$= \frac{1}{L \cdot \mathcal{S}_{\exp}} \cdot \left( \sum_{\ell=1}^{L} -\phi_2(\gamma, x_\ell^\top x_q) \right) \cdot \left( \sum_{\ell=1}^{L} \exp(\gamma \cdot x_\ell^\top x_q) \cdot y_\ell \right)$$

$$- \frac{\gamma}{L \cdot \mathcal{S}_{\exp}} \cdot \left( \sum_{\ell=1}^{L} x_\ell^\top x_q \right) \cdot \left( \sum_{\ell=1}^{L} \exp(\gamma \cdot x_\ell^\top x_q) \cdot y_\ell \right) + \frac{\gamma}{L} \cdot \sum_{\ell=1}^{L} \bar{x}^\top x_q \cdot y_\ell. \tag{F.6}$$

Here, the first equality follows from (F.5). To simplify the notation, we define

$$\text{(ii.a)} = \left( \frac{1}{L} \sum_{\ell=1}^{L} -\phi_2(\gamma, x_\ell^\top x_q) \right) \cdot \left( \frac{1}{L} \sum_{\ell=1}^{L} \exp(\gamma \cdot x_\ell^\top x_q) \cdot y_\ell \right), \quad \text{(ii.b)} = \frac{1}{L} \sum_{\ell=1}^{L} \psi(\gamma, x_\ell^\top x_q) \cdot y_\ell.$$

Then, the first term in (F.6) corresponds to $L/\mathcal{S}_{\exp} \cdot$ **(ii.a)**. Furthermore, to handle the last two terms in (F.6), we denote $\bar{x} = 1/L \cdot \sum_{\ell=1}^{L} x_\ell$ and $\bar{y} = 1/L \cdot \sum_{\ell=1}^{L} y_\ell$. Then, we have

$$\frac{\gamma}{L \cdot \mathcal{S}_{\exp}} \cdot \left( \sum_{\ell=1}^{L} x_\ell^\top x_q \right) \cdot \left( \sum_{\ell=1}^{L} \exp(\gamma \cdot x_\ell^\top x_q) \cdot y_\ell \right) - \frac{\gamma}{L} \cdot \sum_{\ell=1}^{L} \bar{x}^\top x_q \cdot y_\ell$$

$$= \gamma \cdot \bar{x}^\top x_q \cdot \left( L/\mathcal{S}_{\exp} \cdot \text{(ii.b)} + (1 - L/\mathcal{S}_{\exp}) \cdot \bar{y} \right), \tag{F.7}$$

Combining (F.6) and (F.7), we can upper bound term **(ii)** as follows:

$$\left| \text{(ii)} \right| \leq L/\mathcal{S}_{\exp} \cdot \left( |\text{(ii.a)}| + \gamma \cdot |\bar{x}^\top x_q| \cdot |\text{(ii.b)}| \right) + \gamma \cdot |\bar{x}^\top x_q| \cdot (\mathcal{S}_{\exp}/L - 1) \cdot |\bar{y}|).$$

Furthermore, we can easily see that $\exp(\gamma \cdot x_\ell^\top x_q) \geq \mathbb{1}(x_\ell^\top x_q > 0)$ for all $\ell$, $\bar{x}^\top x_q \leq \max_{\ell \in [L]} |x_\ell^\top x_q|$ and $|\bar{y}| \leq \|y\|_2 / \sqrt{L}$. Thus, we can upper bound **(ii)** as

$$\left| \textbf{(ii)} \right| \leq \left( \frac{1}{L} \sum_{\ell=1}^L \mathbb{1}(x_\ell^\top x_q > 0) \right)^{-1} \cdot \left( \left| \textbf{(ii.a)} \right| + \gamma \cdot \max_{\ell \in [L]} |x_\ell^\top x_q| \cdot \left\{ \left| \textbf{(ii.b)} \right| + \left| \textbf{(ii.c)} \right| \cdot \frac{\|y\|_2}{\sqrt{L}} \right\} \right). \tag{F.8}$$

We first consider **(ii.a)**. Note the $\phi_2$ satisfies $|\phi_2(\gamma, x)| \leq \phi_2(\gamma, |x|)$, and for any fixed $\gamma > 0$, $\phi_2(\gamma, \cdot)$ is monotonically increasing in $x$ on $[0, \infty)$. Following a similar argument as in (F.4), we have

$$\left| \textbf{(ii.a)} \right| \leq \phi_2\left( \gamma, \max_{\ell \in [L]} |x_\ell^\top x_q| \right) \cdot \exp\left( \gamma \cdot \max_{\ell \in [L]} |x_\ell^\top x_q| \right) \cdot \frac{\|y\|_2}{\sqrt{L}}. \tag{F.9}$$

Similarly, for term **(ii.b)** and $\mathcal{S}_{\mathsf{exp}}/L - 1$, using the facts that $|\psi(\gamma, \cdot)| \leq \psi(\gamma, |\cdot|)$ and $\psi(\gamma, \cdot) > 0$ monotone increasing on $[0, \infty)$ for $\gamma > 0$, we have

$$\left| \textbf{(ii.b)} \right| \leq \psi\left( \gamma, \max_{\ell \in [L]} |x_\ell^\top x_q| \right) \cdot \frac{\|y\|_2}{\sqrt{L}}, \quad |\mathcal{S}_{\mathsf{exp}}/L - 1| = 1/L \cdot |\mathcal{S}_{\mathsf{exp}} - L| \leq \psi\left( \gamma, \max_{\ell \in [L]} |x_\ell^\top x_q| \right). \tag{F.10}$$

Therefore, combining (F.8), (F.9) and (F.10), we obtain

$$\left| \textbf{(ii)} \right| \leq \left( \frac{1}{L} \sum_{\ell=1}^L \mathbb{1}(x_\ell^\top x_q > 0) \right)^{-1} \cdot \left\{ \phi_2\left( \gamma, \max_{\ell \in [L]} |x_\ell^\top x_q| \right) \cdot \exp\left( \gamma \cdot \max_{\ell \in [L]} |x_\ell^\top x_q| \right) \cdot \frac{\|y\|_2}{\sqrt{L}} \right.$$
$$\left. + 2\gamma \cdot \psi\left( \gamma, \max_{\ell \in [L]} |x_\ell^\top x_q| \right) \cdot \max_{\ell \in [L]} |x_\ell^\top x_q| \cdot \frac{\|y\|_2}{\sqrt{L}} \right\}. \tag{F.11}$$

For term **(iii)** in (F.3), by symmetry, we can derive a similar bound as in (F.11) by changing the parameter $\gamma$ to $-\gamma$. Note that $|\phi_2(-\gamma, \cdot)| \leq \phi_2(\gamma, |\cdot|)$ and $|\psi(-\gamma, \cdot)| \leq \psi(\gamma, |\cdot|)$ given $\gamma > 0$. In addition, $\exp(-\gamma \cdot x_\ell^\top x_q) \geq \mathbb{1}(x_\ell^\top x_q < 0)$ for all $\ell \in [L]$. Thus, similar to (F.11), we have

$$\textbf{(iii)} \leq \left( \frac{1}{L} \sum_{\ell=1}^L \mathbb{1}(x_\ell^\top x_q \leq 0) \right)^{-1} \cdot \left\{ \phi_2\left( \gamma, \max_{\ell \in [L]} |x_\ell^\top x_q| \right) \cdot \exp\left( \gamma \cdot \max_{\ell \in [L]} |x_\ell^\top x_q| \right) \cdot \frac{\|y\|_2}{\sqrt{L}} \right.$$
$$\left. + 2\gamma \cdot \psi\left( \gamma, \max_{\ell \in [L]} |x_\ell^\top x_q| \right) \cdot \max_{\ell \in [L]} |x_\ell^\top x_q| \cdot \frac{\|y\|_2}{\sqrt{L}} \right\}, \tag{F.12}$$

Combining (F.3), (F.4), (F.11) and (F.12), we establish an upper bound on $\hat{y}_q(\theta) - \hat{y}_q^{\mathsf{gd}}(\eta)$ using $\breve{\mu} \asymp \gamma^{-1}$. To simplify the expression, we apply the concentration concentration results, stated in Lemma G.3. With probability at least $1 - \delta$, these three inequalities hold simultaneously:

$$\max_{\ell \in [L]} |x_\ell^\top x_q| \leq \sqrt{12d} \cdot \log(8L/\delta), \quad \frac{1}{L} \sum_{\ell=1}^L y_\ell^2 \leq 1 + \sigma^2 + \mathrm{poly}(d^{-1/2}, L^{-1/2}, \log(1/\delta)),$$
$$\left| \frac{1}{L} \sum_{\ell=1}^L \mathbb{1}(x_\ell^\top x_q > 0) - \frac{1}{2} \right| \leq \min\left\{ \sqrt{\log(8/\delta)/L}, 1/2 \right\}. \tag{F.13}$$

When $L \geq 16 \cdot \log(8/\delta)$, we have $1/L \cdot \sum_{\ell=1}^L \mathbb{1}(x_\ell^\top x_q \leq 0) \in [0.25, 0.75]$, with high probability. When this is the case, $1/L \cdot \sum_{\ell=1}^L \mathbb{1}(x_\ell^\top x_q > 0)$ also lies in $[0.25, 0.75]$. Following (F.13) and the upper bound on **(i)** in (F.4), we have

$$\gamma^{-1} \cdot \left| \textbf{(i)} \right| \lesssim \sqrt{1 + \sigma^2} \cdot \gamma^{-1} \cdot \phi_1\left( \gamma, \max_{\ell \in [L]} |x_\ell^\top x_q| \right)$$
$$\asymp \sqrt{1 + \sigma^2} \cdot \gamma \cdot \max_{\ell \in [L]} |x_\ell^\top x_q|^2 = O\left( \sqrt{1 + \sigma^2} \cdot \gamma \cdot d \cdot \log^2(8L/\delta) \right), \tag{F.14}$$

where $O(\cdot)$ hides constant terms that are independent of $d$, $L$, and $\delta$. Now, we consider the case where $\gamma$ is a sufficiently small constant such that $\gamma \leq C_1 \cdot (\sqrt{d} \cdot \log(L/\delta))^{-1}$ and thus $\exp\left( \gamma \cdot \max_{\ell \in [L]} |x_\ell^\top x_q| \right) \lesssim 1$. Similarly, using the upper bound on **(ii)** given in (F.11), we have

$$\gamma^{-1} \cdot \left| \textbf{(ii)} \right| \lesssim \sqrt{1 + \sigma^2} \cdot \gamma^{-1} \cdot \phi_2\left( \gamma, \max_{\ell \in [L]} |x_\ell^\top x_q| \right) \cdot \exp\left( \gamma \cdot \max_{\ell \in [L]} |x_\ell^\top x_q| \right)$$

$$+ \sqrt{1+\sigma^2} \cdot 2\psi\left(\gamma, \max_{\ell \in [L]} |x_\ell^\top x_q|\right) \cdot \max_{\ell \in [L]} |x_\ell^\top x_q|$$

$$\asymp \sqrt{1+\sigma^2} \cdot \gamma \cdot \max_{\ell \in [L]} |x_\ell^\top x_q|^2 = O\left(\sqrt{1+\sigma^2} \cdot \gamma \cdot d \cdot \log^2(8L/\delta)\right). \tag{F.15}$$

We can also get the same upper bound for **(iii)**, using (F.12). Recall that we let $\theta = (\omega, \mu)$ be the transformer parameter with $\omega = (\gamma, -\gamma)$ and $\mu = (\breve{\mu}, -\breve{\mu})$, where $\gamma, \breve{\mu} > 0$ with $\eta = 2\gamma \cdot \breve{\mu}$, which is a constant. Combining (F.3), (F.14) and (F.15), we conclude that

$$|\widehat{y}_q(\theta) - \widehat{y}_q^{\mathsf{gd}}(\eta)| \lesssim \gamma^{-1} \cdot \left(|\mathbf{(i)}| + |\mathbf{(ii)}| + |\mathbf{(iii)}|\right) = \widetilde{O}\left(\sqrt{1+\sigma^2} \cdot \gamma \cdot d\right), \tag{F.16}$$

where $\widetilde{O}(\cdot)$ omits logarithmic terms. Therefore, we complete the proof. $\qquad\square$

**Remark F.1.** *Here, we also provide a heuristic but simpler derivation of the debiased GD predictor using first-order Taylor expansion expansion. Note that*

$$\sum_{\ell=1}^L \frac{y_\ell \cdot \exp(\gamma \cdot x_\ell^\top x_q)}{\sum_{\ell=1}^L \exp(\gamma \cdot x_\ell^\top x_q)} \approx \sum_{\ell=1}^L \frac{y_\ell \cdot (1 + \gamma \cdot x_\ell^\top x_q)}{L + \sum_{\ell=1}^L \gamma \cdot x_\ell^\top x_q} \approx \sum_{\ell=1}^L \frac{y_\ell}{L} \cdot \left(1 + \gamma \cdot x_\ell^\top x_q - \frac{\gamma}{L}\sum_{\ell=1}^L x_\ell^\top x_q\right),$$

*where the second approximation results from the first-order approximation of reciprocal and ignoring the higher-order $O(\gamma^2)$ terms. Similarly, we can show that*

$$\sum_{\ell=1}^L \frac{y_\ell \cdot \exp(-\gamma \cdot x_\ell^\top x_q)}{\sum_{\ell=1}^L \exp(-\gamma \cdot x_\ell^\top x_q)} \approx \sum_{\ell=1}^L \frac{y_\ell}{L} \cdot \left(1 - \gamma \cdot x_\ell^\top x_q + \frac{\gamma}{L}\sum_{\ell=1}^L x_\ell^\top x_q\right).$$

*By subtracting the two terms above, we can get the desired form of approximation, i.e., $\widehat{y}_q(\theta) \approx \widehat{y}_q^{\mathsf{gd}}(\eta)$, where $\eta = 2\breve{\mu}\gamma$ and $\widehat{y}_q(\theta)$ is given in (F.2).*

### F.2. Proof of Theorem 4.2-(ii): Optimality

*Proof of Theorem 4.2-(ii).* Recall that for fixed $\gamma > 0$, we consider the parameter space defined as

$$\bar{\mathscr{S}} = \{(\omega, \mu) : \forall \gamma > 0, \ \omega^{(h)} = \gamma \cdot \operatorname{sign}(\omega^{(h)}) \text{ for all } h \in [H], \ \min\{|\mathcal{H}_+|, |\mathcal{H}_-|\} > 1\}.$$

For notational simplicity, we denote $\mu_+ = \sum_{h \in \mathcal{H}^+} \mu^{(h)}$, $\mu_- = \sum_{h \in \mathcal{H}^-} \mu^{(h)}$ and $\mu_d = \sum_{h \in [H] \backslash \mathcal{H}_+ \cup \mathcal{H}_-} \mu^{(h)}$, respectively as the sum of OV parameters for the positive, negative and dummy heads. With this parameterization, the transformer estimator takes the form

$$\widehat{y}_q(\theta) = \mu_+ \cdot \sum_{\ell=1}^L \frac{y_\ell \cdot \exp(\gamma \cdot x_\ell^\top x_q)}{\sum_{\ell=1}^L \exp(\gamma \cdot x_\ell^\top x_q)} + \mu_- \cdot \sum_{\ell=1}^L \frac{y_\ell \cdot \exp(-\gamma \cdot x_\ell^\top x_q)}{\sum_{\ell=1}^L \exp(-\gamma \cdot x_\ell^\top x_q)} + \mu_d \cdot \bar{y},$$

and thus it suffices to consider three-head attention with $\omega = (\gamma, -\gamma, 0)$ and $\mu = (\mu_+, \mu_-, \mu_d)$. Recall that we introduce the approximate loss $\widetilde{\mathcal{L}}(\omega, \mu)$ in (4.1), which takes the following form:

$$\begin{aligned}\widetilde{\mathcal{L}}(\omega, \mu) &= 1 - 2\mu^\top \omega + \mu^\top\left(\omega\omega^\top + (1+\sigma^2) \cdot L^{-1} \cdot \exp(d\omega\omega^\top)\right)\mu \\ &= (1 - \mu^\top \omega)^2 + (1+\sigma^2) \cdot L^{-1} \cdot \mu^\top \exp(d\omega\omega^\top)\mu.\end{aligned}$$

Consider the problem of minimizing this loss function $\min_{\omega, \mu} \widetilde{\mathcal{L}}(\omega, \mu)$. For notational simplicity, we let $\omega_a = (\gamma, -\gamma)$ and $\mu_a = (\mu_+, \mu_-)$ denote the parameters of active heads and let $\mu_d$ denote the OV parameters of dummy heads with $\omega_d = 0$. Then, we can write the loss as

$$\widetilde{\mathcal{L}}(\omega_a, \mu_a, \mu_d) = (1 - \mu_a^\top \omega_a)^2 + (1+\sigma^2) \cdot L^{-1} \cdot \left(\mu_a^\top \exp(d\omega_a\omega_a^\top)\mu_a + \mu_d^2 + 2\mu_d \cdot \mu_a^\top \mathbf{1}\right), \tag{F.17}$$

where the last term $2\mu_d \cdot \mu_a^\top \mathbf{1}$ is due to the interaction between the OV of dummy heads and the OV of active heads. In the following, we find the minimizer of $\widetilde{\mathcal{L}}(\omega_a, \mu_a, \mu_d)$. We first optimize $\mu_d \in \mathbb{R}$ with $\omega_a$ and $\mu_a$ fixed. Note that $\widetilde{\mathcal{L}}$ is quadratic

with respect to $\mu_d$. By direct calculation, we obtain that the optimal value of $\mu_d$ is $\mu_d^* = -\mu_a^\top \mathbf{1}$. Plugging $\mu_d^*$ in to the loss $\widetilde{\mathcal{L}}$ in (F.17), we have

$$\widetilde{\mathcal{L}}(\omega_a, \mu_a, \mu_d^*) = (1 - \mu_a^\top \omega_a)^2 + (1 + \sigma^2) \cdot L^{-1} \cdot \left( \mu_a^\top \exp(d\omega_a \omega_a^\top) \mu_a - (\mu_a^\top \mathbf{1})^2 \right). \tag{F.18}$$

With slight abuse of notation, we write $\widetilde{\mathcal{L}}(\omega_a, \mu_a, \mu_d^*)$ as $\widetilde{\mathcal{L}}(\omega_a, \mu_a)$ in the sequel. By rearranging the terms, we can rewrite the above loss as a function of $\omega_a$ and $\mu_a$:

$$\widetilde{\mathcal{L}}(\omega_a, \mu_a) = 1 - 2\omega_a^\top \mu_a + \mu_a^\top \left( \omega_a \omega_a^\top + (1 + \sigma^2) \cdot L^{-1} \cdot \left( \exp(d\,\omega_a \omega_a^\top) - \mathbf{1}\mathbf{1}^\top \right) \right) \mu_a. \tag{F.19}$$

We note that the two-dimensional vector $\omega_a$ is orthogonal to $\mathbf{1} = (1, 1)$. Moreover, $\omega_a$ and $\mathbf{1}$ form an unnormalized orthogonal basis of $\mathbb{R}^2$. To simplify the notation, we define $A(\omega_a) = \omega_a \omega_a^\top + (1 + \sigma^2) \cdot L^{-1} \cdot (\exp(d\omega_a \omega_a^\top) - \mathbf{1}\mathbf{1}^\top)$. Let us decompose $\exp(d\omega_a \omega_a^\top)$ into

$$\exp(d\omega_a \omega_a^\top) = \frac{\exp(d\gamma^2) + \exp(-d\gamma^2)}{2} \cdot \mathbf{1}\mathbf{1}^\top + \frac{\exp(d\gamma^2) - \exp(-d\gamma^2)}{2\gamma^2} \cdot \omega_a \omega_a^\top$$

$$= \cosh(d\gamma^2) \cdot \mathbf{1}\mathbf{1}^\top + \sinh(d\gamma^2) \cdot \frac{\omega_a \omega_a^\top}{\gamma^2}.$$

Thus, $A(\omega_a)$ allows the following rank-one decomposition:

$$A(\omega_a) = \omega_a \omega_a^\top + (1 + \sigma^2) \cdot L^{-1} \cdot \left( \cosh(d\gamma^2) \cdot \mathbf{1}\mathbf{1}^\top + \sinh(d\gamma^2) \cdot \frac{\omega_a \omega_a^\top}{\gamma^2} - \mathbf{1}\mathbf{1}^\top \right)$$

$$= \left( 1 + (1 + \sigma^2) \cdot L^{-1} \cdot \sinh(d\gamma^2) \cdot \frac{1}{\gamma^2} \right) \cdot \omega_a \omega_a^\top + (1 + \sigma^2) \cdot L^{-1} \cdot \left( \cosh(d\gamma^2) - 1 \right) \cdot \mathbf{1}\mathbf{1}^\top.$$

In other words, $\omega_a$ and $\mathbf{1}$ are just the (unnormalized) eigenvectors of $A(\omega_a)$. In particular, since $\sinh(x) \geq 0$ and $\cosh(x) - 1 \geq 0$ for all $x \geq 0$, the matrix $A(\omega_a)$ is positive semi-definite. Thus, minimizing the objective in (F.19) is just a convex optimization problem. Optimizing this quadratic function over $\mu_a$ with $\omega_a$ fixed, the optimal value of $\mu_a$, denoted by $\mu_a^*(\omega_a)$, is given by

$$\mu_a^*(\omega_a) = A(\omega_a)^{-1} \omega_a. \tag{F.20}$$

Plugging $\mu_a^*(\omega_a)$ in (F.20) into $\widetilde{\mathcal{L}}$, we obtain the a closed-form expression of the optimal loss with respect to $\omega_a$:

$$\widetilde{\mathcal{L}}(\omega_a) = \widetilde{\mathcal{L}}(\omega_a, \mu_a^*(\omega_a), \mu_d^*) = 1 - \omega_a^\top A(\omega_a)^{-1} \omega_a. \tag{F.21}$$

Here we abuse the notation slightly by writing the loss function as a function of $\omega_a$ only.

From the discussion above, we see that the inverse of $A(\omega_a)$ is given by

$$A(\omega_a)^{-1} = \left( 1 + (1 + \sigma^2) \cdot L^{-1} \cdot \sinh(d\gamma^2) \cdot \frac{1}{\gamma^2} \right)^{-1} \cdot \frac{\omega_a \omega_a^\top}{\|\omega_a\|_2^4} + C \cdot \mathbf{1}\mathbf{1}^\top, \tag{F.22}$$

where $C_\gamma > 0$ is a number depending $\gamma$. Noting that $\omega_a$ is orthogonal to the all-one vector $\mathbf{1}$, combining (F.22), we can simplify (F.21) as

$$\widetilde{\mathcal{L}}(\omega_a) = 1 - \left( 1 + (1 + \sigma^2) \cdot L^{-1} \cdot \sinh(d\gamma^2) \cdot \frac{1}{\gamma^2} \right)^{-1}.$$

Consequently, minimizing the loss $\widetilde{\mathcal{L}}(\omega_a)$ is equivalent to finding the minimal value of $\sinh(d\gamma^2) \cdot (d\gamma^2)^{-1}$. By simple calculus, we can show that $\sinh(x)/x$ is monotonically increasing on $\mathbb{R}^+$, and thus the minimum value of $\sinh(d\gamma^2) \cdot (d\gamma^2)^{-1}$ is achieved at $\gamma^* = 0^+$. Then we observe that the loss $\widetilde{\mathcal{L}}(\omega_a)$ is a monotonically increasing function of $\gamma$, and the optimal value of $\omega_a$ is $\omega_a^* = (-\gamma^*, \gamma^*)$. Therefore, the optimal value of the loss is given by

$$\widetilde{\mathcal{L}}^* = 1 - \frac{1}{1 + (1 + \sigma^2) \cdot d/L} = \frac{(1 + \sigma^2) \cdot d}{(1 + \sigma^2) \cdot d + L}. \tag{F.23}$$

In addition, the optimal choice of $\mu_a$ and $\mu_d$ under the limit $\gamma \to 0^+$ satisfy

$$
\begin{aligned}
\lim_{\gamma \to 0^+} \langle \mu_a^*(\omega_a^*), \omega_a^* \rangle &= \lim_{\gamma \to 0^+} (\omega_a^*)^\top A(\omega_a^*)^{-1} \omega_a^* \\
&= \lim_{\gamma \to 0^+} \left( 1 + (1+\sigma^2) \cdot L^{-1} \cdot \sinh(d\gamma^2) \cdot \frac{1}{\gamma^2} \right)^{-1} \cdot \frac{((\omega_a^*)^\top \omega_a^*)^2}{\|\omega_a^*\|_2^4} \\
&= \left( 1 + (1+\sigma^2) \cdot L^{-1} \cdot d \right)^{-1}, \quad\quad\quad\quad\quad\quad\quad\quad\quad \text{(F.24)} \\
\lim_{\gamma \to 0^+} \mu_d^*(\mu_a^*) &= \lim_{\gamma \to 0^+} \langle \mu_a^*, \mathbf{1} \rangle = \left( 1 + (1+\sigma^2) \cdot L^{-1} \cdot d \right)^{-1} \cdot \frac{\langle \omega_a^*, \mathbf{1} \rangle}{\|\omega_a^*\|_2^4} = 0,
\end{aligned}
$$

where $\mu_a^* = A(\omega_a^*)^{-1} \omega_a^*$ is parallel to $(-1, 1)$. Recall that we let $\theta_\gamma$ denote any element in $\mathscr{S}_\gamma$. That is, for any $h \in \mathcal{H}_+ \cup \mathcal{H}_-$, $|\omega^{(h)}| = \gamma$, $\mu_+ = -\mu_- = \mu_\gamma$, where $\mu_\gamma$ is defined in (??). Moreover, for any dummy head with $h \notin \mathcal{H}_+ \cup \mathcal{H}_-$, $\mu^{(h)} = 0$. Moreover, by the definition of $\mu_\gamma$, we have

$$
2\gamma \cdot \mu_\gamma = \frac{\gamma^2}{\gamma^2 + (1+\sigma^2) \cdot L^{-1} \cdot \cosh(d\gamma^2)} \to \frac{1}{1 + (1+\sigma^2) \cdot L^{-1} \cdot d}
$$

as $\gamma \to 0^+$. Therefore, by the construction of $\mu_a^*$ and (F.24) we have shown that the global minimizer of $\widetilde{\mathcal{L}}$ over $\mathscr{S}$ is attained in $\mathscr{S}_\gamma$ with $\gamma \to \gamma^* = 0^+$. In other words, the attention model with parameter $(\omega_a^*, \mu_a^*, \mu_d^*)$ is exactly the model with parameter $\theta_{\gamma^*}$.

Furthermore, recall that we have established an upper bound on the difference between transformer predictor and debiased GD predictor in the proof of Theorem 4.2-(i). Let $\eta^*$ denote $(1 + (1+\sigma^2) \cdot L^{-1} \cdot d)^{-1}$ and let $\eta_\gamma = 2\gamma \cdot \mu_\gamma$. Given a fixed input $\{(x_\ell, y_\ell)\}_{\ell \in [L]} \cup \{x_q\}$, as shown in (F.16), for any sufficiently small $\gamma > 0$, we have

$$
|\widehat{y}_q(\theta_\gamma) - \widehat{y}_q^{\text{gd}}(\eta_\gamma)| \leq \widetilde{O}\big(\sqrt{1+\sigma^2} \cdot \gamma \cdot d\big).
$$

Taking the limit $\gamma \to \gamma^* = 0^+$, we prove that the predictor that minimizes the approximate loss $\widetilde{\mathcal{L}}$ over $\mathscr{S}_\gamma$, $\widehat{y}_q(\theta_{\gamma^*})$, coincides with the debiased GD predictor $\widehat{y}_q^{\text{gd}}(\eta_\gamma^*)$. This completes the proof. $\qquad\square$

### F.3. Proof of Theorem 4.2-(iii): Bayes Risk

In Theorem 4.2-(i), we have shown that the difference between the transformer predictor and debiased GD predictor are close when $\theta = (\omega, \mu) \in \mathscr{S}$ and $\gamma$ is small. In the following, we show that the risk of the debiased GD predictor $\widehat{y}_q^{\text{gd}}(\eta^*)$ is comparable to the Bayes optimal predictor, where $\eta^*$ denotes the optimal learning rate. This implies that the transformer model approximately learns the Bayes optimal predictor.

*Proof of Theorem 4.2-(iii).* To prove this theorem, we use the vanilla gradient descent predictor as the bridge. In the first step, we characterize the risk of the vanilla GD predictor. Then, in the second step, we prove that the risk of debiased GD predictor is close to that of the vanilla GD predictor. Finally, in the last step, we establish the risk of the Bayes optimal predictor.

**Step 1: Risk of Vanilla Gradient Descent Predictor.** We define the predictor given by the vanilla gradient descent as follows:

$$
\widehat{y}_q^{\text{vgd}}(\eta) = \frac{\eta}{L} \cdot y^\top X x_q = \frac{\eta}{L} \cdot \sum_{\ell=1}^{L} y_\ell \cdot x_\ell^\top x_q,
$$

where $\eta > 0$ is the learning rate. We characterize $\mathcal{E}(\widehat{y}_q^{\text{vgd}}(\eta))$, the $\ell_2$-risk of vanilla GD predictor as follows. By direct calculation, we have

$$
\begin{aligned}
\mathbb{E}\big[(\widehat{y}_q^{\text{vgd}}(\eta) - y_q)^2\big] &= \mathbb{E}\left[ \left( \frac{\eta}{L} \cdot y^\top X x_q - \beta^\top x_q \right)^2 \right] + \sigma^2 = \mathbb{E}\left[ \left\| \frac{\eta}{L} \cdot X^\top (X\beta + \epsilon) - \beta \right\|_2^2 \right] + \sigma^2 \\
&= \underbrace{\mathbb{E}\left[ \left\| \left( \frac{\eta}{L} \cdot X^\top X - I_d \right) \beta \right\|_2^2 \right]}_{\text{(i)}} + \frac{\eta^2}{L^2} \cdot \underbrace{\mathbb{E}\left[ \|X^\top \epsilon\|_2^2 \right]}_{\text{(ii)}} + \sigma^2, \quad\quad\quad \text{(F.25)}
\end{aligned}
$$

where in the second equality, we take expectation over $x_q \sim \mathcal{N}(0, I_d)$. Furthermore, we have

$$\textbf{(i)} = \mathbb{E}\big[\, \|\beta\|_2^2 \,\big] - \frac{2\eta}{L} \cdot \mathbb{E}[\|X\beta\|_2^2] + \frac{\eta^2}{L^2} \cdot \mathbb{E}\Big[\big\|X^\top X\beta\big\|_2^2\Big]$$

$$= 1 - \frac{2\eta}{L} \cdot \sum_{\ell=1}^{L} \mathbb{E}[(x_\ell^\top \beta)^2] + \frac{\eta^2}{L^2} \cdot \sum_{\ell=1}^{L} \mathbb{E}\big[\|x_\ell\|_2^2 \cdot (x_\ell^\top \beta)^2\big]$$

$$+ \frac{2\eta^2}{L^2} \cdot \sum_{1 \le \ell \ne j \le L} \mathbb{E}\big[x_\ell^\top x_j \cdot \beta^\top x_j x_\ell^\top \beta\big],$$

where we use $\beta \sim \mathcal{N}(0, I_d/d)$. By moment calculations, we have $\mathbb{E}[(x_\ell^\top \beta)^2] = \mathbb{E}[\|\beta\|_2^2] = 1$ and

$$\mathbb{E}\big[\|x_\ell\|_2^2 \cdot (x_\ell^\top \beta)^2\big] = \frac{1}{d} \cdot \mathbb{E}\big[\|x_\ell\|_2^2 \cdot \mathrm{tr}(x_\ell x_\ell^\top)\big] = \frac{1}{d} \cdot \mathbb{E}\big[\|x_\ell\|_2^4\big] = d + 2,$$

$$\mathbb{E}\big[x_\ell^\top x_j \cdot \beta^\top x_j x_\ell^\top \beta\big] = \frac{1}{d} \cdot \mathbb{E}\big[x_\ell^\top x_j \cdot \mathrm{tr}(x_j x_\ell^\top)\big] = \frac{1}{d} \cdot \mathbb{E}[(x_\ell^\top x_j)^2] = 1,$$

for all $1 \le j \ne \ell \le L$. Combining the results above, we have

$$\textbf{(i)} = 1 - 2\eta + \frac{\eta^2}{L^2} \cdot L(d + 2) + \frac{2\eta^2}{L^2} \cdot \frac{L(L-1)}{2} = 1 - 2\eta + \frac{\eta^2}{L} \cdot (d + L + 1). \tag{F.26}$$

Moreover, based on the i.i.d assumption of the demonstration examples, we have

$$\textbf{(ii)} = \sum_{\ell=1}^{L} \mathbb{E}\big[\|\epsilon_\ell \cdot x_\ell\|_2^2\big] = L \cdot \mathbb{E}[\|x_\ell\|_2^2] \cdot \mathbb{E}[\epsilon_\ell^2] = dL \cdot \sigma^2. \tag{F.27}$$

Combine (F.25), (F.26) and (F.27), the $\ell_2$-risk of the vanilla GD predictor is given by

$$\mathbb{E}\big[\big(\widehat{y}_q^{\mathsf{vgd}}(\eta) - y_q\big)^2\big] = 1 + \sigma^2 - 2\eta + \eta^2/L \cdot (d(1 + \sigma^2) + L + 1).$$

This is a quadratic function of $\eta$, and the optimal learning rate that minimizes this risk is given by

$$\eta^{\mathsf{vgd},*} = L/(d(1 + \sigma^2) + L + 1). \tag{F.28}$$

The optimal $\ell_2$-risk is given by

$$\mathcal{E}\big(\widehat{y}_q^{\mathsf{gd}}(\eta^{\mathsf{vgd},*})\big) = 1 + \sigma^2 - \frac{L}{d(1 + \sigma^2) + L + 1}. \tag{F.29}$$

**Step 2: Risk of Debiased Gradient Descent Predictor.** Now we compare $\widehat{y}_q^{\mathsf{gd}}(\eta)$ and $\widehat{y}_q^{\mathsf{vgd}}(\eta)$ in terms of the $\ell_2$-risk. By direct calculation, we have

$$\mathbb{E}\big[\big(\widehat{y}_q^{\mathsf{gd}}(\eta) - y_q\big)^2\big] - \mathbb{E}\big[\big(\widehat{y}_q^{\mathsf{vgd}}(\eta) - y_q\big)^2\big]$$

$$= \mathbb{E}\big[\big(\widehat{y}_q^{\mathsf{vgd}}(\eta) - y^\top \mathbf{1}_L \cdot \mathbf{1}_L^\top X x_q/L^2 - y_q\big)^2\big] - \mathbb{E}\big[\big(\widehat{y}_q^{\mathsf{vgd}}(\eta) - y_q\big)^2\big]$$

$$= \frac{1}{L^4} \cdot \mathbb{E}\Big[\big(y^\top \mathbf{1}_L \cdot \mathbf{1}_L^\top X x_q\big)^2\Big] - \frac{2}{L^2} \cdot \mathbb{E}\Big[\Big(\frac{\eta}{L} \cdot y^\top X x_q - \beta^\top x_q\Big)\big(y^\top \mathbf{1}_L \cdot \mathbf{1}_L^\top X x_q\big)\Big]$$

$$= \frac{1}{L^4} \cdot \underbrace{\mathbb{E}\Big[\big\|X^\top \mathbf{1}_L \cdot \mathbf{1}_L^\top y\big\|_2^2\Big]}_{\textbf{(iii)}} - \frac{2\eta}{L^3} \cdot \underbrace{\mathbb{E}\big[y^\top \mathbf{1}_L \cdot \mathbf{1}_L^\top X X^\top y\big]}_{\textbf{(iv)}} + \frac{2}{L^2} \cdot \underbrace{\mathbb{E}\big[y^\top \mathbf{1}_L \cdot \mathbf{1}_L^\top X \beta\big]}_{\textbf{(v)}}.$$

In the sequel, we define $\bar{x}_L = X^\top \mathbf{1}_L$. Then, $\bar{x}_L \sim \mathcal{N}(0, L \cdot I_d)$. By direct calculation, we have

$$\textbf{(iii)} = \mathbb{E}\Big[\big\|X^\top \mathbf{1}_L \cdot \mathbf{1}_L^\top (X\beta + \epsilon)\big\|_2^2\Big] = \mathbb{E}\Big[\big\|X^\top \mathbf{1}_L \cdot \mathbf{1}_L^\top X\beta\big\|_2^2\Big] + \mathbb{E}\Big[\big\|X^\top \mathbf{1}_L \cdot \mathbf{1}_L^\top \epsilon\big\|_2^2\Big]$$

$$= \mathbb{E}\big[\big\|\bar{x}_L \cdot \bar{x}_L^\top \beta\big\|_2^2\big] + \mathbb{E}\big[(\epsilon^\top \mathbf{1}_L)^2\big] \cdot \mathbb{E}\,\|\bar{x}_L\|_2^2$$

$$= \frac{1}{d} \cdot \mathbb{E}\left[\|\bar{x}_L\|_2^4\right] + dL^2\sigma^2 = L^2 \cdot (d(1+\sigma^2) + 2). \tag{F.30}$$

Moreover, for term **(iv)**, we have

$$
\begin{aligned}
\textbf{(iv)} &= \mathbb{E}\left[(X\beta + \epsilon)^\top \mathbf{1}_L \cdot \mathbf{1}_L^\top X X^\top (X\beta + \epsilon)\right] \\
&= \mathbb{E}\left[\beta^\top X^\top \mathbf{1}_L \cdot \mathbf{1}_L^\top X X^\top X\beta\right] + \mathbb{E}\left[\epsilon^\top \mathbf{1}_L \cdot \mathbf{1}_L^\top X X^\top \epsilon\right] \\
&= \frac{1}{d} \cdot \mathbb{E}\left[\|X X^\top \mathbf{1}_L\|_2^2\right] + \sigma^2 \cdot \mathbb{E}\left[\|X^\top \mathbf{1}_L\|_2^2\right] = \frac{1}{d} \cdot \sum_{\ell=1}^{L} \mathbb{E}\left[(x_\ell^\top \bar{x}_L)^2\right] + \sigma^2 \cdot \mathbb{E}\left[\|\bar{x}_L\|_2^2\right],
\end{aligned}
$$

where the last equality results from $XX^\top \mathbf{1}_L = X\bar{x}_L$. Furthermore, by direct calculation, we have

$$\mathbb{E}[(x_\ell^\top \bar{x}_L)^2] = \mathbb{E}\|x_\ell\|_2^4 + \sum_{1 \le j \ne \ell \le L} \mathbb{E}\left[(x_\ell^\top x_j)^2\right] + \sum_{1 \le k \ne j \le L} \mathbb{E}\left[\|x_\ell\|_2^2 \cdot x_k^\top x_j\right] = d \cdot (d + L + 1).$$

Based on the arguments above, we obtain a closed-form expression of **(iv)**:

$$\textbf{(iv)} = (d + L + 1) \cdot L + dL \cdot \sigma^2. \tag{F.31}$$

Following a similar argument, for term **(v)**, we have

$$\textbf{(v)} = \mathbb{E}\left[(X\beta + \epsilon)^\top \mathbf{1}_L \cdot \mathbf{1}_L^\top X\beta\right] = \mathbb{E}\left[(\beta^\top X^\top \mathbf{1}_L)^2\right] = \frac{1}{d} \cdot \mathbb{E}\|\bar{x}_L\|_2^2 = L. \tag{F.32}$$

Combining (F.30), (F.31) and (F.32), we have

$$\mathbb{E}\left[\left(\widehat{y}_q^{\mathsf{gd}}(\eta) - y_q\right)^2\right] - \mathbb{E}\left[\left(\widehat{y}_q^{\mathsf{vgd}}(\eta) - y_q\right)^2\right] = \frac{d(1+\sigma^2) + 2 + 2L}{L^2} - \frac{2\eta}{L^2}\left(d(1+\sigma^2) + L + 1\right).$$

When $\eta$ is bounded by a constant, given $L \to \infty$ and $d/L \to \xi$, we know that

$$\mathbb{E}\left[\left(\widehat{y}_q^{\mathsf{gd}}(\eta) - y_q\right)^2\right] - \mathbb{E}\left[\left(\widehat{y}_q^{\mathsf{vgd}}(\eta) - y_q\right)^2\right] \to 0 \qquad \text{as } L \to \infty \text{ with } d/L \to \xi.$$

Hence, we have $\mathcal{E}(\widehat{y}_q^{\mathsf{gd}}(\eta)) \to \mathcal{E}(\widehat{y}_q^{\mathsf{vgd}}(\eta))$ uniformly for all bounded $\eta$. Thus, under the proportional regime, the optimal learning rate of $\widehat{y}^{\mathsf{gd}}(\eta)$ coincides with that of $\widehat{y}^{\mathsf{vgd}}(\eta)$, which is given by (F.28). Furthermore, the asymptotically optimal learning rate and the limiting $\ell_2$ risk are given by

$$
\begin{aligned}
\eta^* &= \lim_{\substack{d/L \to \xi \\ L \to \infty}} \frac{L}{d(1+\sigma^2) + L + 1} = \frac{1}{\xi \cdot (1+\sigma^2) + 1}, \\
\lim_{\substack{d/L \to \xi \\ L \to \infty}} \mathcal{E}(\widehat{y}_q^{\mathsf{gd}}(\eta^*)) &= \sigma^2 + \frac{\xi \cdot (1+\sigma^2)}{\xi \cdot (1+\sigma^2) + 1}.
\end{aligned}
\tag{F.33}
$$

Here we take the limit with $L \to \infty$ and $d/L \to \xi$ in (F.28) and (F.29) respectively.

**Step 3: Risk of Bayes Optimal Predictor.** As a comparison, we also consider the Bayes risk under this setup. Note that under the Gaussian prior over $\beta$, it is a classical result that the Bayesian optimal estimator of $y_q$ given $\{(x_\ell, y_\ell)\}_{\ell \in [L]} \cup x_q$ is given by ridge regression, which takes the form of $\widehat{y}_q^{\mathsf{ridge}} = \widehat{\beta}^{\mathsf{ridge}} x_q$. Here, we let $\widehat{\beta}^{\mathsf{ridge}, \top}$ denote the ridge estimator of $\beta$, and let $\mathsf{BayesRisk}_{\xi, \sigma^2}$ denote the Bayes risk of $\widehat{y}_q^{\mathsf{ridge}}$, where subscripts $\xi$ and $\sigma^2$ are used to indicate the dependence on $\xi$ in the limiting regime and the noise level $\sigma^2$. The ridge estimator and the Bayes risk are given by

$$\widehat{\beta}^{\mathsf{ridge}} = (X^\top X + \lambda I_d)^{-1} X^\top y, \qquad \mathsf{BayesRisk}_{\xi, \sigma^2} = \mathbb{E}\left[|\widehat{y}_q^{\mathsf{ridge}} - y_q|_2^2\right].$$

The connection between the ridge estimator and the Bayes optimal estimator is as follows. With the Gaussian prior $\mathcal{N}(0, \tau^2 \cdot I_d)$ for some parameter $\tau$, using the Gaussian likelihood of the linear model, we can calculate the posterior distribution of $\beta$. The Bayes optimal estimator of $\beta$ is given by the posterior mean, which takes the form of $\widehat{\beta}^{\mathsf{ridge}}$ with

$\lambda = \sigma^2/\tau^2$. Thus, in our case where $\beta \sim \mathcal{N}(0, I_d/d)$, the Bayes optimal estimator coincides with the ridge estimator with $\lambda = d\sigma^2$.

In the following, we derive the Bayes risk which largely follows the main proof in Dobriban & Wager (2018). We set $\lambda = d\sigma^2$ hereafter. Conditioning on $X$, the risk of $\widehat{y}_q^{\text{rigde}}$ is given by

$$\mathbb{E}[(\widehat{y}_q^{\text{rigde}} - y_q)^2 \mid X] = \mathbb{E}[(\widehat{\beta}^{\text{rigde},\top} x_q - \beta^\top x_q)^2 \mid X] + \sigma^2 = \mathbb{E}[\|\widehat{\beta}^{\text{rigde}} - \beta\|_2^2 \mid X] + \sigma^2,$$

where the expectation is with respect to the randomness of $x_q$, $\epsilon$, and $\beta$. By direct calculation, we simplify $\mathbb{E}[(\widehat{y}_q^{\text{rigde}} - y_q)^2 \mid X]$ as follows:

$$\begin{aligned}
\mathbb{E}[\|\widehat{\beta}^{\text{rigde}} - \beta\|_2^2 \mid X] &= \mathbb{E}[\|(X^\top X + d\sigma^2 \cdot I_d)^{-1} X^\top (X\beta + \epsilon) - \beta\|_2^2 \mid X] \\
&= \mathbb{E}\big[\big\|\big((X^\top X + d\sigma^2 \cdot I_d)^{-1} X^\top X - I_d\big)\beta\big\|_2^2 \mid X\big] \\
&\quad + \mathbb{E}[\|(X^\top X + d\sigma^2 \cdot I_d)^{-1} X^\top \epsilon\|_2^2 \mid X] \\
&= d\sigma^4 \cdot \text{tr}\big((X^\top X + d\sigma^2 \cdot I_d)^{-2}\big) \\
&\quad + \sigma^2 \cdot \text{tr}\big((X^\top X + d\sigma^2 \cdot I_d)^{-1} X^\top X (X^\top X + d\sigma^2 \cdot I_d)^{-1}\big) \\
&= \sigma^2 \cdot \text{tr}\big((X^\top X + d\sigma^2 \cdot I_d)^{-1}\big),
\end{aligned}$$

where the third equality follows from the fact that $(X^\top X + d\sigma^2 I_d)^{-1} X^\top X - I_d = d\sigma^2 \cdot (X^\top X + d\sigma^2 I_d)^{-1}$, and the last equality is obtained by plugging in $X^\top X = (X^\top X + d\sigma^2 \cdot I_d) - d\sigma^2 \cdot I_d$. By denoting $\xi_{d,L} = d/L$ and $\widehat{\Sigma} = X^\top X/L$, we can rewrite the equation above as

$$\sigma^{-2} \cdot \mathbb{E}[\|\widehat{\beta}^{\text{rigde}} - \beta\|_2^2 \mid X] = 1 + \frac{\xi_{d,L}}{d} \cdot \text{tr}\big((\widehat{\Sigma} + \sigma^2 \xi_{d,L} \cdot I_d)^{-1}\big). \tag{F.34}$$

The asymptotic behavior of the trace above can be tracked using random matrix theory (e.g., Ledoit & Péché, 2011; Wang et al., 2024). Let $m_F$ denote the Stieltjes transform of $F$ (Marchenko & Pastur, 1967), where $F$ is the limiting spectral distribution $\widehat{F}(t) = \frac{1}{d}\sum_{j=1}^d \mathbb{1}\{\lambda_j(\widehat{\Sigma}) \le t\}$. Then,

$$\frac{\xi_{d,L}}{d} \cdot \text{tr}\big((\widehat{\Sigma} + \xi_{d,L} \cdot I_d)^{-1}\big) \xrightarrow{\text{a.s.}} \xi \cdot m_F(-\sigma^2 \xi) \text{ with } \xi_{d,L} \to \xi. \tag{F.35}$$

Under the isotropic case where $x_\ell \overset{\text{i.i.d.}}{\sim} \mathcal{N}(0, I_d)$, $m_F$ has an explicit expression

$$m_F(t) = (1 - \xi - t - \sqrt{(1 - \xi - t)^2 - 4\xi t})/2\xi t$$

for all $t \in \mathbb{C}\backslash\mathbb{R}^+$. Based on (F.34) and (F.35), we have

$$\text{BayesRisk}_{\xi,\sigma^2} \xrightarrow{\text{a.s.}} \frac{1}{2}\Big(\sigma^2 + 1 - \xi^{-1} + \sqrt{4\sigma^2 + (\sigma^2 + \xi^{-1} - 1)^2}\Big). \tag{F.36}$$

Combining (F.33) and (F.36), the excess risk of $\widehat{y}_q^{\text{gd}}(\eta^*)$ compared to the Bayes risk satisfies that

$$\begin{aligned}
&\mathcal{E}(\widehat{y}_q^{\text{gd}}(\eta^*)) - \text{BayesRisk}_{\xi,\sigma^2} \\
&= \sigma^2 + 1 - \frac{1}{\xi(1 + \sigma^2) + 1} - \frac{1}{2}\Big(\sigma^2 + 1 - \xi^{-1}\Big) - \frac{1}{2}\sqrt{4\sigma^2 + (\sigma^2 + \xi^{-1} - 1)^2} \\
&= \frac{1}{2}\big(\sigma^2 + 1 + \xi^{-1} - \sqrt{4\sigma^2 + (\sigma^2 + \xi^{-1} - 1)^2}\big) - (\xi(1 + \sigma^2) + 1)^{-1} \\
&= 2\xi^{-1} \cdot \big(\sigma^2 + \xi^{-1} + 1 + \sqrt{4\sigma^2 + (\sigma^2 + \xi^{-1} - 1)^2}\big)^{-1} - (\xi(1 + \sigma^2) + 1)^{-1} \\
&\le \xi^{-1} \cdot \{(1 + \xi\sigma^2) \cdot (1 + \sigma^2 + \xi^{-1})\}^{-1}
\end{aligned}$$

where the inequality results from $\sqrt{4\sigma^2 + (\sigma^2 + \xi^{-1} - 1)^2} \ge \sigma^2 + \xi^{-1} - 1$ since we assume $\sigma^2 + \xi^{-1} \ge 1$. Furthermore, since $\text{BayesRisk}_{\xi,\sigma^2} \ge \sigma^2$ and we assume $\sigma^2 > 0$, then it holds that

$$\frac{\mathcal{E}(\widehat{y}_q^{\text{gd}}(\eta^*))}{\text{BayesRisk}_{\xi,\sigma^2}} \le 1 + 2\sigma^{-2} \cdot \{(1 + \xi\sigma^2) \cdot (1 + \sigma^2 + \xi^{-1})\}^{-1},$$

which gives that $\widehat{y}_q^{\text{gd}}(\eta^*)$ is near optimal when $\xi$ or $\sigma^2$ is large and then we complete the proof. $\qquad\square$

# G. Proof of Technical Lemmas

## G.1. Proof of Lemma E.2

*Proof of Lemma E.2.* We prove this lemma by applying the Stein's Lemma, which states that $\mathbb{E}[x \cdot g(x)] = \mathbb{E}[\nabla g(x)]$ holds for $x \sim \mathcal{N}(0, I_d)$ and any differentiable $g$.

STEP 1: DECOMPOSITION USING STEIN'S LEMMA.

To apply Stein's Lemma, we rewrite the $\mathbb{E}[\widetilde{p}^\top X X^\top p]$ into the summation of a sequence of functions of Gaussian random variables:

$$\mathbb{E}[\widetilde{p}^\top X X^\top p] = \mathbb{E}\big[\langle X^\top p, X^\top \widetilde{p}\rangle\big] = \mathbb{E}\bigg[\bigg\langle \sum_{\ell=1}^{L} p_\ell \cdot x_\ell, \sum_{\ell=1}^{L} \widetilde{p}_\ell \cdot x_\ell \bigg\rangle\bigg]$$

$$= \sum_{\ell=1}^{L} \mathbb{E}[\|x_\ell\|_2^2 \cdot p_\ell \widetilde{p}_\ell] + \sum_{1 \le \ell \ne k \le L} \mathbb{E}[\langle x_\ell, x_k\rangle \cdot p_\ell \widetilde{p}_k]. \tag{G.1}$$

Here we let $p_\ell$ and $\widetilde{p}_\ell$ denote the $\ell$-th entry of softmax vectors $p$ and $\widetilde{p}$, respectively. In the sequel, for any $j \in [d]$ and any $\ell \in [L]$, we let $x_{\ell,j}$ denote the $j$-th entry of $x_\ell$. For notational simplicity, we let $\partial_{\ell,j}$ denote as the partial derivative of function with respect to $x_{\ell,j}$. By applying the Stein's Lemma twice, we can show that

$$\mathbb{E}[\|x_\ell\|_2^2 \cdot p_\ell \widetilde{p}_\ell] = \sum_{j=1}^{d} \mathbb{E}[p_\ell \widetilde{p}_\ell] + \sum_{j=1}^{d} \mathbb{E}[x_{\ell,j} \cdot p_\ell \cdot \partial_{\ell,j} \widetilde{p}_\ell] + \sum_{j=1}^{d} \mathbb{E}[x_{\ell,j} \cdot \widetilde{p}_\ell \cdot \partial_{\ell,j} p_\ell]$$

$$= d \cdot \mathbb{E}[p_\ell \widetilde{p}_\ell] + 2\sum_{j=1}^{d} \mathbb{E}[\partial_{\ell,j} p_\ell \cdot \partial_{\ell,j} \widetilde{p}_\ell] + \sum_{j=1}^{d} \mathbb{E}[p_\ell \cdot \partial_{\ell,j}^2 \widetilde{p}_\ell] + \sum_{j=1}^{d} \mathbb{E}[\widetilde{p}_\ell \cdot \partial_{\ell,j}^2 p_\ell]. \tag{G.2}$$

Here the first equality is obtained by applying Stein's lemma with $g(x_{\ell,j}) = x_{\ell,j} \cdot p_\ell \cdot \widetilde{p}_\ell$. Similarly, for any $k \ne \ell$, by applying Stein's Lemma twice, we have

$$\mathbb{E}[\langle x_\ell, x_k\rangle \cdot p_\ell \widetilde{p}_k] = \sum_{j=1}^{d} \mathbb{E}[x_{k,j} \cdot p_\ell \cdot \partial_{\ell,j} \widetilde{p}_k] + \sum_{j=1}^{d} \mathbb{E}[x_{k,j} \cdot \widetilde{p}_k \cdot \partial_{\ell,j} p_\ell]$$

$$= \sum_{j=1}^{d} \mathbb{E}[p_\ell \cdot \partial_{k,j} \partial_{\ell,j} \widetilde{p}_k] + \sum_{j=1}^{d} \mathbb{E}[\widetilde{p}_k \cdot \partial_{k,j} \partial_{\ell,j} p_\ell]$$

$$+ \sum_{j=1}^{d} \mathbb{E}[\partial_{k,j} p_\ell \cdot \partial_{\ell,j} \widetilde{p}_k] + \sum_{j=1}^{d} \mathbb{E}[\partial_{k,j} \widetilde{p}_k \cdot \partial_{\ell,j} p_\ell]. \tag{G.3}$$

STEP 2: DERIVATIVES CALCULATIONS AND SIMPLIFICATIONS.

By (G.2) and (G.3), we write (G.1) as a sum of expectations involving $p$, $\widetilde{p}$, and first- and second-order their derivatives. By direct calculations, the first-order derivatives of $p$ are given by

$$\partial_{\ell,j} p_\ell = \omega \cdot v_j \cdot (p_\ell - p_\ell^2), \qquad \partial_{k,j} p_\ell = -\omega \cdot v_j \cdot p_\ell p_k. \tag{G.4}$$

The second-order derivatives of $p$ are given by

$$\partial_{\ell,j}^2 p_\ell = \omega \cdot v_j \cdot (1 - 2p_\ell) \cdot \partial_{\ell,j} p_\ell = \omega^2 \cdot v_j^2 \cdot (1 - 2p_\ell) \cdot (p_\ell - p_\ell^2), \tag{G.5}$$

$$\partial_{k,j} \partial_{\ell,j} p_\ell = \partial_{\ell,j} \partial_{k,j} p_\ell = \omega \cdot v_j \cdot (1 - 2p_\ell) \cdot \partial_{k,j} p_\ell = -\omega^2 \cdot v_j^2 \cdot (1 - 2p_\ell) \cdot p_\ell p_k. \tag{G.6}$$

We can similarly derive the first- and second-order derivatives of $\widetilde{p}$.

In the following, we conclude the proof of this lemma by substituting the calculations above back into (G.2) and (G.3), and simplifing the expression. First, note that, if we view $\mathbb{E}[\widetilde{p}^\top X X^\top p]$ as a bivariate function of $(\omega, \widetilde{\omega})$, it is a quadratic

function. The constant term is given by

$$d \cdot \sum_{\ell=1}^{L} \mathbb{E}[p_\ell \widetilde{p}_\ell] = d \cdot \mathbb{E}[p^\top \widetilde{p}]. \tag{G.7}$$

Thus, we can write $\mathbb{E}[\widetilde{p}^\top X X^\top p]$ as

$$\mathbb{E}[\widetilde{p}^\top X X^\top p] = A_0 \cdot \omega^2 + A_1 \cdot \omega \cdot \widetilde{\omega} + A_2 \cdot \widetilde{\omega}^2 + d \cdot \mathbb{E}[p^\top \widetilde{p}] \tag{G.8}$$

for some constants $A_0, A_1, A_2$ to be determined. By examining the derivatives of $p$ and $\widetilde{p}$, we notice that these coefficients are given respectively by

$$A_0 = \sum_{\ell=1}^{L} \sum_{j=1}^{d} \mathbb{E}[\widetilde{p}_\ell \cdot \partial_{\ell,j}^2 p_\ell] + \sum_{1 \le \ell \ne k \le L} \sum_{j=1}^{d} \mathbb{E}[\widetilde{p}_\ell \cdot \partial_{k,j} \partial_{\ell,j} p_k], \tag{G.9}$$

$$A_1 = 2 \sum_{\ell=1}^{L} \sum_{j=1}^{d} \mathbb{E}[\partial_{\ell,j} p_\ell \cdot \partial_{\ell,j} \widetilde{p}_\ell] + \sum_{1 \le \ell \ne k \le L} \sum_{j=1}^{d} \{\mathbb{E}[\partial_{k,j} p_\ell \cdot \partial_{\ell,j} \widetilde{p}_k] + \mathbb{E}[\partial_{k,j} \widetilde{p}_k \cdot \partial_{\ell,j} p_\ell]\}, \tag{G.10}$$

$$A_2 = \sum_{\ell=1}^{L} \sum_{j=1}^{d} \mathbb{E}[p_\ell \cdot \partial_{\ell,j}^2 \widetilde{p}_\ell] + \sum_{1 \le \ell \ne k \le L} \sum_{j=1}^{d} \mathbb{E}[p_\ell \cdot \partial_{k,j} \partial_{\ell,j} \widetilde{p}_k]. \tag{G.11}$$

For $A_2$, combining (G.5), (G.6), and (G.11), we have

$$A_2 = \widetilde{\omega}^2 \cdot \|v\|_2^2 \cdot \left( \mathbb{E}[p^\top \widetilde{p}] - 2 \cdot \mathbb{E}[p^\top \widetilde{p}^{\odot 2}] \right)$$
$$- \widetilde{\omega}^2 \cdot \|v\|_2^2 \cdot \left\{ \sum_{\ell=1}^{L} \mathbb{E}[p_\ell \widetilde{p}_\ell \cdot \widetilde{p}_\ell] + \sum_{1 \le \ell \ne k \le L} \mathbb{E}[p_\ell \widetilde{p}_\ell \cdot \widetilde{p}_k] \right\}$$
$$+ 2\widetilde{\omega}^2 \cdot \|v\|_2^2 \cdot \left\{ \sum_{\ell=1}^{L} \mathbb{E}[p_\ell \widetilde{p}_\ell \cdot \widetilde{p}_\ell^2] + \sum_{1 \le \ell \ne k \le L} \mathbb{E}[p_\ell \widetilde{p}_\ell \cdot \widetilde{p}_k^2] \right\}.$$

By direct calculation, we have

$$\sum_{\ell=1}^{L} \mathbb{E}[p_\ell \widetilde{p}_\ell \cdot \widetilde{p}_\ell] + \sum_{1 \le \ell \ne k \le L} \mathbb{E}[p_\ell \widetilde{p}_\ell \cdot \widetilde{p}_k] = \mathbb{E}\left[ \left( \sum_{\ell=1}^{L} p_\ell \widetilde{p}_\ell \right) \cdot \left( \sum_{k=1}^{L} \widetilde{p}_k \right) \right] = \mathbb{E}[p^\top \widetilde{p}],$$

$$\sum_{\ell=1}^{L} \mathbb{E}[p_\ell \widetilde{p}_\ell \cdot \widetilde{p}_\ell^2] + \sum_{1 \le \ell \ne k \le L} \mathbb{E}[p_\ell \widetilde{p}_\ell \cdot \widetilde{p}_k^2] = \mathbb{E}\left[ \left( \sum_{\ell=1}^{L} p_\ell \widetilde{p}_\ell \right) \cdot \left( \sum_{k=1}^{L} \widetilde{p}_k^2 \right) \right] = \mathbb{E}[p^\top \widetilde{p} \cdot \|\widetilde{p}\|_2^2],$$

where in the first equality, we use the fact that $\widetilde{p}$ is a prabability distribution. Combining the above three equalities, we can simplify $A_2$ as

$$A_2 = \widetilde{\omega}^2 \cdot \|v\|_2^2 \cdot \left( \mathbb{E}[p^\top \widetilde{p}] - 2 \cdot \mathbb{E}[p^\top \widetilde{p}^{\odot 2}] \right)$$
$$- \widetilde{\omega}^2 \cdot \|v\|_2^2 \cdot \mathbb{E}[p^\top \widetilde{p} \cdot \widetilde{p}^\top \mathbf{1}_L] + 2\omega^2 \cdot \|v\|_2^2 \cdot \mathbb{E}[p^\top \widetilde{p} \cdot \|\widetilde{p}\|_2^2]$$
$$= 2\widetilde{\omega}^2 \cdot \|v\|_2^2 \cdot \mathbb{E}[-p^\top \widetilde{p}^{\odot 2} + p^\top \widetilde{p} \cdot \|\widetilde{p}\|_2^2]. \tag{G.12}$$

Following the same argument, we can show that $A_0$ defined in (G.9) can be simplified as

$$A_0 = 2\omega^2 \cdot \|v\|_2^2 \cdot \mathbb{E}[-\widetilde{p}^\top p^{\odot 2} + p^\top \widetilde{p} \cdot \|p\|_2^2]. \tag{G.13}$$

Then, it remains to consider $A_1$. Note that by the first equality in (G.4), we have

$$\sum_{\ell=1}^{L} \sum_{j=1}^{d} \mathbb{E}[\partial_{\ell,j} p_\ell \cdot \partial_{\ell,j} \widetilde{p}_\ell] = \omega \widetilde{\omega} \cdot \sum_{\ell=1}^{L} \sum_{j=1}^{d} v_j^2 \cdot \mathbb{E}[p_\ell \cdot \widetilde{p}_\ell - p_\ell^2 \cdot \widetilde{p}_\ell - p_\ell \cdot \widetilde{p}_\ell^2 + p_\ell^2 \cdot \widetilde{p}_\ell^2]$$

$$= \omega\widetilde{\omega} \cdot \|v\|_2^2 \cdot \mathbb{E}[p^\top \widetilde{p} - p^\top \widetilde{p}^{\odot 2} - \widetilde{p}^\top p^{\odot 2}] + \omega\widetilde{\omega} \cdot \|v\|_2^2 \cdot \sum_{\ell=1}^{L} \mathbb{E}\left[p_\ell^2 \widetilde{p}_\ell^2\right].$$

By the second equality in (G.4), for any $k \neq \ell$, we have

$$\mathbb{E}[\partial_{k,j}\widetilde{p}_k \cdot \partial_{\ell,j}p_\ell] = \omega\widetilde{\omega} \cdot v_j^2 \cdot \mathbb{E}[p_\ell p_k \cdot \widetilde{p}_\ell \widetilde{p}_k]$$

$$\mathbb{E}[\partial_{k,j}\widetilde{p}_k \cdot \partial_{\ell,j}p_\ell] = \omega\widetilde{\omega} \cdot v_j^2 \cdot \mathbb{E}[p_\ell \cdot \widetilde{p}_\ell - p_\ell^2 \cdot \widetilde{p}_\ell - p_\ell \cdot \widetilde{p}_\ell^2 + p_\ell^2 \cdot \widetilde{p}_\ell^2].$$

Thus, combining (G.10) with the above three equalities, we can simplify the first part of $A_1$ as

$$\sum_{\ell=1}^{L}\sum_{j=1}^{d}\mathbb{E}[\partial_{\ell,j}p_\ell \cdot \partial_{\ell,j}\widetilde{p}_\ell] + \sum_{1\leq\ell\neq k\leq L}\sum_{j=1}^{d}\mathbb{E}[\partial_{k,j}p_\ell \cdot \partial_{\ell,j}\widetilde{p}_k]$$

$$= \omega\widetilde{\omega} \cdot \|v\|_2^2 \cdot \mathbb{E}[p^\top \widetilde{p} - p^\top \widetilde{p}^{\odot 2} - \widetilde{p}^\top p^{\odot 2}]$$

$$+ \omega\widetilde{\omega} \cdot \|v\|_2^2 \cdot \left\{\sum_{\ell=1}^{L}\mathbb{E}\left[(p_\ell\widetilde{p}_\ell)^2\right] + \sum_{1\leq\ell\neq k\leq L}\mathbb{E}[p_\ell\widetilde{p}_\ell \cdot p_k\widetilde{p}_k]\right\}$$

$$= \omega\widetilde{\omega} \cdot \|v\|_2^2 \cdot \mathbb{E}[p^\top \widetilde{p} - p^\top \widetilde{p}^{\odot 2} - \widetilde{p}^\top p^{\odot 2} + (p^\top \widetilde{p})^2]. \tag{G.14}$$

Similarly, the second part of $A_1$ can be written as

$$\sum_{\ell=1}^{L}\sum_{j=1}^{d}\mathbb{E}[\partial_{\ell,j}p_\ell \cdot \partial_{\ell,j}\widetilde{p}_\ell] + \sum_{1\leq\ell\neq k\leq L}\sum_{j=1}^{d}\mathbb{E}[\partial_{k,j}\widetilde{p}_k \cdot \partial_{\ell,j}p_\ell]$$

$$= \omega\widetilde{\omega} \cdot \|v\|_2^2 \cdot \sum_{\ell=1}^{L}\mathbb{E}[p_\ell \cdot \widetilde{p}_\ell - p_\ell^2 \cdot \widetilde{p}_\ell - p_\ell \cdot \widetilde{p}_\ell^2 + p_\ell^2 \cdot \widetilde{p}_\ell^2]$$

$$+ \omega\widetilde{\omega} \cdot \|v\|_2^2 \cdot \sum_{1\leq\ell\neq k\leq L}\mathbb{E}[p_\ell \cdot \widetilde{p}_k - p_\ell^2 \cdot \widetilde{p}_k - p_\ell \cdot \widetilde{p}_k^2 + p_\ell^2 \cdot \widetilde{p}_k^2]$$

$$= \omega\widetilde{\omega} \cdot \|v\|_2^2 \cdot \mathbb{E}[p^\top \mathbf{1}_L \cdot \widetilde{p}^\top \mathbf{1}_L - \|p\|_2^2 \cdot \widetilde{p}^\top \mathbf{1}_L - \|\widetilde{p}\|_2^2 \cdot p^\top \mathbf{1}_L + \|p\|_2^2 \cdot \|\widetilde{p}\|_2^2]$$

$$= \omega\widetilde{\omega} \cdot \|v\|_2^2 \cdot \mathbb{E}[(1 - \|p\|_2^2) \cdot (1 - \|\widetilde{p}\|_2^2)]. \tag{G.15}$$

Thus, combining (G.14) and (G.15), we can write $A_1$ as

$$A_1 = \omega\widetilde{\omega} \cdot \|v\|_2^2 \cdot \mathbb{E}\left[p^\top \widetilde{p} - p^\top \widetilde{p}^{\odot 2} - \widetilde{p}^\top p^{\odot 2} + (p^\top \widetilde{p})^2 + (1 - \|p\|_2^2) \cdot (1 - \|\widetilde{p}\|_2^2)\right]. \tag{G.16}$$

Finally, by combining (G.7), (G.8), (G.12), (G.13), and (G.16), we complete the proof. $\square$

## G.2. Proof of Concentration Arguments

**Lemma G.1** ($\chi^2$-concentration)**.** *Consider a random vector $x = (x_1, \ldots, x_d) \in \mathbb{R}^d$ with each entry i.i.d. sampled from $\mathcal{N}(0,1)$. For any positive $t > 0$, the following inequality holds:*

$$\mathbb{P}(\|x\|_2^2 \geq d + 2\sqrt{dt} + 2t) \leq \exp(-t), \quad \mathbb{P}(\|x\|_2^2 \leq d - 2\sqrt{dt}) \leq \exp(-t).$$

*Also, it holds that $\mathbb{P}(\|x\|_2^2 \geq d \cdot (1+t)^2) \leq \exp(-dt^2/2)$.*

*Proof of Lemma G.1.* See Lemma 1 in Laurent & Massart (2000) and Lemma A.1 in Bai et al. (2024) for detailed proofs. $\square$

**Lemma G.2** (Binomial concentration)**.** *Consider Binomial random variable $X \sim \mathrm{Binom}(n,p)$ with $n \in \mathbb{N}$ and $p \in (0,1)$, then for any $t \in (0,1)$, it holds that*

$$\mathbb{P}(X \geq (1+t) \cdot np) \leq \exp(-t^2 np/3), \quad \mathbb{P}(X \geq (1-t) \cdot np) \leq \exp(-t^2 np/2).$$

*Proof of Lemma G.2.* The result follows a direct implementation of Chernoff's bound. $\square$

**Lemma G.3.** *Consider* $x_\ell \overset{i.i.d.}{\sim} \mathcal{N}(0, I_d)$, $\beta \overset{i.i.d.}{\sim} \mathcal{N}(0, I_d/d)$ *and* $y_\ell = \beta^\top x_\ell + \epsilon_\ell$, *where* $\epsilon_\ell \overset{i.i.d.}{\sim} \mathcal{N}(0, \sigma^2)$ *with* $L \in \mathbb{N}$. *For any* $\delta \in (0,1)$, *with probability at least* $1 - \delta$, *the following events hold simultaneously:*

*(i).* $\max_{\ell \in [L]} |x_\ell^\top x_q| \le \sqrt{12d} \cdot \log(8L/\delta)$;

*(ii).* $\frac{1}{L} \sum_{\ell=1}^{L} y_\ell^2 \le 1 + \sigma^2 + \mathrm{poly}(d^{-1/2}, L^{-1/2}, \log(1/\delta))$;

*(iii).* $\left| \frac{1}{L} \sum_{\ell=1}^{L} \mathbb{1}(x_\ell^\top x_q > 0) - \frac{1}{2} \right| \le \min\{\sqrt{\log(8/\delta)/L}, 1/2\}$.

*Proof of Lemma G.3.* We first consider the following good events hold:

$$\mathcal{E}_1 = \left\{ \|x_q\|_2 \le \sqrt{6d\log(4/\delta)}, \quad \|\beta\|_2 \le 1 + \sqrt{2d^{-1} \cdot \log(4/\delta)} \right\},$$

and we can upper bound $\mathbb{P}(\mathcal{E}_1) \le \delta/4$ using Lemma G.1. Note that for fixed $x_q$, we have $x_\ell^\top x_q \overset{i.i.d.}{\sim} \mathcal{N}(0, \|x_q\|_2^2)$ and hence by applying a Gaussian tail bound, we have

$$\mathbb{P}\left(\exists \ell \in [L] : |x_\ell^\top x_q| \ge t \mid x_q\right) \le \sum_{\ell=1}^{L} \mathbb{P}\left(|x_\ell^\top x_q| \ge t \mid x_q\right) \le 2L \cdot \exp\left(-t^2/2\|x_q\|_2^2\right).$$

Hence, $\max_{\ell \in [L]} |x_\ell^\top x_q| \le \sqrt{12d} \cdot \log(8L/\delta)$ with probability greater than $1 - \delta/4$ under good event $\mathcal{E}_1$. Similarly, we have $y_\ell = \beta^\top x_\ell + \epsilon \overset{i.i.d.}{\sim} \mathcal{N}(0, \|\beta\|_2^2 + \sigma^2)$. Based on Lemma G.1, it holds that

$$\mathbb{P}\left(\|y\|_2^2 > L \cdot (\|\beta\|_2^2 + \sigma^2) \cdot (1+t)^2 \mid \beta\right) \le \exp(-Lt^2/2),$$

Following this, under good event $\mathcal{E}_1$, with probability greater than $1 - \delta/4$, we have

$$\frac{1}{L} \sum_{\ell=1}^{L} y_\ell^2 \le \left(1 + \sigma^2 + 5d^{-1/2} \cdot \log(4/\delta)\right) \cdot \left(1 + 5L^{-1/2} \cdot \log(4/\delta)\right)$$

$$= 1 + \sigma^2 + \mathrm{poly}(d^{-1/2}, L^{-1/2}, \log(1/\delta)).$$

Furthermore, since $x_\ell^\top x_q \overset{i.i.d.}{\sim} \mathcal{N}(0, \|x_q\|_2^2)$ and $\mathbb{P}(x_\ell^\top x_q > 0 \mid x_q) = 1/2$ for all $x_q$, then using the binomial tail bound in Lemma G.2, we can show that

$$\mathbb{P}\left(\left| \frac{1}{L} \sum_{\ell=1}^{L} \mathbb{1}(x_\ell^\top x_q > 0) - \frac{1}{2} \right| < t/2 \mid x_q\right) \le 2\exp(-t^2 L/4),$$

and hence $\left| \frac{1}{L} \sum_{\ell=1}^{L} \mathbb{1}(x_\ell^\top x_q > 0) - \frac{1}{2} \right| \le \min\{L^{-\frac{1}{2}} \sqrt{\log(8/\delta)}, 1/2\}$ with probability at least $1 - 4\delta$. Combining all the arguments above, we can complete the proof. $\qquad\square$

### G.3. Proof of Anisotropic Pre-Conditioned GD Estimator

For simplicity, we consider a symmetric pre-conditioning matrix $\Gamma$ which proved to be the global optimal parametrization for pre-conditioned GD, which can be extended to the general case with more careful arguments (e.g., Ahn et al., 2023a).

**Lemma G.4.** *Suppose* $\beta \sim \mathcal{N}(0, I_d/d)$, $X = [x_1, \ldots, x_L]^\top \in \mathbb{R}^{L \times d}$ *with* $x_\ell \overset{i.i.d.}{\sim} \mathcal{N}(0, \Sigma)$ *and* $y_\ell = \beta^\top x_\ell + \epsilon_\ell$ *with* $\epsilon_\ell \overset{i.i.d.}{\sim} \mathcal{N}(0, \sigma^2)$. *For any* $\Gamma \in \mathbb{R}^{d \times d}$, *define* $\widehat{y}_q^{\mathsf{vgd}} := \widehat{y}_q^{\mathsf{vgd}}(\Gamma) = L^{-1} \cdot (X\beta + \epsilon)^\top X\Gamma^{-1} x_q$. *Consider the ICL prediction risk* $\mathbb{E}[(\widehat{y}_q^{\mathsf{vgd}} - \widehat{y}_q)^2]$ *as a function of* $\Gamma$. *Then, the minimizer of* $\mathbb{E}[(\widehat{y}_q^{\mathsf{vgd}} - \widehat{y}_q)^2]$ *is given by*

$$\Gamma^* = (L+1)/L \cdot \Sigma + (\mathrm{tr}(\Sigma) + d\sigma^2)/L \cdot I_d.$$

*Proof of Lemma G.4.* For any matrix $\Gamma \in \mathbb{R}^{d \times d}$, and the risk of corresponding estimator follows

$$\mathbb{E}[(\widehat{y}_q^{\mathsf{vgd}} - \widehat{y}_q)^2] = \mathbb{E}\left[\left(\frac{1}{L}(X\beta + \epsilon)^\top X\Gamma^{-1} x_q - \beta^\top x_q\right)^2\right] + \sigma^2$$

$$= \frac{1}{d} \cdot \underbrace{\mathbb{E}\left[\text{tr}\left((X^\top X \Gamma^{-1}/L - I_d)^\top (X^\top X \Gamma^{-1}/L - I_d)\Sigma\right)\right]}_{\textbf{(i)}}$$

$$+ \frac{\sigma^2}{L^2} \cdot \underbrace{\mathbb{E}[\text{tr}(\Gamma^{-1} X^\top X \Gamma^{-1}\Sigma)]}_{\textbf{(ii)}} + \sigma^2. \tag{G.17}$$

Following the decomposition above, by simple calculations, the second term follows

$$\textbf{(ii)} = \mathbb{E}[\text{tr}(X^\top X \Gamma^{-1}\Sigma\Gamma^{-1})] = L \cdot \text{tr}\left((\Sigma\Gamma^{-1})^2\right). \tag{G.18}$$

Furthermore, note that the first term takes the form

$$\textbf{(i)} = \text{tr}(\Sigma) - \frac{2}{L} \cdot \mathbb{E}[\text{tr}(X^\top X \Gamma^{-1}\Sigma)] + \frac{1}{L^2} \cdot \mathbb{E}[\text{tr}(\Gamma^{-1}(X^\top X)^2 \Gamma^{-1}\Sigma)]$$

$$= \text{tr}(\Sigma) - 2 \cdot \text{tr}(\Sigma\Gamma^{-1}\Sigma) + \frac{1}{L^2} \cdot \mathbb{E}[\text{tr}((X^\top X)^2 \Gamma^{-1}\Sigma\Gamma^{-1})], \tag{G.19}$$

where the second equation follows $X^\top X = \sum_{\ell=1}^{L} x_\ell x_\ell^\top$ and we can show that

$$\mathbb{E}[(X^\top X)^2] = \sum_{\ell=1}^{L} \mathbb{E}[(x_\ell x_\ell^\top)^2] + \sum_{1 \le \ell \ne j \le L} \mathbb{E}[x_\ell x_\ell^\top \cdot x_j x_j^\top] = L \cdot \mathbb{E}[(x_\ell x_\ell^\top)^2] + L(L-1) \cdot \Sigma^2.$$

Specifically, for all $(i,j) \in [d] \times [d]$, the $(i,j)$-th entry of the first matrix satisfies that

$$\mathbb{E}[(x_\ell x_\ell^\top)^2_{i,j}] = \sum_{k=1}^{d} \mathbb{E}[x_{\ell,i} x_{\ell,j}] \cdot \mathbb{E}[x_{\ell,k}^2] + 2 \cdot \sum_{k=1}^{d} \mathbb{E}[x_{\ell,i} x_{\ell,k}] \cdot \mathbb{E}[x_{\ell,j} x_{\ell,k}] = \Sigma_{ij} \cdot \text{tr}(\Sigma) + \Sigma_{:,i}^\top \Sigma_{:,j},$$

where the first inequality uses the Isserlis' theorem and thus we have $\mathbb{E}[(x_\ell x_\ell^\top)^2] = \text{tr}(\Sigma)\Sigma + 2\Sigma^2$. For notational simplicity, denote $\widetilde{\Gamma} = \Sigma^{-1}\Gamma$. Combining (G.17), (G.18) and (G.19), it holds that

$$\mathbb{E}[(\widehat{y}_q^{\text{vgd}} - \widehat{y}_q)^2] = \sigma^2 + \frac{\text{tr}(\Sigma)}{d} - \frac{2}{d} \cdot \text{tr}(\Sigma\widetilde{\Gamma}^{-1}) + \frac{\text{tr}(\Sigma) + d\sigma^2}{dL} \cdot \text{tr}\left(\widetilde{\Gamma}^{-2}\right) + \frac{L+1}{dL} \cdot \text{tr}\left(\Sigma\widetilde{\Gamma}^{-2}\right)$$

$$= \sigma^2 + \frac{\text{tr}(\Sigma)}{d} - \frac{2}{d} \cdot \text{tr}\left(\Sigma\widetilde{\Gamma}^{-1}\right) + \frac{1}{dL} \cdot \text{tr}\left(\left\{(\text{tr}(\Sigma) + d\sigma^2) \cdot I_d + (L+1) \cdot \Sigma\right\}\widetilde{\Gamma}^{-2}\right).$$

Since $(\text{tr}(\Sigma) + d\sigma^2) \cdot I_d + (L+1) \cdot \Sigma$ is symmetric and invertible, the minimizer is given by

$$\widetilde{\Gamma}^{*,-1} = L \cdot \left\{(\text{tr}(\Sigma) + d\sigma^2) \cdot I_d + (L+1) \cdot \Sigma\right\}^{-1} \Sigma, \quad \text{s.t.} \quad \Gamma^* = \left(1 + \frac{1}{L}\right)\Sigma + \frac{\text{tr}(\Sigma) + d\sigma^2}{L} \cdot I_d,$$

due to the quadratic form and then we complete the proof. $\qquad\square$

