# OpenReview forum: "In-Context Linear Regression Demystified: Training Dynamics and Mechanistic Interpretability of Multi-Head Softmax Attention"
_ICML.cc/2025/Conference — ICML 2025 poster_

### Official Review · Reviewer_bTEk · 2025-03-13

**Overall Recommendation:** 4

**Summary:**

The authors find that a multi-head softmax attention effectively becomes a two-head softmax attention network that approximates linear attention better than a single-head softmax one. The advantage of softmax based attention over linear attention is that one does not have to train separately for different context lengths. The experiments bears out these observations. There is also some analysis to training dynamics.

## Update after rebuttal

I maintain my score.

**Claims And Evidence:**

The claims are justified.

**Essential References Not Discussed:**

I did not notice obvious omissions.

**Experimental Designs Or Analyses:**

Figure 7 provides empirical evidence of the arguments being right.

**Methods And Evaluation Criteria:**

It is mostly a theoretical paper, needing simple experimental evidence, which was provided.

**Other Comments Or Suggestions:**

This seems to be well-written paper.

**Other Strengths And Weaknesses:**

I think the analysis is useful for many but the key results were relatively obvious to me.

**Questions For Authors:**

I do not have any.

**Relation To Broader Scientific Literature:**

In recent past, there has been many papers analyzing in-context learning in toy problems. Many of these involve single-head attention. There are not many papers among them tackling multi-head attention. Although some of the arguments are informal, there is value in this work.

**Theoretical Claims:**

The theoretical claims involve informal but reasonable Taylor expansion based arguments.

---

> ### Author Rebuttal · Authors · 2025-03-31
>
> We sincerely thank the reviewer for their positive feedback and appreciation of our work. Here's our clarifications on theoretical analysis.
>
> **On Informal Theoretical Claims.** We would like to clarify that most of the theoretical arguments presented in this paper are rigorous, with the exception of the analysis for Stage I in Section 4.2, i.e., theoretical derivation of feature emergence via gradient optimization. While both Stage I and Stage II admit a Taylor expansion-based argument, the analyses are derived from different perspectives. We will provide further clarification as follows.
>
> - (`Analysis for Stage I`) In Stage I, we perform a gradient-based analysis to identify the driving components in the gradient that lead to the emergence of patterns. We acknowledge that in this stage, we employ an informal argument by focusing on the first-order approximation of $\exp(\cdot)$, leveraging the small-scale initialization that renders higher-order terms negligible. To support this argument, we compare the informal theoretical conjectures with experimental results, which show perfect alignment.
> - (`Analysis for Stage II`) In contrast, during Stage II, we analyze the loss landscape to understand the convergence manifold. Assuming that the patterns identified in the first stage have already emerged, we examine the higher-order terms in the Taylor expansion of the loss function to demonstrate that homogeneous KQ scaling is a necessary condition for loss minimization. We emphasize that the expansion-based argument in this stage is rigorous for the following reasons: (a) the parameter scale remains small during the final stage, allowing the series summation to be analytically tractable, and (b) we derive the optimality condition for each term of the form $\langle\mu_t,\omega_t^{\odot k}\rangle$ for $k\in\mathbb{Z}$, without truncation, ensuring that the result is both rigorous and concise.

---

> > ### Comment · Reviewer_bTEk · 2025-04-04
> >
> > I thank the authors for their response. I might agree with some other reviewers that the work is somewhat limited. Having worked on this area, upon reading the paper, my first sets of feelings were:
> > 1) Now that I have seen it, the results are obvious.
> > 2) Why did I not think of this?
> > I think the authors deserve credit for invoking such feelings. Hence, I would like to maintain my score. Best of luck!

---

> > > ### Author Response · Authors · 2025-04-08
> > >
> > > Thanks for your kind appreciation of our work!

---

### Official Review · Reviewer_Y3fQ · 2025-03-14

**Overall Recommendation:** 3

**Summary:**

The paper provides an comprehensive understanding of multi-head softmax attention in conducting ICL for linear regression tasks. Through empirical investigation, it observes the specific patterns for the optimal parameters within in the multi-head transformer structures and the superiority of multi-head over single-head. Based on the observations of the specific patterns, it reparameterizes the attention models to conduct theoretical understanding of the training dynamics and expressive power of multi-head softmax attention. It shows that multi-head softmax attention emulates preconditioned gradient descent and achieves near-Bayesian optimality in in-context learning. Additionally, the paper extends its findings to non-isotropic covariates and multi-task regression and also provide comparison with linear attention.

**Claims And Evidence:**

In general, the claims made in the submissions are clear and well-written. The paper provides empirical evidence for the observed specific patterns for the optimal configuration of multi-head softmax attention and the superiority of multi-head attention. For theoretical results, it also provides some empirical validations and proofs of the statements.

**Essential References Not Discussed:**

This paper focuses on investigating the training dynamics and expressive power of multi-head softmax attention. [2] investigating similar topics but the paper did not cite and compare with [2]. In addition, given that [1] also investigates the training dynamic and optimality, the paper did not provide a very clear explanation about the difference between this work and [1]. [3] also have a close relation to the content of this paper. These works are highly related and important, and it is necessary to discuss them for a careful literature review.



[1] Chen, S., Sheen, H., Wang, T., & Yang, Z.. Training dynamics of multi-head softmax attention for in-context learning: Emergence, convergence, and optimality. COLT 2024.

[2] Cui, Y., Ren, J., He, P., Tang, J., & Xing, Y. Superiority of multi-head attention in in-context linear regression. 2024.

[3] Li, H., Wang, M., Lu, S., Cui, X., & Chen, P. Y. How do nonlinear transformers learn and generalize in in-context learning? ICML 2024

**Experimental Designs Or Analyses:**

I have checked the details of the experiments setting, results and analysis in Section 3 (and some further explanations and discussion in Appendix). In general, the empirical settings are reasonable and align well with the design of theoretical settings, and I did not observe particular issues.

**Methods And Evaluation Criteria:**

The paper did not propose methods / evaluation criteria.

**Other Comments Or Suggestions:**

1. It is suggested that the authors can provide some practical implications of the findings in this paper for in-context learning on real-world text data. Understanding this is important, as the theoretical analyses focus on simplified settings, and their relevance to real NLP tasks remains uncertain.

2. The paper considers that for multi-head attentions, the outputs of all heads are summed. Would incorporating a learned weighted sum for each head further improve the prediction? Intuitively, it can provide more adaptability by allowing the model to assign different importance to different heads. Additionally, the paper states that increasing the number of heads beyond H = 2 provides no additional benefit. If a weighted sum is used, could further increasing the number of heads lead to performance improvements?

3. The training dynamics analysis is based on a reparameterized model. If I understand correctly, it assumes that throughout the entire training process, the QK matrix is constrained to always have a diagonal structure, with only its diagonal values being trained. Although empirical results suggest that the optimal configuration after training is indeed diagonal, this simplification deviates from the actual training process, potentially oversimplifying the learning dynamics and failing to capture the complex interactions that may emerge in real transformer training.

**Other Strengths And Weaknesses:**

Strength:
1. In general, the paper is well-written and easy to follow.

2. The analysis is comprehensive, as it involves both empirical and theoretical results of multi-head softmax attention, and cover different perspectives, including optimal model configuration, training dynamics, expressive power, comparison between multi-head / single-head attentions, comparison with linear attention, and extensions to non-isotropic covariates and multi-task regression.

Weakness:
1. As indicates in the previous review section, the novelty of the paper is limited, as the main contribution has been covered by previous papers [1][2].

2. The paper observers the optimal configuration of softmax multi-head attentions by only empirical observations. In contrast, [2] provides theory showing the diagonal property of the optimal QK weight and other configurations. Additionally, compared to [1] and [2], the theoretical analysis in this paper reparameterizes the attention parameters into just two vectors, significantly simplifying the complexity of the theoretical derivation.

[1]  Chen, S., Sheen, H., Wang, T., & Yang, Z.. Training dynamics of multi-head softmax attention for in-context learning: Emergence, convergence, and optimality. COLT 2024.

[2] Cui, Y., Ren, J., He, P., Tang, J., & Xing, Y. Superiority of multi-head attention in in-context linear regression. 2024.

**Questions For Authors:**

N/A

**Relation To Broader Scientific Literature:**

The key contribution of the paper, claimed by the authors, includes identifying the optimal configuration of multi-head softmax attentions for in-context linear regression, analyzing the training dynamics and expressive power, and demonstrating its advantages over single-head models. The analysis builds on prior studies of ICL for linear regression using transformers.

**However, a major concern regarding this paper is its novelty, as many of its contributions appear to have been covered by previous works.** For instance, [1] has already analyzed the training dynamics of multi-head softmax attention in ICL, discussing optimality through upper/lower bounds on the ICL loss and providing results based on Bayesian risk analysis. Additionally, [2] has presented similar findings regarding the optimal configuration, such as the diagonal structure of the QK matrix. [2] has also compared multi-head and single-head attention and extended the discussion to external scenarios such as non-Isotropic covariates (correlated features).

[1] Chen, S., Sheen, H., Wang, T., & Yang, Z. Training dynamics of multi-head softmax attention for in-context learning: Emergence, convergence, and optimality. COLT 2024.

[2] Cui, Y., Ren, J., He, P., Tang, J., & Xing, Y. Superiority of multi-head attention in in-context linear regression. arXiv preprint arXiv:2401.17426. 2024.

**Theoretical Claims:**

I have checked the sketch of the proofs for the main theorems which are shown in Appendix. I did not observe mistakes in the proofs.

---

> ### Author Rebuttal · Authors · 2025-03-31
>
> Thank you for your comments and assessments. Here's our response to the comments.
>
> **On Comparision with Existing Work.**
> - **Comparison with [1].**
>     - (`[1] corresponds to single-head case in our paper`) [1] considers the multi-head attention for **multi-task** linear regression, where the number of heads equals the number of tasks. Under a specialized initialization, [1] proves that **each head learns to handle one task**,  corresponding to the **single-head case** in our paper. In contrast, we consider a more flexible setup in which multiple heads can be freely allocated to a single task, allowing the model architecture to be more expressive and complex. As a result, while [1] shows that the single-head model learns a **nonparametric** predictor with KQ scaling as $1/\sqrt{d}$, we demonstrate that the multi-head model learns a **parametric** GD predictor with KQ converging to $0^+$. Hence, the analysis are **fundamentally different**.
>     - (`[1] also adopts reparametrization`) Note that [1] assumes decomposable weights (Definition 3.1 in [1]), which ensures that the eigenvector space remains fixed. Thus, the analysis in [1] also employs a **reparameterization**, showing that diagonalizability is preserved along the gradient flow and focusing on the eigenvalues, i.e., the diagonal entries in the isotropic case.
> - **Comparison with [2].**
>     - (`Empirical Insights`) [2] only identifies the diagonal KQ patterns with potentially positive and negative values for **two-head** attention, as well as identical performance when the number of heads exceeds two. We further **quantitively** figure out the detailed **sign-matching**, **homogeneous KQ magnitude**, and **zero-sum OV** patterns beyond $H=2$, and reveals that the multi-head softmax attention learns to implement **GD predictor**, which are not covered in [2]. Thus, we establish a more complete understanding.
>     - (`Theoretical Insights`) We provide a comprehensive explanation by analyzing *how patterns emerge* through **training dynamics**, as well as *how the trained model operates* via a **function approximation and optimality analysis**. While [2] considers full-model parameterization, it focuses solely on a loss approximation similar to our Proposition 4.1. However, it does not capture the training dynamics and function approximation----that we develop based on the loss approximation results. Moreover, the main result in [2]—the superiority of multi-head over single-head attention—is a **natural corollary** of our findings, since we  show that single-head attention learns a nonparametric kernel regressor, whereas multi-head attention learns a parametric GD. By applying the approximate loss from Proposition 4.1 with $H = 1$ and $H = 2$, we can reproduce the result. Finally, [2] only analyze KQ parameters with OV parameters fixed to the corresponding optimal solution, while we analyze with both KQ and OV parameters.
> - **Comparison with [3].** [3] focus on the binary classification tasks while we work on the regression task, marking a clear distinction.
>
> **On Real-Word Implications.** Understanding what transformers learn for specific task serves as a starting point for investigating general ICL meachanism. The theoretical and emppirical insights lay the foundation for understanding how deep models learn language.
>
> **On Transformer Architecture.** In this paper, we aim to understand the mechanism of the **standard** multi-head attention architecture and the weighted sum of head is beyond the scope. Also, $\mu^{(h)}$'s naturally act as "weights" for each head, and are expected to learn the "optimal" weight during the training. Thus, we do not anticipate any improvement.
>
> **On Reparametrized Model.** As shown in Observation 4 of §3, the attention model develops a diagonal-only pattern in the KQ circuit and a last-entry only pattern in the OV circuit during the **early stages of training** and then continues optimizing within this regime. This indicates that a diagonal parameterization is sufficient to capture the core behavior of model training. Importantly, we **do not impose any diagonal constraints** on the KQ parameters—the diagonal pattern **emerges naturally** during **full-model** training and persists throughout (see Figure 4). Moreover, if the KQ parameters are initialized as a diagonal matrix, this structure is preserved over the course of training.
>
> We thank the reviewer for pointing out the related work, and we will incorporate these comparisons in the revised version.
>
> [1] Chen, S., Sheen, H., Wang, T., & Yang, Z.. Training dynamics of multi-head softmax attention for in-context learning: Emergence, convergence, and optimality. COLT 2024.
>
> [2] Cui, Y., Ren, J., He, P., Tang, J., & Xing, Y. Superiority of multi-head attention in in-context linear regression. 2024.
>
> [3] Li, H., Wang, M., Lu, S., Cui, X., & Chen, P. Y. How do nonlinear transformers learn and generalize in in-context learning? ICML 2024

---

> > ### Comment · Reviewer_Y3fQ · 2025-04-07
> >
> > Thanks for the response! Some of my concerns are resolved and I update the score accordingly. However, I believe in the current version, the discussion of comparison with existing works is insufficient. Through quick search, I found additional related works which are missing, e.g., [1,2]. Together with postponing related work section in appendix, all these could mislead the readers in understanding the contribution of this paper. Could you please put the related work section in the main paper, and revise it comprehensively? I understand that ICML does not allow revising pdf directly. I would like to at least take a look at the revised paragraphs. Then I would like to further update the score.
> >
> > [1] Linear Transformers with Learnable Kernel Functions are Better In-Context Models
> >
> > [2] In-Context Learning of Polynomial Kernel Regressionin Transformers with GLU Layers

---

> > > ### Author Response · Authors · 2025-04-08
> > >
> > > We thank the reviewer for pointing out the overlooked related work. Below is the revised discussion, which we will incorporate into the main article in the updated version. Due to word constraints in the rebuttal, we may omit the full list of references here.
> > >
> > > **In-Context Linear Regression.** To better understand how transformers acquire ICL abilities, researchers study the linear regression task, examining both the model’s expressive power and training dynamics. The pioneering work of Garg et al. (2022) empirically investigates the performance of the transformer on linear regression, demonstrating that transformers achieve near Bayes-optimal results. Von Oswald et al. (2023) studies a simplified linear transformer and reveals that it learns to
> > > implement a gradient-based inference algorithm. From a theoretical perspective, Zhang et al. (2024); Ahn et al. (2023a) show that one-layer linear attention provably learns to perform preconditioned gradient descent, using training dynamics and loss landscape analysis, respectively. Furthermore, Chen et al. (2024a) provides the first theoretical insight into standard softmax-based attention, showing that under certain initialization schemes, the trained model converges to a kernel regressor. Concurrently, Cui et al. (2024) examine how multi-head softmax attention learns linear regression in context, identifying a learned diagonal KQ pattern with both positive and negative entries from experimental perspective. In addition, Bai et al. (2024) explores the expressive power of transformers to implement various linear regression algorithms. Recent studies also examine how transformers handle variants of linear regression, including two-stage least squares for addressing data with endogeneity (Liang et al., 2024), adaptive algorithms for sparse linear regression (Chen et al., 2024c), EM-based learning of mixture of linear models (Jin et al., 2024), and multi-step gradient descent within the loop transformer architecture (Gatmiry et al., 2024). Besides, Aksenov, et al. (2024) and Sun, et al. (2025) are also broadly related to our work, in which they investigate the role of the nonlinear softmax activation within the context of regression tasks.
> > >
> > > **Comparison with Related Work.** We provide a detailed discussion of the differences between our work and that of Chen et al. (2024a) and Cui et al. (2024), which are among the most closely related studies. Different from our setup, Chen et al. (2024a) considers multi-head attention in the context of multi-task linear regression, where the number of heads matches the number of tasks. Under a specialized initialization, they show that each head independently learns to solve a distinct task--effectively reducing to a single-head per task setup, which corresponds to the single-head case in our framework. In contrast, our setup allows multiple heads to be flexibly allocated to a single task, enabling a more expressive and complex model architecture. As a result, while Chen et al. (2024a) shows that the single-head model learns a nonparametric, kernel-type predictor with scaling of KQ parameters as $\Theta(1/\sqrt{d})$, we demonstrate that the multi-head model instead learns a parametric gradient descent predictor with KQ converging to $0^+$. This not only recovers the known results for linear attention (e.g., Zhang et al., 2024) but also reveals that multi-head softmax attention can outperform the single-head one by effectively encoding the linear architecture through an explicit approximation.
> > >
> > > The work of Cui et al. (2024) identifies the diagonal KQ patterns with potentially positive and negative values in two-head softmax attention, and observe identical performance when the number of heads exceeds two. We go further by quantitatively characterizing the learned model. Specifically, we reveal detailed sign-matching, homogeneous KQ magnitudes, and zero-sum OV patterns for head counts beyond $H=2$, and show that multi-head softmax attention effectively learns to implement a gradient descent predictor. From a theoretical perspective, Cui et al. (2024) adopts full-model parameterization and conducts a loss landscape analysis. In contrast, we begin by establishing an approximate loss and then develop a comprehensive explanation based on training dynamics, function approximation, and optimality analysis. Our results reinforce and go beyond the core argument in Cui et al. (2024) regarding the superiority of multi-head over single-head attention: we not only compare the testing loss but also explicitly demonstrate that single-head attention learns a nonparametric kernel regressor, while multi-head attention learns a more powerful parametric gradient descent predictor.

---

### Official Review · Reviewer_M7UK · 2025-03-14

**Overall Recommendation:** 3

**Summary:**

This paper investigates the training dynamics of multi-head softmax attention in in-context learning. Through experimental analysis, the authors discover two key patterns: (1) QK weight matrices develop a diagonal structure, with diagonal elements being nearly uniform across all heads, and (2) QK weights and effective OV weights share the same sign, with their average approaching zero. Based on these findings, they propose a simplified training model with 2H parameters and describe the training process in two distinct stages. The authors also analyze the expressive power of their simplified multi-head attention model, comparing it with single-head attention, and validate their findings through extensive experimentation.

**Claims And Evidence:**

The claims are well-supported by both theoretical analysis and empirical evidence.

**Essential References Not Discussed:**

No.

**Experimental Designs Or Analyses:**

The experimental designs are robust and well-constructed, providing sufficient evidence to support the paper's conclusions.

**Methods And Evaluation Criteria:**

The methodology and evaluation criteria are sound and appropriate.

**Other Comments Or Suggestions:**

If there are any misunderstandings on my part, please point them out, and I will reconsider my evaluation of this work.

**Other Strengths And Weaknesses:**

**Strength**:

1. The paper effectively uses experimental analysis to reveal parameter patterns in trained softmax attention models, providing insights into both training dynamics and operational mechanisms.
2. The research offers a thorough comparative analysis of softmax and multi-head attention architectures.

** Weaknesses**:

1. The analysis of multi-head attention training dynamics relies heavily on intuitive reasoning, with Appendix C limiting its formal analysis to two-head attention only.

**Questions For Authors:**

Regarding the apparent discrepancy between theoretical predictions and experimental results: While the analysis in Appendix C suggests w should approach 0, Figures 2(b) and 1(b) show convergence to a non-zero constant. Could you explain this apparent contradiction?

**Relation To Broader Scientific Literature:**

This research contributes to our understanding of large language models by examining how multi-head softmax attention networks learn to perform in-context linear regression, offering valuable insights into LLM training processes and mechanisms.

**Theoretical Claims:**

While I haven't examined all proofs in detail, the theoretical framework appears well-documented and logically sound.

---

> ### Author Rebuttal · Authors · 2025-03-31
>
> We sincerely thank the reviewer for the positive feedback and appreciation of our work. Here's our response to the questions.
>
> **On Heuristic Derivation and Two-Head Gradient Flow.**
> We acknowledge that in Section 4.2, we employ a heuristic derivation to explain how the observed patterns emerge during gradient optimization, using an expansion-based argument. To establish a more rigorous foundation, we analyze the gradient flow in Appendix C to explain why these patterns persist once they emerge and to characterize the trajectory of the parameter evolution.
> - (`Multi-head Attention Effectively Acts as Two-head Model`) Furthermore, our experiments (see Figure 3) demonstrate that a multi-head model essentially implements the same estimator as a two-head model. As detailed in Equation (4.5), on the solution manifold the multi-head model ultimately separates into two distinct head types, determined by the signs of $\mu^{(h)}$ and $\omega^{(h)}$. Within each group, the heads behave equivalently to a single head with $\omega = \omega^{(h)}$ and a $\mu$ equal to the sum of the $\mu^{(h)}$ values within that group. Thus, analyzing the dynamics of a two-head model provides all the essential insights.
> - (`Dynamics are Similar Under Symmetry`) Besides, note that if we initialize a multi-head model with identical absolute values for $\mu$ and $\omega$ (see Definition C.1) and randomly assign each head a positive or negative sign, then as the number of heads increases, each type group is highly likely to contain nearly the same number of heads. By symmetry, tracking the effective KQ and OV parameters within each group is equivalent to analyzing a two-head model.
>
>
> **On Discrepancy between Theoretical Predictions and Experimental Results.**
> - (`Experimental Justification: Small KQ already Behaves as the Limiting Model`)  As shown by the theoretical results in Section 4.3, when KQ approaches 0, the model exactly implements the GD predictor. Therefore, we compare the performance—measured in terms of mean squared error (MSE)—of the learned model with small KQ against that of the GD predictor. Figure 3(b) demonstrates that the loss curves for the learned transformer model and the GD predictor coincide, indicating **identical performance** and confirming that the approximation (see Section 4.3) holds well even for a small constant level KQ. Moreover, because the model is trained via gradient optimization, the **negligible loss gap** between the small KQ multi-head attention and its limiting case suggests that further optimization from a small constant KQ to the limit of 0 would require an exceptionally long time —likely exceeding the number of training steps used in our experiments.
> - (`Potential Meachanism: Edge of Stability`) In practice, the particular magnitudes of KQ parameters learned by the optimization algorithm depends on particular choices of learning rate, training steps, and batch size (if SGD or Adam is applied). This is partly due to the *"river valley"* loss landscape near the global minimum (see Figure 5(b)). From the theoretical perspective, this phenomenon can be attributed to the **edge of stability** (e.g., [1]), which is commonly observed in neural network training. In this regime, gradient descent progresses non-monotonically, oscillating between the “valley walls” of the loss surface and failing to fully converge to the minimum. Experimental results indicate that choosing a *smaller learning rate*, a *larger batch size*, and *sufficiently many training steps* can drive the KQ parameters to eventually approach zero.
>
> We would thank the reviewer for the nice questions and we will incorporate these clarifications in the revised version.
>
> [1] Cohen, J. M., Kaur, S., Li, Y., Kolter, J. Z. and Talwalkar, A. (2021). Gradient descent on neural networks typically occurs at the edge of stability. arXiv preprint arXiv:2103.00065.

---

### Decision · Program_Chairs · 2025-05-01

**Decision:**

Accept (poster)

**Comment:**

This paper investigates the training dynamics of multi-head softmax attention in in-context learning. The reviews for the paper are mostly positive. While reviewers expressed concerns regarding novelty and relationship with earlier works, they were mostly resolved during the rebuttal process. I recommend acceptance and encourage the authors to address all the reviewer's concerns in the final version.